

# The ESCAPE project: Energy-efficient Scalable Algorithms for Weather Prediction at Exascale

Andreas Müller[1], Willem Deconinck[1], Christian Kühnlein[1], Gianmarco Mengaldo[1], Michael Lange[1], Nils Wedi[1], Peter Bauer[1], Piotr K. Smolarkiewicz[1], Michail Diamantakis[1], Sarah-Jane Lock[1], Mats Hamrud[1], Sami Saarinen[1], George Mozdzynski[1], Daniel Thiemert[1], Michael Glinton[2], Pierre Bénard[2], Fabrice Voitus[2], Charles Colavolpe[2], Philippe Marguinaud[2], Yongjun Zheng[2], Joris Van Bever[3], Daan Degrauwe[3], Geert Smet[3], Piet Termonia[3,4], Kristian P. Nielsen[5], Bent H. Sass[5], Jacob W. Poulsen[5], Per Berg[5], Carlos Osuna[6], Oliver Fuhrer[6], Valentin Clement[7], Michael Baldauf[8], Mike Gillard[9], Joanna Szmelter[9], Enda O'Brien[10], Alastair McKinstry[10], Oisín Robinson[10], Parijat Shukla[10], Michael Lysaght[10], Michał Kulczewski[11], Milosz Ciznicki[11], Wojciech Piątek[11], Sebastian Ciesielski[11], Marek Błażewicz[11], Krzysztof Kurowski[11], Marcin Procyk[11], Pawel Spychala[11], Bartosz Bosak[11], Zbigniew Piotrowski[12], Andrzej Wyszogrodzki[12], Erwan Raffin[13], Cyril Mazauric[13], David Guibert[13], Louis Douriez[13], Xavier Vigouroux[13], Alan Gray[14], Peter Messmer[14], Alexander J. Macfaden[15], and Nick New[15]

[1]European Centre for Medium-Range Weather Forecasts (ECMWF), Shinfield Park, Reading RG2 9AX, UK
[2]Centre National de Recherches Météorologiques, Météo-France, Toulouse, France
[3]Royal Meteorological Institute (RMI), Ringlaan 3, Brussels, Belgium
[4]Department of Physics and Astronomy, Ghent University, Ghent, Belgium
[5]The Danish Meteorological Institute (DMI), Copenhagen, Denmark
[6]Federal Office of Meteorology and Climatology MeteoSwiss, Zurich, Switzerland
[7]Center for Climate System Modeling, Zurich, Switzerland
[8]Deutscher Wetterdienst (DWD), Offenbach, Germany
[9]Loughborough University, Leicestershire LE11 3TU, UK
[10]Irish Centre for High-End Computing (ICHEC), National University of Ireland, Galway, Ireland
[11]Poznan Supercomputing and Networking Center (PSNC), Jana Pawła II 10 Street, 61-139, Poznań, Poland
[12]Institute of Meteorology and Water Management - National Research institute (IMGW-PIB), Podleśna 61, Warsaw, Poland
[13]Bull, an ATOS company, Bezons, France
[14]NVIDIA Switzerland, Technoparkstr. 1, 8005 Zurich, Switzerland
[15]Optalysys Ltd., Flemming Court, Whistler Drive, Glasshoughton, WF10 5HW, UK

**Correspondence:** Andreas Müller (andreas.mueller@ecmwf.int)

**Abstract.** In the simulation of complex multi-scale flow problems, such as those arising in weather and climate modelling, one of the biggest challenges is to satisfy operational requirements in terms of time-to-solution and energy-to-solution yet without compromising the accuracy and stability of the calculation. These competing factors require the development of state-of-the-art algorithms that can optimally exploit the targeted underlying hardware and efficiently deliver the extreme computational

5   capabilities typically required in operational forecast production. These algorithms should (i) minimise the energy footprint along with the time required to produce a solution, (ii) maintain a satisfying level of accuracy, (iii) be numerically stable and resilient, in case of hardware or software failure.





The European Centre for Medium Range Weather Forecasts (ECMWF) is leading a project called ESCAPE (Energy-efficient SCalable Algorithms for weather Prediction on Exascale supercomputers) which is funded by Horizon 2020 (H2020) under initiative Future and Emerging Technologies in High Performance Computing (FET-HPC). The goal of the ESCAPE project is to develop a sustainable strategy to evolve weather and climate prediction models to next-generation computing technolo-
gies. The project partners incorporate the expertise of leading European regional forecasting consortia, university research, experienced high-performance computing centres and hardware vendors.

This paper presents an overview of results obtained in the ESCAPE project in which weather prediction have been broken down into smaller building blocks called dwarfs. The participating weather prediction models are: IFS (Integrated Forecasting System), ALARO – a combination of AROME (Application de la Recherche à l'Opérationnel à Meso-Echelle) and ALADIN
(Aire Limitée Adaptation Dynamique Développement International) and COSMO-EULAG – a combination of COSMO (Consortium for Small-scale Modeling) and EULAG (Eulerian/semi-Lagrangian fluid solver). The dwarfs are analysed and optimised in terms of computing performance for different hardware architectures (mainly Intel Skylake CPUs, NVIDIA GPUs, Intel Xeon Phi). The ESCAPE project includes the development of new algorithms that are specifically designed for better energy efficiency and improved portability through domain specific languages. In addition, the modularity of the algorithmic
framework, naturally allows testing different existing numerical approaches, and their interplay with the emerging heterogeneous hardware landscape. Throughout the paper, we will compare different numerical techniques to solve the main building blocks that constitute weather models, in terms of energy efficiency and performance, on a variety of computing technologies.

# 1   Introduction

Numerical weather prediction (NWP) has made significant progress over the past decades (Bauer et al., 2015). Nevertheless,
the prediction of extreme weather events, like heavy precipitation patterns which could lead to flooding and tropical cyclones, is still far from being satisfying. An improvement in the forecast of these events would have a huge economic and societal benefit. One of the main sources of errors in NWP is insufficient resolution. Global NWP and climate simulations for the foreseeable future will not reach what may be considered "sufficient resolution", which for practical NWP purposes may be in the 1km global resolution range. As a result a substantial part of the cost of a model timestep is used in physical parameterisations that
describe the effect of unresolved processes on the resolved scale, but also to describe diabatic effects such as radiation and water phase changes.

One of the key components of the ESCAPE project is to guarantee the continued efficiency of NWP models while transitioning to emerging computing architectures including accelerators. A particular challenge arises from the need to achieve time-to-solution and energy-to-solution efficient solutions for operating global, complex, high-resolution, ensemble based sys-
tems on high-performance computers so that they will remain affordable given tight operational schedules. A comprehensive assessment of the modelling infrastructure of ECMWF's Integrated Forecasting System (IFS) can be found in Wedi et al. (2015). The authors there also stress a need to enhance model complexity where it improves forecast skill through fully cou-



pled simulations of the atmosphere with ocean, sea-ice and land surfaces and including interactive chemical processes in the atmosphere.

Beside the need for increased model complexity we also see a need for significantly increased resolution. It is ECMWF's strategic goal by 2025 to establish the efficiency required to run at least twice daily global ensembles at 5km globally uniform
resolution. The potential scope of such an ensemble is illustrated best in the context of extreme events such as tropical storm Irma. Experiments with a 5km resolution ensemble show dramatic improvement in the forecast of the wind intensity compared to the currently operational resolution of 18km (Magnusson et al., 2018). ECMWF is world leading in terms of track forecast accuracy and increasing the resolution to 5km does not compromise the accuracy of the track forecast.

The supercomputers used for NWP have changed dramatically over the past decades. Many of the algorithms used in NWP
were designed before the multi-core era started. Optimisations have been continuously developed with every new computer that was bought by NWP centres. These optimisations were usually limited to adapting the same FORTRAN code base. The envisioned increase in the resolution of NWP models together with the increased model complexity requires a much more radical redesign of the algorithms and codes used for weather prediction.

We start with introducing the concept of Weather & Climate Dwarfs in Section 2. In Section 3 we present our work on
optimising codes used in the dynamical cores and Section 4 shows results for physics parameterisations. The paper ends with a comparison between different methods in Section 5 and conclusions in Section 6.

## 2   The dwarf concept

Weather prediction and climate models are too big to adapt to new computing architectures at once. For this reason we disassemble the model into smaller building blocks which we call Weather & Climate Dwarfs (Figure 1), in analogy to the Berkeley
Dwarfs (Asanović et al., 2006). We define a Weather & Climate Dwarf as a key functional component of an earth system model. Each dwarf is associated with specific computation and communication patterns which lead to characteristic bottlenecks relevant for the performance of the overall model. In the ESCAPE project we adapt these dwarfs to different hardware architectures. The knowledge gained in this adaptation is used to research alternative numerical algorithms which are better suited for those new architectures. Eventually the goal is to speedup the entire simulation by using the optimised dwarfs in the
full model.

### 2.1   The data structure framework Atlas

To avoid code duplication across dwarfs, we use a data structure framework called Atlas, that handles both the mesh generation and parallel communication aspects. The main idea behind Atlas (library developed at ECMWF, Deconinck et al., 2017) is illustrated in Figure 2a: Atlas provides functionality to generate the grid, create the mesh and partition it. The partitioning step
also generates halos (overlapping regions between different MPI partitions to enable stencil operations in a parallel context), if needed by the numerical method. Atlas provides *function spaces* for different numerical methods as well as storage objects





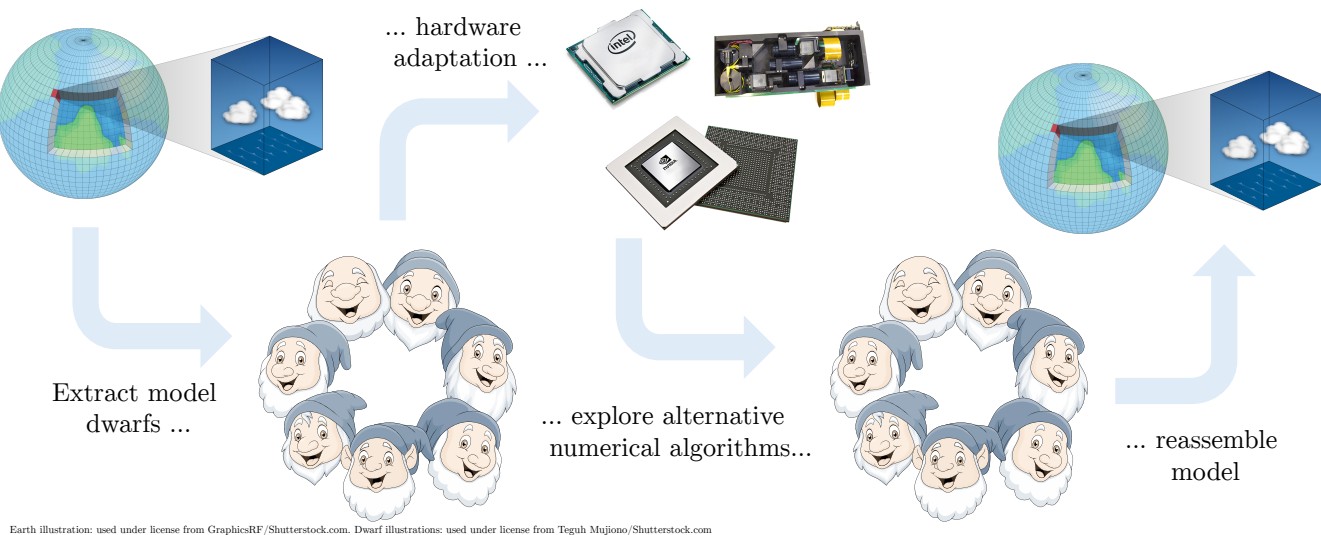

Earth illustration: used under license from GraphicsRF/Shutterstock.com. Dwarf illustrations: used under license from Teguh Mujiono/Shutterstock.com

**Figure 1.** Illustration of the main idea behind the ESCAPE project. The entire model is broken down into smaller building blocks called dwarfs. These are adapted to different hardware architectures. Based on the feedback from hardware vendors and high performance computing centres alternative numerical algorithms are explored. These improvements are eventually built into the operational model.

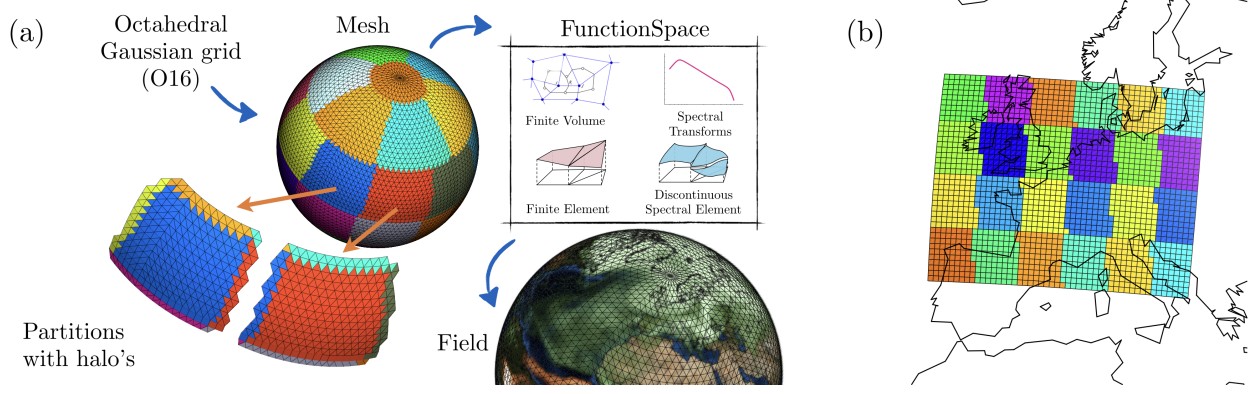

**Figure 2.** (a) Illustration of the concept behind the data structure framework Atlas. Further explanation can be found in the main text. (b) Example of the limited area extension of Atlas. Shown here is a Lambert projection centred around 4° longitude and 50° latitude. The grid consists of 50 grid cells in zonal direction, 40 grid cells in meridional direction and the resolution is in both directions 0.5°. Color shading shows the domain decomposition.

called *fields*, which can be used by the dwarfs to store their data. More information about Atlas can be found in Deconinck et al. (2017).

In the ESCAPE project, Atlas has been improved and extended by adding support for limited area grids (Figure 2b) and adding support for accelerators through GridTools (Deconinck, 2017a, b).



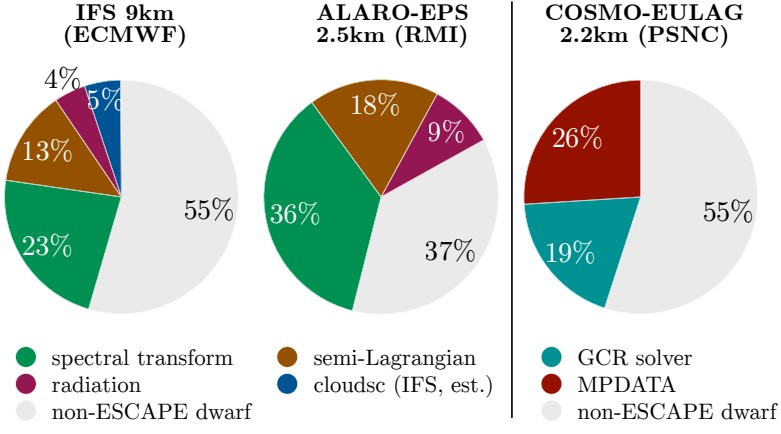

**Figure 3.** Portion of the forecast runtime spent in ESCAPE dwarfs for the three different models IFS (left), ALARO ensemble prediction system (EPS) (middle) and COSMO-EULAG (right). The measurements for IFS were taken during an operational run on 352 nodes (1408 MPI processes, 18 OpenMP threads per process). The limited area models ALARO-EPS and COSMO-EULAG used each 576 MPI processes for the simulations shown here.

## 2.2 List of dwarfs

Table 1 gives an overview of the dwarfs defined in the ESCAPE project. These dwarfs have been chosen because they have a significant contribution to the overall runtime of many weather forecasting systems (Figure 3). In addition they represent fundamental computational patterns that are also relevant for those parts of the model that are not explicitly covered by these dwarfs.

Each of the released dwarfs has been documented (Mengaldo, 2016; Müller et al., 2017) and can have different implementations. Table 1 lists the models from which the dwarf code originated. The models include the spectral IFS, the IFS-finite volume module (IFS-FVM), ALARO/AROME and COSMO-EULAG. As part of the ESCAPE project we created a new global shallow water model called GRASS (Global Reduced A-grid Spherical-coordinate System). This model is introduced in Section 3.6. Table 1 shows which dwarfs are using Atlas for handling the mesh. This table also shows the programming model adopted

for each dwarf. In particular, we used MPI for distributed-memory parallelism, OpenMP and OpenACC for shared-memory parallelism, and DSL, that stands for domain-specific language and uses GridTools. Also note that the spectral transform on the sphere with spherical harmonics and in limited area domains with biFFT has also been implemented with the optical processor by Optalysys (Macfaden et al., 2017). We will give an overview of the work with the optical processor in Section 3.1.

The work on the hardware adaptation and performance optimisation focused mainly on three of our dwarfs: the spherical

harmonics spectral transform dwarf, the MPDATA advection dwarf and the ACRANEB2 radiation dwarf. The work on the radiation dwarf has already been described in detail in Poulsen and Berg (2017). We will give a short summary of that work in Section 4.1 and focus our description in Section 3 on the work on the spectral transform and MPDATA advection which has not been published elsewhere. For the same reason we will also describe the algorithms and computational challenges for these two dwarfs in greater detail.



**Table 1.** Overview of work performed in the ESCAPE project, the corresponding Section in this paper and the models from which the dwarfs originated. The column Atlas shows which of the dwarfs are based on the data structure framework Atlas (see Section 2.1). Dwarfs for which MPI and OpenMP/OpenACC is available both can be used together as hybrid parallelisation. LAITRI is used in the full model in each MPI process individually and therefore does not include MPI parallelisation inside the dwarf. Further explanations can be found in the text.

| | Section | IFS (spectral) | IFS-FVM | ALARO/AROME | COSMO-EULAG | GRASS | dwarf released | Atlas (Section 2.1) | MPI | OpenMP | OpenACC | DSL (Section 3.5.4) | Optalysys (Section 3.1.5) |
|---|---|---|---|---|---|---|---|---|---|---|---|---|---|
| **Dynamics dwarfs:** | 3 | | | | | | | | | | | | |
| • time-stepping: | | | | | | | | | | | | | |
| spectral transform: | | | | | | | | | | | | | |
| – spherical harmonics | 3.1 | ✓ | | | | | ✓ | ✓ | ✓ | ✓ | ✓ | | ✓ |
| – biFFT | 3.1 | | | ✓ | | ✓ | | | ✓ | ✓ | ✓ | | ✓ |
| elliptic solver - GCR | 3.2 | | ✓ | | ✓ | | ✓ | ✓ | ✓ | ✓ | | | |
| elliptic solver - multigrid | 3.2 | | ✓ | | | | ✓ | ✓ | ✓ | ✓ | | | |
| HEVI | 3.3 | | | | | ✓[1] | | | ✓ | ✓ | | | |
| • advection: | | | | | | | | | | | | | |
| semi-Lagrangian | 3.4 | ✓ | | ✓ | | | ✓ | ✓ | ✓ | ✓ | | | |
| – LAITRI (3d interpolation) | 3.4 | ✓ | | ✓ | | | ✓ | ✓ | | ✓ | ✓ | | |
| MPDATA | 3.5 | | ✓ | | ✓ | | ✓ | ✓ | ✓ | ✓ | ✓ | ✓ | |
| • gradient computation: | | | | | | | | | | | | | |
| high order finite difference | 3.6 | | | | ✓ | | | | ✓ | ✓ | | | |
| finite volume | 5 | | ✓ | | ✓ | | | ✓ | ✓ | ✓ | | | |
| **Physics dwarfs:** | 4 | | | | | | | | | | | | |
| • radiation scheme: | | | | | | | | | | | | | |
| ACRANEB2 | 4.1 | | | ✓ | | | ✓ | | ✓ | ✓ | ✓ | | |
| • cloud microphysics: | | | | | | | | | | | | | |
| CLOUDSC | 4.2 | ✓ | ✓ | | | | ✓ | | ✓ | ✓ | ✓ | | |

[1] envisioned for the future, currently only shallow-water

    In NWP, we distinguish between different components of our weather prediction systems. One of the main components is data assimilation which computes the initial conditions by using observations and previous forecasts. We call the component



that takes this initial condition and computes the future weather the forecast component. As mentioned before, the dwarfs in Table 1 have a significant contribution to the overall runtime of the forecast component. The ESCAPE project has not explicitly covered code used for data assimilation. However, several algorithmic motifs found in data assimilation can be found in the collection of dwarfs. In addition the cost of data assimilation depends very much on the method that is used. Some NWP centres

use 3DVAR like the new ensemble prediction system ensemble prediction system developed at RMI in Belgium (RMI-EPS). Our studies in ESCAPE have shown that for RMI-EPS, 99% of the energy consumption is spent in the forecast component. Data assimilation takes less than 1% of the energy consumption even though it uses 35% of the runtime. The reason for this is that most computations inside the data-assimilation in RMI-EPS are running on a small number of nodes which gives them a relatively large contribution in terms of runtime and very small contribution in terms of energy consumption. More details

about this study of RMI-EPS can be found in Van Bever et al. (2018b). The 4DVAR data assimilation used at ECMWF is much more expensive but most of the runtime of 4DVAR is spent in linearised versions of the forecast model (Hamrud, 2010). The dwarfs represent therefore also significant components of the 4DVAR data assimilation at ECMWF.

## 2.3 Simple theoretical model for using accelerators

Before we describe how the different dwarfs have been adapted and optimised for different hardware architectures, we first

discuss the general benefit of using hybrid-machines with CPUs and accelerators in a simple cost model. It is well known that data transfer between CPUs and accelerators on hybrid-machines can take a significant amount of time and needs to be avoided as much as possible. Data transfer can be avoided to some degree by running the entire model on the accelerator and transferring only the initial condition at the beginning of the simulation to the accelerator and the necessary output back to the CPU when required. In our simple cost model we show that even if we neglect the overhead caused by data transfer we still get

a significant overhead if the computation on the CPU cannot overlap with computation on the accelerator. Unfortunately many NWP models are currently not able to overlap these computations.

The derivation of our simple cost model can be found in the Appendix A. This cost model considers specifically the case of the NWP centres where the runtime is fixed and the number of processors need to be adjusted to reach the given runtime. If the simulation started earlier we would not have all of the necessary observations and if the simulation finished later the

result would not reach our customers on time. To get some practical numbers from our theoretical model we use the situation at Météo-France as an example. If we start with a CPU-only code and port a small part of the code to the accelerator the total cost per time-step first increases if we cannot overlap computations on the CPU with computations on the accelerator (solid lines in Figure 4a). This overhead can be avoided if we find a way to overlap computations on the CPU with computations on the accelerator (dashed lines in Figure 4a).

The sensitivity of the results on the scaling efficiency of the model is shown in Figure 4b. The cost decreases with increasing strong scaling efficiency. Our simple cost model assumes that CPU and accelerator part of the code have the same strong scaling efficiency. Under this assumption fast accelerators would have a larger benefit for code that has worse scaling efficiency because the speedup of the accelerator would allow to reduce the number of devices. Having the same scaling efficiency on CPU and





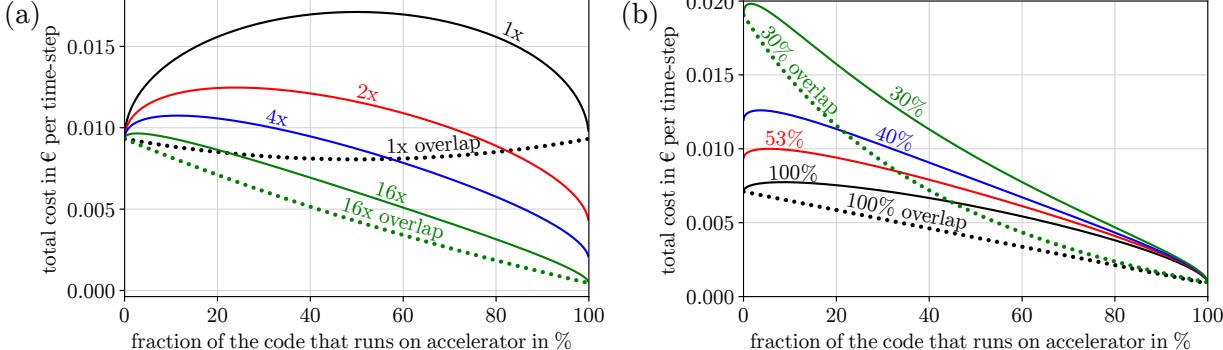

**Figure 4.** Impact of (a) speedup of the accelerator and (b) strong scaling efficiency on the total cost of one time-step as a function of how much of the code is running on the accelerator. This Figure is based on the specific case of the AROME model at Météo-France. The strong scaling efficiency of 53% (used for all curves in (a) and for the red curve in (b)) at 1000 CPU sockets compared to one CPU socket was found through linear regression experimentally and should therefore be the most realistic case for the AROME model. Dotted lines illustrate the result for two of the curves if we assume perfect overlap between computations on the CPUs and accelerators while the solid lines assume no overlap. In (b) we assume an accelerator speedup of 8x.

accelerator will be difficult to achieve in reality because accelerators usually require a lot of inherent parallelism inside each device.

We conclude from this simple cost model that having no overlap between computations on the CPU and on the accelerator can produce a significant overhead. This overhead can be minimised by running most of the NWP model on the accelerator

(as done in Fuhrer et al., 2018). At the same time we should invest in research on overlapping computations on CPUs and accelerators. Overlapping these computations is not part of the work done in the ESCAPE project. We plan to investigate this overlapping in our future research.

In the next two sections, we introduce the dynamics dwarfs (Section 3) and the dwarfs associated to the physical parametri-sations (Section 4). This conceptual categorisation follows the rationale adopted for classifying the various dwarfs mentioned

above and reported in Table 1.

## 3  Dynamics dwarfs

This section describes the work done in the ESCAPE project on the dynamics dwarfs which are used in the dynamical cores of the models. The subsections follow the list of dynamics dwarfs as given in Table 1. Three important features shared by these dwarfs are: a.) solution of the governing equations in the classical meteorological latitude-longitude coordinates, b.) compati-

15 bility with reduced Gaussian grids, c.) co-located (i.e. unstaggered) arrangement of the prognostic variables with respect to the grid. Having these shared features between the dwarfs facilitates incorporation into the IFS code infrastructure.



## 3.1 Spectral transform

### 3.1.1 Background

Each time-step is split between computations in grid point space and computations in spectral space. Semi-Lagrangian advection, physical parameterisations and products of terms are computed most efficiently in grid point space while horizontal

gradients, semi-implicit calculations and horizontal diffusion are computed more efficiently in spectral space. The transform between these two spaces is performed on the sphere with spherical harmonics, that is computing these results along longitudes in a Fast Fourier transform (FFT) and a Legendre transform (LT) along latitudes. Limited area models replace the Legendre transform with another FFT which leads to the name biFFT.

In spectral transform methods such as the one used in IFS (Wedi et al., 2013), the specific form of the semi-implicit system

facilitating large time-steps (and thus time-to-solution efficiency) is derived from subtracting a system of equations, linearised around a horizontally homogeneous reference state. The solution of this linear system is greatly accelerated by the separation of the horizontal and the vertical part, which matches the large anisotropy of horizontal to vertical grid dimensions prevalent in atmospheric and oceanic models. In spectral transform methods one uses the special property of the horizontal Laplacian operator in spectral space on the sphere

$$\nabla^2 \psi_n^m = -\frac{n(n+1)}{a^2} \psi_n^m,\qquad(1)$$

where $\psi$ symbolises a prognostic variable, $a$ is the Earth radius, and $(n,m)$ are the total and zonal wavenumbers of the spectral discretisation (Wedi et al., 2013). This conveniently transforms the 3D Helmholtz problem into an array (for each zonal wavenumber) of 2D matrix operator inversions with the dimension of the vertical levels square, or in the case of treating the Coriolis term implicitly, vertical levels times the maximum truncation, resulting in a very cheap direct solve.

In this paper we focus on the computational aspects and especially on the data layout. We illustrate here the inverse spectral transform on the sphere which goes from spectral space to grid point space. The direct transform adds one numerical integration. Otherwise it works in the same way but in opposite direction.

The inverse spectral transform begins with the spectral data $\mathbf{D}(f,\mathrm{i},n,m)$ which is a function of field index $f$ (for the variables pressure, wind vorticity, wind divergence and temperature at each height level), real and imaginary part i and wave numbers

(zonal wave number $m = 0, \ldots, N_T$ and total wave number $n = 0, \ldots, N_T - m$ where $N$ is the spectral truncation). Please note that we deviate here from the usual notation where total wavenumber goes from $m$ to $N_T$ because this simplifies the separation between even and odd $n$. We use here column-major order like in Fortran, i.e. the field index $f$ is the fastest moving index and the zonal wave number $m$ is the slowest moving index. Typical dimensions can be seen in the operational high resolution (9km) forecast run at ECMWF: the number of fields is in this case 412 and the number of zonal wave numbers is given by the

truncation $N_T = 1279$. The number of latitudes is $2N_T + 2 = 2560$ and the number of longitudes increases linearly from 20 next to the poles to $4N_T + 20 = 5136$ next to the equator.

We take advantage of the symmetry of the Legendre polynomials for even $n$ and anti-symmetry for odd $n$. The coefficients of the Legendre polynomials are pre-computed and stored in $\mathbf{P}_{e,m}(n,\phi)$ for even $n$ and $\mathbf{P}_{o,m}(n,\phi)$ for odd $n$, where $\phi$ stands





for the latitudes of our Gaussian mesh. Only latitudes on the northern hemisphere are computed. Latitudes on the southern hemisphere are reconstructed from the northern latitudes as we will show later. In the same way we split the spectral data for each $m$ into even part $\mathbf{D}_{e,m}(f,\mathrm{i},n)$ and odd part $\mathbf{D}_{o,m}(f,\mathrm{i},n)$. We write variables over which we can parallelise our computations as indices. The inverse Legendre transform is performed by computing the following matrix multiplications using BLAS:

$$
\begin{aligned}
\mathbf{S}_m(f,\mathrm{i},\phi) &= \sum_n \mathbf{D}_{e,m}(f,\mathrm{i},n) \cdot \mathbf{P}_{e,m}(n,\phi), \\
\mathbf{A}_m(f,\mathrm{i},\phi) &= \sum_n \mathbf{D}_{o,m}(f,\mathrm{i},n) \cdot \mathbf{P}_{o,m}(n,\phi).
\end{aligned}
\tag{2}
$$

The resulting array for the symmetric and antisymmetric parts are now combined into the Fourier coefficients on the northern and southern hemisphere:

$$
\begin{aligned}
\phi > 0 &: \mathbf{F}(\mathrm{i},m,\phi,f) = \mathbf{S}_m(f,\mathrm{i},\phi) + \mathbf{A}_m(f,\mathrm{i},\phi), \\
\phi < 0 &: \mathbf{F}(\mathrm{i},m,\phi,f) = \mathbf{S}_m(f,\mathrm{i},\phi) - \mathbf{A}_m(f,\mathrm{i},\phi).
\end{aligned}
$$

These Fourier coefficients are finally used to compute the fields in grid point space at each longitude $\lambda$ via FFT:

$$
\mathbf{G}_{\phi,f}(\lambda) = \mathrm{FFT}(\mathbf{F}_{\phi,f}(\mathrm{i},m)).
\tag{3}
$$

### 3.1.2 Computational challenges

The computations in grid point space and spectral space require all the fields $f$ to be on the same computational node. The summation over the total wavenumber $n$ in the Legendre transform (2) makes it most efficient to have all total wavenumbers on the same node and the Fourier transform (3) over $(\mathrm{i},m)$ makes it most efficient to have all of the zonal wavenumbers $m$ with real and imaginary part on the same node. This is only possible if the data is transposed before and after the spectral transform as well as in between Legendre and Fourier transform. These transpositions produce substantial communication which increases the contribution of the spectral transform to the overall runtime for future resolutions (Figure 5).

Simplified simulations of the MPI communications performed in ESCAPE show that the strong scalability of the communication time for the spectral transform transpositions is better than for halo communication required by semi-Lagrangian advection and global norm computation commonly used in semi-implicit methods (Figure 6). The reason for the relatively good scalability of the spectral transform is that the transpositions are not global but each transposition acts only on a much smaller communicator. The transposition between the computations in spectral space and the Legendre transform is exchanging field index $f$ and total wavenumber $n$. This transposition is therefore independent between different zonal wavenumbers $m$. The transposition between Legendre and Fourier transform exchanges zonal wavenumber $m$ with latitude $\phi$ and is independent between different fields. Finally the transposition between Fourier transform and computations in grid point space exchanges longitude $\lambda$ and field index $f$ and is independent between different latitudes $\phi$. The transposition between Legendre and Fourier transform is therefore most costly because the number of independent communicators is with the number of fields much smaller than for the other transpositions. We also see this behaviour in measurements with IFS (not shown).




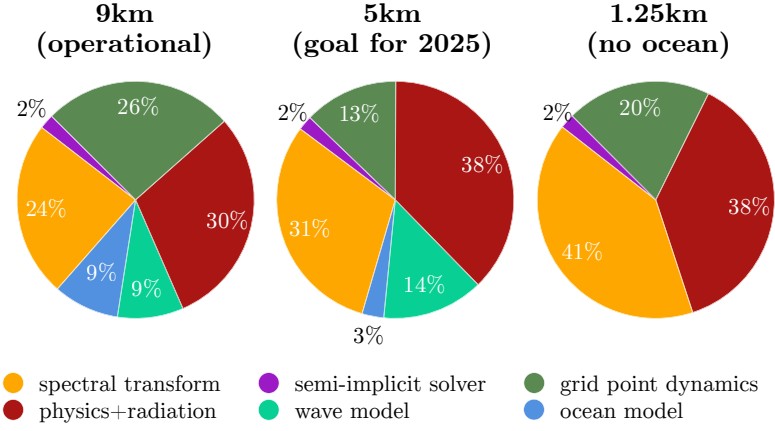

**Figure 5.** Cost profiles of the significant components of the IFS NWP model in percent of CPU time, at the operational 9km horizontal resolution with 137 vertical levels (left), the anticipated future horizontal resolution of 5km with 137 vertical levels (middle) and an experimental resolution of 1.25km with 62 vertical levels (right). The GP_dynamics represents the advection and gridpoint computations related to the dynamical core, SI_solver represents the computations and communications internal to spectral space, SP_transforms relates to the communications and computations in the transpositions from gridpoint to spectral and reverse as well as the FFT and DGEMM computations (see also spectral transform schematic below), PHYSICS+RAD relates to the cost of physical parametrisations including radiation, and finally accounting for the additional components of the wave and the ocean model. The simulation at 1.25km is without ocean and waves.

Halo communication as used in semi-Lagrangian advection and global norm computation as often used in semi-implicit methods have much worse strong scalability (Figure 6). These results indicate that halo communication will become almost as costly as the transpositions in the spectral transform if we use a very large number of MPI processes. An alternative which avoids transpositions and halo communication is given by the spectral element method shown in Müller et al. (2018) with explicit time integration in the horizontal direction. This leads to a very small amount of data that is communicated in each time-step because this method only communicates the values that are located along the interface between different processor domains. This method, however, requires much smaller time-steps which leads overall to an even larger communication volume (Figure 7). Figure 7 is based on the model comparison presented in Michalakes et al. (2015) and does not include all of the optimisations for the spectral element method presented in Müller et al. (2018). The spectral transform results are based on the operational version of IFS and do not contain the optimisations presented in this paper. Both models have significant potential for optimisation and it is not obvious if there will be a clear winner in terms of overall communication volume. The only true solution to avoid waiting time during communication is to overlap different parts of the model such that useful computation can be done while the data is communicated (Mozdzynski et al., 2015).



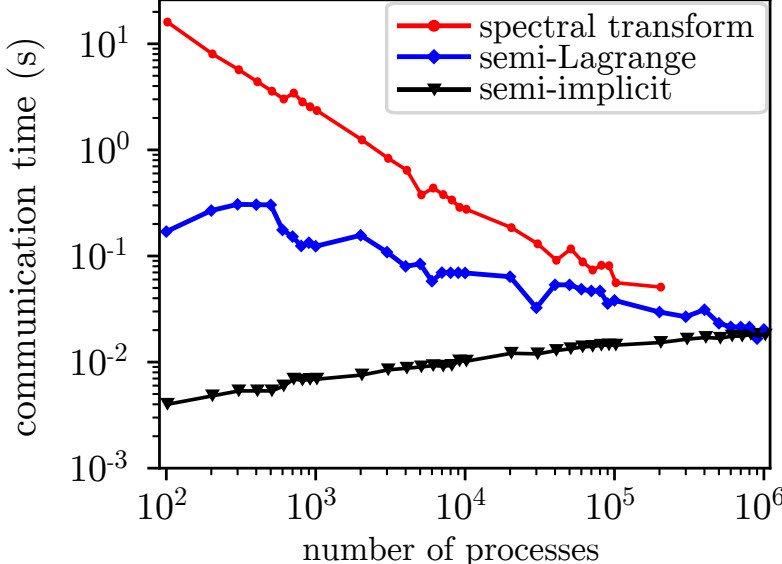

**Figure 6.** Simulation of the strong scaling of MPI communication time for minimal routing algorithm and dragonfly topology adapted from Zheng and Marguinaud (2018). The result for the spectral transform ends at $2 \times 10^5$ processes because larger number of processes exceeded the maximum runtime of the cluster used for these simulations of 24 hours. Halo communication for the semi-Lagrangian advection is assumed to use 20 elements which resembles the operational setting in IFS. The semi-implicit method uses MPI all-reduce to gather data from all processes and send the resulting norm to all of the processes.

### 3.1.3 GPU optimisation

For the GPU version, we restructure the code to: allow the grid-based parallelism to be fully exposed to the GPU in a flexible manner; ensure that memory coalescing is achieved; and optimise data management. We will now describe each of these in some more detail.

5  The grid is a two-dimensional sphere, with a third altitude dimension represented by multiple fields at each point on the sphere. The updates are inherently parallel across this grid, so all this parallelism should be exposed to the GPU to get maximal performance. However, the original implementation had a sequential loop over one of the spherical dimensions (at a high level in the call tree of the application). We re-structured the code such that, for each operation, the loops over the three dimensions became tightly nested, and when mapping these to the GPU via OpenACC directives we used the "collapse" clause to instruct

10  the compiler to collapse these to a single loop, such that it can map all inherent parallelism to hardware in an efficient manner. Similarly, for library calls it is important to maximally expose parallelism through use of the provided batching interfaces. On the GPU we perform all of matrix multiplications in the Legendre transform (2) with a single batched call of the cuBlasDgemm library. The different matrices in (2) have different sizes because the total wavenumber goes from 0 to NT. To use the fully batched matrix multiplication we pad each matrix with zeroes up to the largest size, since the library does not support differing





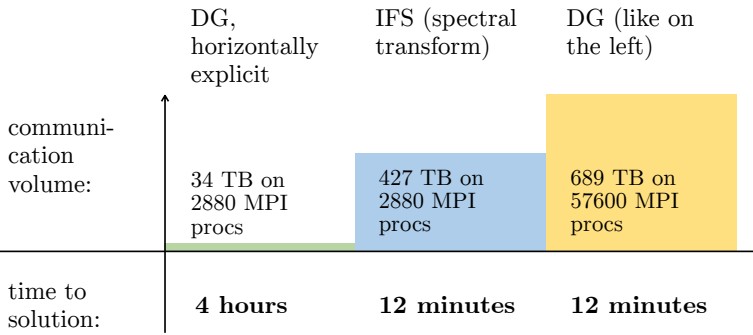

**Figure 7.** Overall communication volume comparing spectral element (SEM) from Müller et al. (2018) and the global spectral transform methods. The SEM requires a substantially lower amount of communication at the same number of cores, but due to the smaller timestep requires a much higher frequency of repeated communications for the given 2-day simulation. Increasing the number of MPI processes to achieve the same time to solution results in a larger amount of communication for the SEM. Here we assume SEM $\Delta t = 4s$; IFS $\Delta t = 240s$; communication volume is calculated for a 48 hour forecast SEM as 290.4kBytes per MPI task and $\Delta t$; IFS as 216mBytes per MPI task and $\Delta t$; IFS time-to-solution = 20 x SEM based on the performance results in Michalakes et al. (2015).

sizes within a batch. This step increases the overall number of floating point operations by almost a factor 10 but still improves the overall performance (Figure 8). We perform the FFT in equation (3) with the cuFFT library, where we batch over the altitude dimension but multiple calls are still needed over the spherical dimension (noting FFTs cannot be padded in a similar way to matrix multiplications). Therefore the code remains suboptimal here: we are still not fully exposing parallelism and

there would be scope for further improvements if a FFT batching interface supporting differing sizes were to become available.

      We restructured array layouts to ensure that multiple threads on the GPU can cooperate to load chunks of data from memory in a "coalesced" manner. This would allow a high percentage of available memory throughput. This is achieved when the fastest moving index in the multidimensional array corresponds to the OpenACC loop index occurring at the innermost level in the collapsed loop nest described above. Sometimes matrix transposes are necessary, but where possible these were pushed into

the DGEMM library calls, which have much higher-performing implementations of transposed data accesses. There remain transpose patterns within kernels involved in transposing grid point data from column structure to latitudinal (and inverse) operations, which naturally involve transposes and are thus harder to fix through restructuring. However, we optimised these using the "tile" OpenACC clause, which instructs the compiler to stage the operation through multiple relatively small tiles which can perform the transpose operations within fast on-chip memory spaces, such that the accesses to global memory are

much more regular.





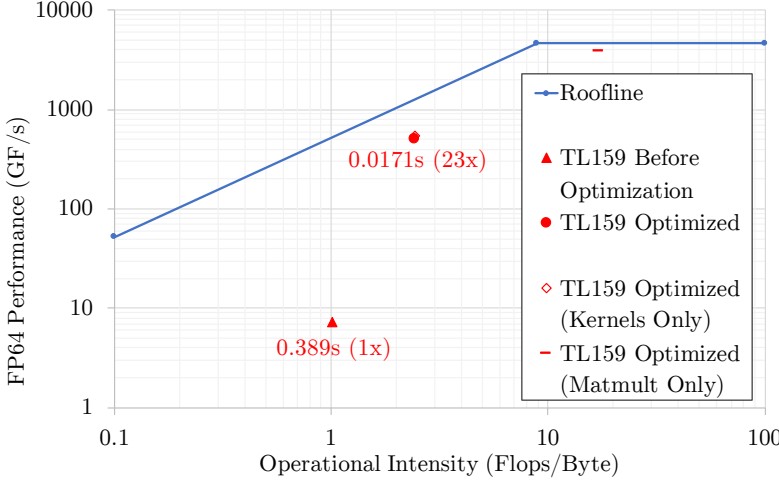

**Figure 8.** Roofline plot for the spectral transform dwarf at TL159 resolution on the NVIDIA Tesla P100 GPU. The full time-step of the original code is represented by the solid triangle. The corresponding time-step for the optimised code is represented by the solid circle. Also included are partial results for kernels only (open diamond) and matrix multiplication only (open rectangle). Each point is positioned in the plot according to its operational intensity: points under the sloping region of the roofline are limited by available memory bandwidth, and points under the horizontal region are limited by peak computational performance.

Data allocation on the GPU is expensive, as is data movement between the CPU and GPU. We structured the code such that the fields stay resident on the GPU for the whole timestep loop: all allocations/frees have been moved outside the timestep loop with re-use of temporary arrays, and thus all data transfer has been minimized.

The restructured algorithm achieves an overall speedup factor of 23x compared to the initial version which also used cuBlas

and cuFFT but followed the CPU version more closely. Matrix multiplication performance is higher than the overall performance (in flops) and the operational intensity is increased into the compute-bound regime. Note that matrix multiplication is associated with $O(N^3)$ computational complexity for $O(N^2)$ memory accesses. The extra padding operations lead to larger N and therefore also to increased operational intensity. More details about the single GPU optimisations can be found in Mazauric et al. (2017b).

Going beyond a single GPU to multiple GPUs we see a massive benefit by using the modern NVLink interconnect and the recently announced NVSwitch due to the high importance of communication for the transpositions described in section 3.1.2. Each GPU features multiple ports of high-bandwidth NVlink connections, each providing 50 GB/s of bi-directional bandwidth when using the Volta GPUs. For full bandwidth connectivity when using more than 4 GPUs we use the NVSwitch interconnect on the DGX-2 server. The DGX-2 server has 16 Volta V100 GPUs: each with six 50 GB/s NVLink connections into the switch

with routing to any of the other GPUs in the system. This allows 300 GB/s communications between any pair of GPUs in the system, or equivalently 2.4 TB/s total throughput.



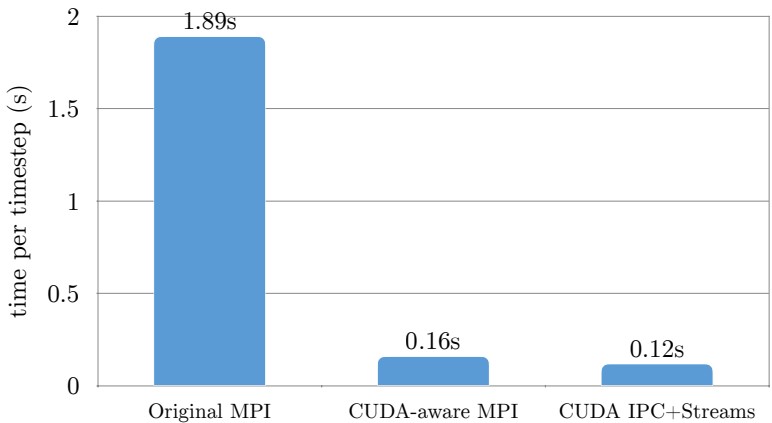

**Figure 9.** Computational performance of the spectral transform dwarf at TCo639 ( 18km ensemble member) on 4 NVIDIA V100 GPUs of the DGX1 with the original MPI implementation (left), CUDA-aware MPI communication (middle) and NVLink optimised communication (right).

When running a single application across multiple GPUs, it is necessary to transfer data between the distinct memory spaces. Traditionally, such transfers needed to be realised via host memory and required the participation of the host CPU. Not only did this introduce additional latency, but also limited the overall bandwidth to the bandwidth offered by the PCIe bus connecting CPU and GPUs. However, modern MPI implementations are CUDA-aware. This means that pointers to GPU memory can

be passed directly into the MPI calls, avoiding unnecessary transfers (both in the application and in the underlying MPI implementation). This is particularly useful when using a server that features high-bandwidth NVLink connections between GPUs, in which case CUDA-aware MPI will use these links automatically. Moving our dwarf to CUDA-aware MPI gave us a speedup of 12x (Figure 9). However, even with this optimisation the all-to-all operations remained inefficient because communication between different GPUs was not exchanged concurrently. Perfect overlap was achieved by implementing an

optimised version of the all-to-all communication phase directly in CUDA using the Inter Process Communication (IPC) API. Using memory handles, rather than pointers, CUDA IPC allows to share memory spaces between multiple processes, thus allowing one GPU to directly access memory on another GPU. This allowed another speedup of about 30% (Figure 9).

In Figure 10 we demonstrate how the use of DGX-2 with NVSwitch allows significantly better scaling than the use of DGX-1 for the Spherical Harmonics TCo639 test case. Note that we tune the number of MPI tasks in use: we use the NVIDIA

Multi Process Service to allow oversubscription of GPUs such that, e.g. the 8 GPU result on DGX-2 uses 16 MPI tasks across the 8 GPUs (i.e. 2 operating per GPU). This is because such oversubscription can sometimes be beneficial to spread out any load imbalance resulting from the spherical grid decomposition (see below) and hide latencies. We chose the best performing number of MPI tasks per GPU in each case.

As we increase the number of GPUs, the scaling on DGX-1V is limited: This is because we no longer retain full connectivity

and some messages must go through the lower-bandwidth PCIe and QPI links and/or Infiniband when scaling across multiple





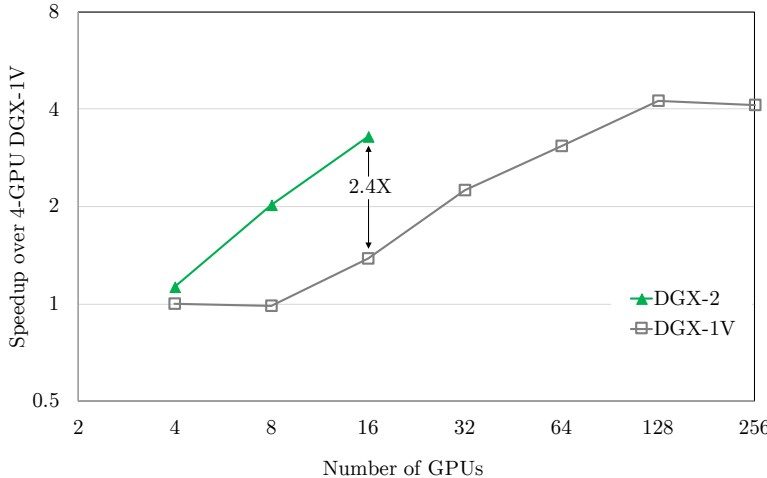

**Figure 10.** Computational performance of the spectral transform dwarf at resolution TCo639 ( 18km ensemble member) on up to 16 GPUs on one DGX-2 and up to 32 DGX-1 servers connected with Infiniband. The DGX-1V uses MPI for $\geq 8$ GPUs (due to lack of AlltoAll links), all others use CUDA IPC. DGX-2 results use pre-production hardware.

servers. But on DGX-2 with NVSwitch, all 16 GPUs have full connectivity: that is we have maximum peak bandwidth of 300 GB/s between each pair of GPUs in use. The performance is seen to scale well out to the full 16 GPUs on DGX-2, where the difference with the 16 GPU (2-server) DGX-1V result is 2.4x. It can also be seen that the speedup going from 4 to 16 GPUs on DGX-2 is 3.2x, whereas the ideal speedup would be 4x. However, initial investigations reveal that this deviation from ideal

scaling is not primarily due to communication overhead but instead to load imbalance between the MPI tasks from the spherical grid decomposition that is chosen by the application in each case, which would indicate that better scaling would be observed with a more balanced decomposition.

### 3.1.4 CPU optimisation

The spectral transform dwarf is based on the operational code used in IFS and has been continuously optimised over multiple

decades. According to profiling results, it clearly appeared that the main computational intensive kernels are the FFT and matrix multiplication executed by a dedicated highly tuned library (as Intel Mathematics Kernel Library, called MKL). In support of this work we looked into different data scope analysis tools. A comparison of the different tools is available in Mazauric et al. (2017a). The first optimisation strategy concentrated the effort on non-intrusive optimisations which have the advantage of being portable and maintainable. Among these optimisations, the use of extensions to the x86 instruction set

architecture (ISA) as SSE, AVX, AVX2, AVX-512 is interesting, because it indicates how much of the source code can be vectorised by the compiler. When the compiler failed at vectorising some loops or loop nests, a deeper investigation of how to use compiler directives followed. As the different instruction sets are not supported by all processors, the study proposed an intra node scalability comparison study among several available systems (at the time of benchmarking).





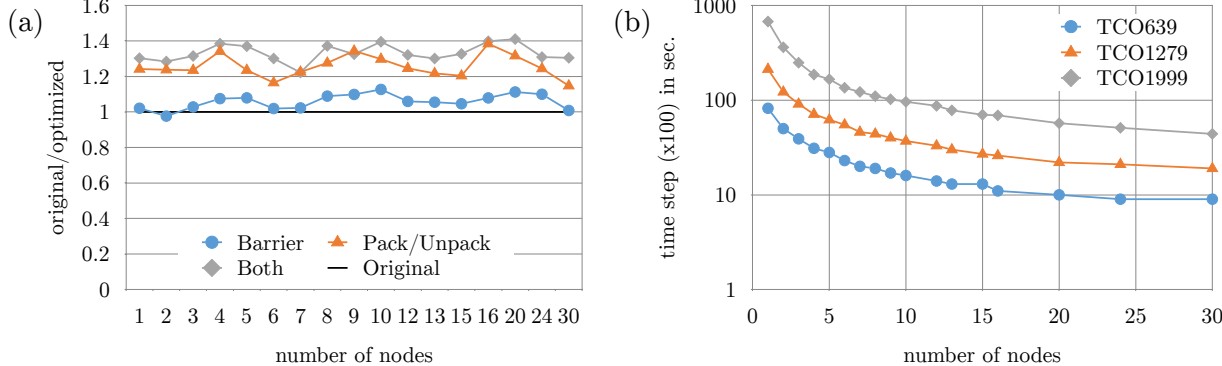

**Figure 11.** Performance measurements on up to 30 Skylake-Xeon (SKX) nodes. Subpanel (a) shows the speed-up at resolution TCo639 (18km ensemble member) with regards to different optimisations of the communication preparation phase. Subpanel (b) shows the computational performance for three different resolutions of TCo639 (18km ensemble member), TCo1279 (9km) and TCo1999 (5km) representative of current and future operational requirements. Note the use of log-scale.

System tuning using Turbo frequency (TUR), Transparent Huge Page (THP), memory allocator (MAP) can be done without modifying the source code. This exposes both performance gains and interesting information on dwarf behaviour. Indeed, enabling turbo offers a gain equal to 11%, enabling THP gives 22%, MAP 27%, and finally the best performance (35% of performance gain) is achieved by the combination of MAP and TUR. This shows that memory management is a key point. More details about the single-node CPU optimisation can be found in Mazauric et al. (2017b).

Multi-node optimisation for CPUs focused on improving the MPI communication. The largest potential for optimisations was found to be in the preparation phase of point-to-point communications. During the preparation phase the sender side gathers the local data into a contiguous buffer (Pack operation) and hands it off to the MPI library. On the receiver side, data is then scattered from a contiguous user buffer to its correct location (Unpack operation). Pack and Unpack are nearly inevitable with scattered data because Remote Direct Memory Access (RDMA) with no gather-scatter operations are known to be often less effective, notably due to the memory pinning latency (Wu et al., 2004). It also means that sender and receiver must share their memory layout as they may differ. In the spectral transform dwarf the Pack and Unpack algorithms were scanning memory multiple times. We reduced this with a global performance gain on the whole dwarf of about 20% (Figure 11a). The computational performance up to 30 nodes on Skylake is shown in Figure 11b. This optimisation can be immediately applied to the operational model IFS due to its non-intrusiveness.

Applying some of the optimisations found in restructuring the code for the GPUs to the CPU version requires some more fundamental changes which are more difficult to apply in the fully parallelised version of the dwarf. We will continue to work on applying them to the parallel version of the dwarf. As a first step towards this goal we used them in a newly developed serial spectral transform inside Atlas which will soon be used operationally for post-processing purposes. Post-processing is run in serial mode due to the large number of concurrent post-processing jobs. Compared to the current operational serial transform



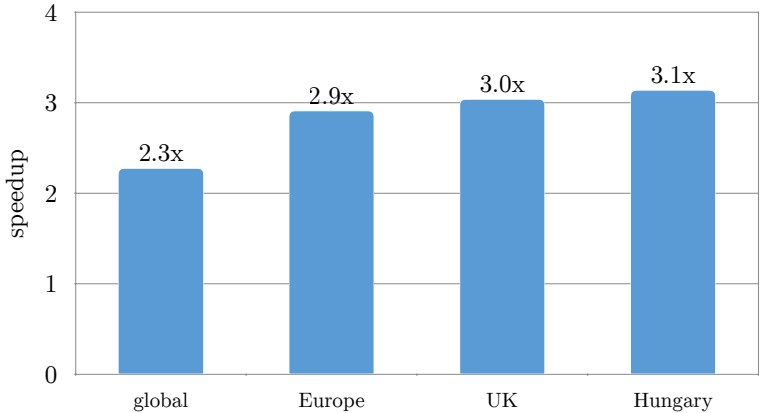

**Figure 12.** Speedup of the spectral transform by porting optimisations introduced in the GPU version back to the CPUs. The base line for this comparison is the current operational post-processing library used at ECMWF. This version of the spectral transform allows the computation on limited area domains. The speedup is given here for the global transform and three examples of limited domains (Europe, UK and Hungary).

.

used for post-processing we find a speedup of about 3x (Figure 12). Most of the speedup seems to be due to avoiding some of the temporary arrays and transpositions of temporary matrices.

### 3.1.5 Optical processors

Optalysys have been investigating an optical implementation of the spectral transform dwarf (biFFT for limited area models as well as spherical harmonics for global models). The fundamental idea behind optical processors is to encode information into a laser beam by adjusting the magnitude and phase in each point of the beam. This information becomes the Fourier transform of the initial information in the focal plane of a lens. The information can be encoded into the optical beam by using spatial light modulators (SLMs) as illustrated in Figure 13. The result of the Fourier transform can be recorded by placing a camera in the focal plane of a lens. A photo of an early prototype is shown in Figure 14.

SLMs are optical devices with an array of pixels which can modulate an optical field as it propagates through (or is reflected by) the device. These pixels can modulate the phase (essentially applying a variable optical delay) or polarisation state of the light. Often they modulate a combination of the two. When combined with polarisers, this polarisation modulation can be converted into an amplitude modulation. Hence the modulation capability of a given SLM as a function of 'grey level' can be expressed by a complex vector, which describes an operating curve on the complex plane. Each pixel of the SLM is generally a 1-parameter device; arbitrary complex modulation is not offered by the SLM, only some sub-set. This is one of the key issues with regards to exploiting the optical Fourier transform.



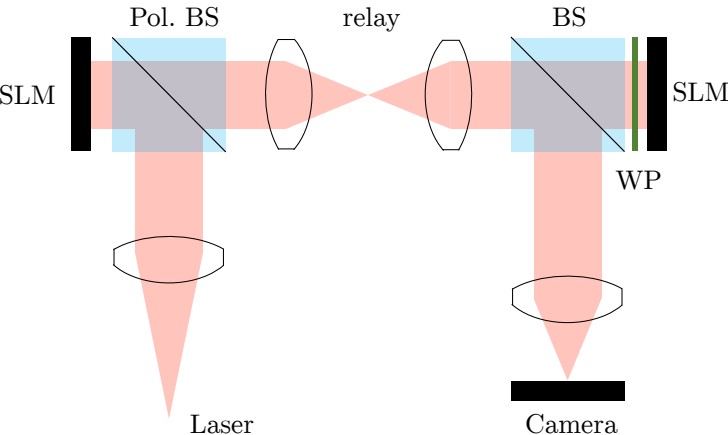

**Figure 13.** Illustration of the fundamental idea behind the optical processor. The laser beam is emitted on the bottom left. Two spatial light modulators (SLM) are used together to input the complex function. The system uses beamsplitters (BS) and an optical relay to image one reflective SLM onto another, followed by a lens assembly which approximates an ideal thin lens and renders the optical Fourier transform on a camera sensor. The half-waveplate (WP) before the second SLM is used to rotate linearly polarised light onto the axis of SLM action (the direction in which the refractive index switches), thus causing it to act as a phase modulator.

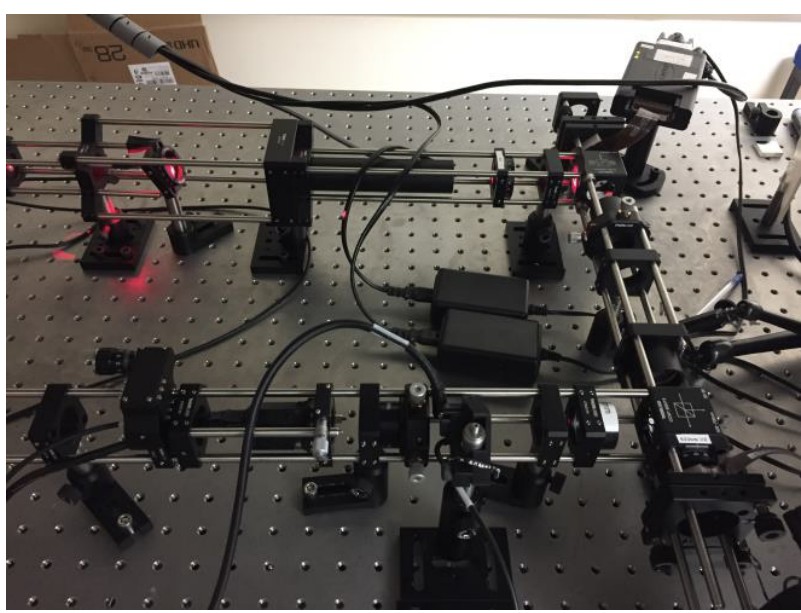

**Figure 14.** Photo of the first prototype of the optical processor. The final product is built into an enclosure of similar size like a GPU.



Sensor arrays - essentially common camera sensors - are used to digitise the optical field. They are in general sensitive to the intensity of the light, which is the magnitude of the amplitude squared. This poses a difficulty to sensitively measuring the amplitude. Moreover, they are not sensitive to optical phase. We overcome this with a perturbative method which determines the (constrained) optical phase from intensity-only measurements.

5 Each pixel of the SLM is addressable with an 8-bit value (256 levels). The SLM is not capable of independently modulating the magnitude and phase of the optical field. In the Optalysys processing system, the SLMs are configured to modulate both the amplitude and phase in a coupled manner, such that optimal correlation performance is achieved. The optical Fourier transform and all of the functions are inherently two dimensional. The propagating light beam can be thought of as a 2D function propagating and transforming along a third direction. The system is most naturally applied to 2D datasets, and many 10 problems can be mapped to an appropriate representation.

A critical aspect to realizing the potential of optical processing systems is the interface to a traditional computing platform. Bridging this gap has been a significant undertaking for Optalysys, and has resulted in the development of a custom PCIe drive board. This board interfaces to a host machine over PCIe and has direct memory access (DMA) to the system memory (RAM). It provides an interface to 4 SLMs and 2 cameras. The cameras are 4K (4096x3072). Initially, they operate at 100 Hz, but a 15 future firmware upgrade will unlock 300 Hz operation, and 600 Hz half-frame operation, dramatically increasing the potential data throughput.

There are currently two options for the SLMs. One option is using high speed binary SLMs which operate at 2.4kHz. This offers correlation at binary precision. The second option is greyscale SLMs which operate at 120 Hz. This is currently the only option to reach more than binary precision. The performance of the entire processor is determined by the part with the lowest 20 frequency. The main bottleneck with multiple bit precision is the operating frequency of the greyscale SLM. There is currently no easy solution to increase the frequency of greyscale SLMs.

Optical processing is more appropriately applied to cases where high-throughput relatively-complex operations are the priority, with less of an emphasis on numerical precision. The inherent ability of optical correlators to rapidly process convolutions naturally leads to the formation of convolution neural nets and machine learning technologies. More details about the Optalysys 25 optical processor have been published in Macfaden et al. (2017) and Mazauric et al. (2017b).

### 3.1.6 Comparison between processors in terms of runtime and energy consumption

As explained in Section 2.3 we will loose a lot of the speedup achieved by running the spectral transform on accelerators if the CPUs are idle during the computation on the accelerators. Also we need to take the cost of data transfer between CPU and accelerator into account which has not been included in the speedup numbers in this section. To take full advantage of the 30 NVLink and NVSwitch we would need to run the entire simulation on a single node which requires at the currently operational resolutions more work on optimising the memory footprint of the model.

For the CPUs and GPUs used in this paper the overall cost is dominated by the cost of the hardware and therefore by the number of sockets/devices required to reach the desired runtime (see also Section 2.3 and the Appendix A). In addition to the number of devices we also compare the energy consumption. The large number of zero operations caused in the optimised



GPU version by the padding of the matrices in the Legendre transform makes it impossible to do a fair comparison between CPU and GPU by comparing metrics based on floating point operations including comparing roofline plots.

In the full operational model the TCo639 ensemble member using 30 nodes on the Cray XC40 takes about 1.4s per time-step and the spectral transform component is about 15 percent (0.21s). Measurements with the dwarf on the Cray XC40 at resolution TCo639 resulted in 4.35s per timestep on a single node (4 MPI tasks, 18 threads per task), and 1.77s on 2 nodes. The energy consumption was measured at around 0.3Wh on the Cray XC40, which compares to 0.026 Wh measured on 4 V100 GPUs on a DGX1 which take 0.12s per time-step. The energy measurement on the Cray is based on Cray power management counters. The measurement on the V100 GPUs uses the nvidia-smi monitoring tool.

Tests in ESCAPE on the latest generation of Intel Skylake CPUs have shown 0.12s per timestep using 13 SKX nodes (connected via a fat-tree EDR network) as shown in Figure 11. This parallel CPU version has not seen the more radical changes which have been used in redesigning the algorithm for the parallel GPU and serial CPU version. There might still be potential for more substantial optimisations in the parallel CPU version which we will explore in future research.

A comparison between CPUs and optical processor with greyscale SLM is shown in Figure 15. The energy consumption of the optical processor is much lower than for the CPU. The runtime of the optical processor is larger due to the relatively slow performance of the greyscale SLM which is currently necessary to reach the precision necessary for NWP applications. More details about the comparison between CPU and optical processor can be found in Van Bever et al. (2018a).

## 3.2 Elliptic solver

### 3.2.1 Background

The dwarf originates from the elliptic solver used in the semi-implicit time integration of IFS-FVM (Smolarkiewicz et al., 2016; Kühnlein et al., 2018). We employ the Generalised Conjugate Residual (GCR, Eisenstat et al., 1983) approach to solve the following linear elliptic problem

$$\mathcal{L}(\psi) = \sum_{I=1}^{3} \frac{\partial}{\partial x^I} \left( \sum_{J=1}^{3} C^{IJ} \frac{\partial \psi}{\partial x^J} + D^I \psi \right) - A\psi = Q, \tag{4}$$

with variable coefficients $A, C^{IJ}, D^I$ and rhs $Q$, assuming either periodic (Szmelter and Smolarkiewicz, 2010) or Neumann boundary conditions. This dwarf intends to explore preconditioning strategies, where (4) is augmented to $\mathcal{P}^{-1}[\mathcal{L}(\psi) - Q] = 0$, with $\mathcal{P}$ a linear preconditioning operator that approximates $\mathcal{L}$ but is easier to invert. Given a suitable preconditioner, this auxiliary problem can converge faster due to a closer clustering of the eigenvalues with the superposition of $\mathcal{P}^{-1}$ and $\mathcal{L}$ (Thomas et al., 2003). More details about the elliptic solver dwarf can be found in Mengaldo (2016).

### 3.2.2 Multigrid Preconditioner

A challenge for the elliptic solver is to find an effective preconditioner $\mathcal{P}$ which is a good approximation to the linear operator $\mathcal{L}$ and is more economical to solve. The inversion of $\mathcal{P}$ can be simplified by two adaptations of $\mathcal{L}$; firstly by taking the matrix $C$ in (4) to be diagonal and secondly by separating the inversion of the unstructured horizontal and structured vertical components





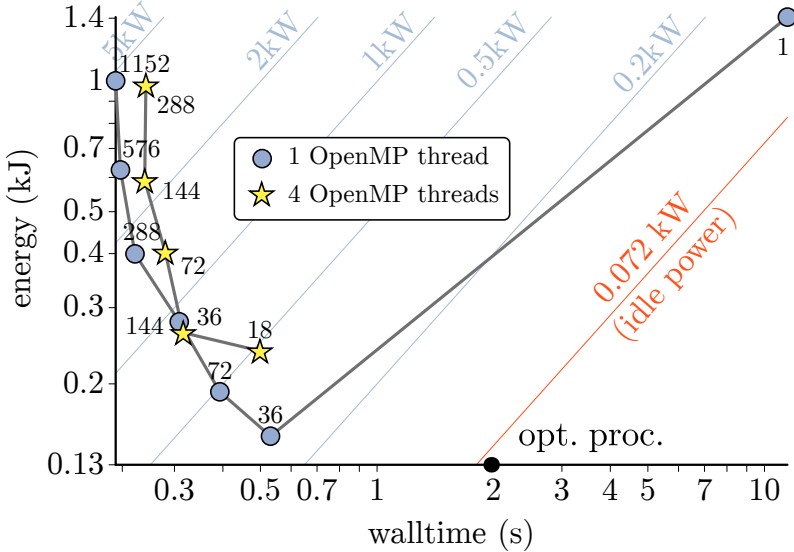

**Figure 15.** Log-log plot of the energy consumption vs. wall-clock time for the BiFFT dwarf and corresponding to the combination of one direct and one inverse transformation for 525 fields. Each data point is the result of averaging the outcome of two separate runs. Grey lines connect runs with the same number of OpenMP threads (1 resp. 4). Added are lines of constant power (light blue lines), including the power delivered by a node in the idle state (orange line). Indices next to each data point denote the number of MPI tasks. The black dot represents the estimate of the Optalysys optical processor when using a greyscale SLM. The performance of the optical processor at binary precision is much higher (not shown).

of $\mathcal{P}$—mirroring the natural anisotropy of the terrestrial atmosphere. Solutions to $\mathcal{P}$ are sought using iterative procedures akin to those in appendix B of Thomas et al. (2003). Such solvers are quick to damp errors associated with length scales of the underlying discrete meshes on which the computations are performed, but can be slower damping errors associated with longer length scales of the horizontal domain. The basic idea behind the multigrid preconditioner is to use a nested tower of grids

of varying resolution to quickly eliminate solution errors associated with those grids. Each subsequent coarser grid evaluation iteratively solves the residual problem of the previous finer grid level, which provides an error correction to the finer grid solution in a V-cycle configuration (Figure 16).

To solve the problem at a coarser resolution the field needs to be restricted to the coarser mesh. Each coarsening step incorporates a smoothing step, which minimises the solution errors associated with the given resolution. The coarsest grid

correction is usually found using some sort of direct solve technique which may be too costly to perform at finer resolutions. This solver may or may not match the smoothing method employed on the intermediate grids. After obtaining an estimate on the coarsest mesh, the errors are interpolated back to the finest mesh via each intermediate mesh. The coarser grid error is used to correct the solution error on a finer grid at each stage. This may require an additional smoothing step to be performed if the coarser grid correction reintroduces errors associated with the finer resolution. The whole cycle might employ some form of

smoother/solver to both initialise and finalise the solution. The preconditioner utilises either a vertically implicit Richardson



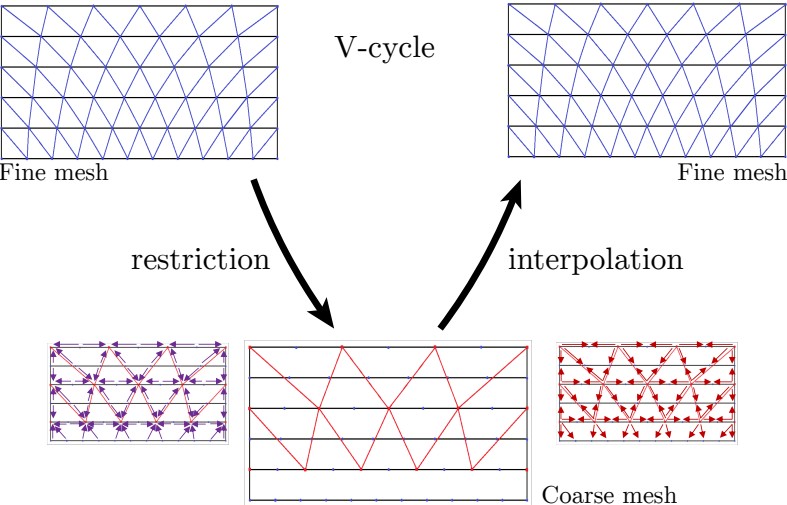

**Figure 16.** Restriction and interpolation arrangements on the octahedral mesh for the multigrid elliptic solver dwarf.

| grid | O180 | O360 | O720 |
|---|---|---|---|
| resolution | 61km | 31km | 16km |
| iterations without multigrid | 11 | 15 | 26 |
| iterations with multigrid | 6 | 7 | 11 |

**Table 2.** Number of iterations at three different resolutions for the baroclinic instability test case.

or weighted line Jacobi method as both smoother and solver for the horizontal inversion of the preconditioner operator. The vertical operator is inverted using a bespoke tridiagonal solver, since the vertical component has a structured data arrangement which is always local to the same process.

5      The employed mesh coarsening strategies utilise the Atlas framework for mesh generation and create a tower of nested octahedral meshes, where all nodes on the coarser grids correspond to nodes on the finer grids. The nested mesh structure maintains the parallel decomposition.

     First results for the baroclinic instability benchmark show a significant reduction in the number of iterations of the elliptic solver (Table 2). Work on optimising the multigrid preconditioner is still in progress. Performance measurements will therefore follow in a later publication. More information about the multigrid preconditioner can be found in Müller et al. (2017).

10      Furthermore, the unstructured-mesh weighted line Jacobi method developed as a smoother for multigrid can be employed as a single level preconditioner, with a significantly improved convergence of GCR compared to the Richardson preconditioner; see also Kühnlein et al. (2018) for discussion.



### 3.3 HEVI time-integration

As part of the ESCAPE project a review paper of different time discretisation strategies for NWP applications has been published in Mengaldo et al. (2018) since time integration affects cost considerably. In this review we found that for hydrostatic models like the currently operational IFS using semi-Lagrangian advection with semi-implicit time-integration is extremely

well suited and very difficult to beat due to its superior performance. At some point in the future we expect to reach resolutions at which we will need to use Eulerian-based time-integration (EBTI), like for example horizontally explicit vertically implicit (HEVI) schemes. As an example of this class of time integration methods we worked on Runge-Kutta implicit-explicit (RK-IMEX) methods. The class of RK-IMEX time discretization schemes may be defined as follows. First a partitioning of the right hand side (RHS) terms of the system to be solved is introduced through

$$\partial_t X = \mathcal{E} + \mathcal{I}, \tag{5}$$

where the term $\mathcal{E}$ denotes the part of the system RHS to be treated explicitly, and $\mathcal{I}$ the part of the system RHS to be treated implicitly. Then, two different multi-stage RK schemes are respectively applied to $\mathcal{E}$ and $\mathcal{I}$ parts. The RK scheme applied to $\mathcal{E}$ is purely explicit whereas that applied to $\mathcal{I}$ allows implicit evaluations at each sub-stage. The result may be written under the general form

$$\frac{X^{(j)} - X^0}{\Delta t} = \sum_{i=1}^{j-1} \widetilde{a}_{ji} \mathcal{E}\left[X^{(i)}\right] + \sum_{i=1}^{j} a_{ji} \mathcal{I}\left[X^{(i)}\right], \tag{6}$$

$$\frac{X^+ - X^0}{\Delta t} = \sum_{j=1}^{\nu} \widetilde{b}_j \mathcal{E}\left[X^{(j)}\right] + \sum_{j=1}^{\nu} b_j \mathcal{I}\left[X^{(j)}\right], \tag{7}$$

where $\nu \geq 2$ is the total number of sub-stages of the RK-IMEX scheme, $i, j$ are integer indexes such as $1 \leq i \leq j \leq \nu$, $X^{(j)}$ denotes the value of the state variable at the $j$-th sub-stage, and the superscripts "0" and "+" correspond to the values of the state variable at times $t$ and $t + \Delta t$, respectively. Notations like $\mathcal{E}\left[X^{(i)}\right]$ indicate that the terms of the sub-system $\mathcal{E}$ are evaluated

using the state variable at sub-stage $X^{(i)}$.

The coefficients $\mathcal{A} = (a_{ji})$, $\widetilde{\mathcal{A}} = (\widetilde{a}_{ji})$ for $(i,j) \in [1, \nu] \times [1, \nu]$, and the weight-vectors $(b_j, c_j = \sum_{i=1}^{j} a_{ji})$ and $(\widetilde{b}_j, \widetilde{c}_j = \sum_{i=1}^{j} \widetilde{a}_{ji})$ for $j \in [1, \nu]$ may be classically represented by a double Butcher tableau:

$$
\begin{array}{c|c}
\widetilde{c} & \widetilde{\mathcal{A}} \\
\hline
& \widetilde{b}^{\,\mathrm{T}}
\end{array}
\qquad
\begin{array}{c|c}
c & \mathcal{A} \\
\hline
& b^{\mathrm{T}}
\end{array}
$$

The first Butcher tableau defined by $(\widetilde{\mathcal{A}}, \widetilde{b}, \widetilde{c})$ describes the explicit part so that $\widetilde{a}_{ij} = 0$ for $i \geq j$, and the second one $(\mathcal{A}, b, c)$

corresponds to the implicit part of the scheme. RK-IMEX schemes, using such a double Butcher tableau, are traditionally labelled in literature with the nomenclature [NAME]$k(s, \sigma, p)$, where $k$ denotes the order of accuracy of the explicit part, $s$, the number of implicit inversions to be performed in the implicit part (i.e. the number of non-zero diagonal coefficients in $\mathcal{A}$),



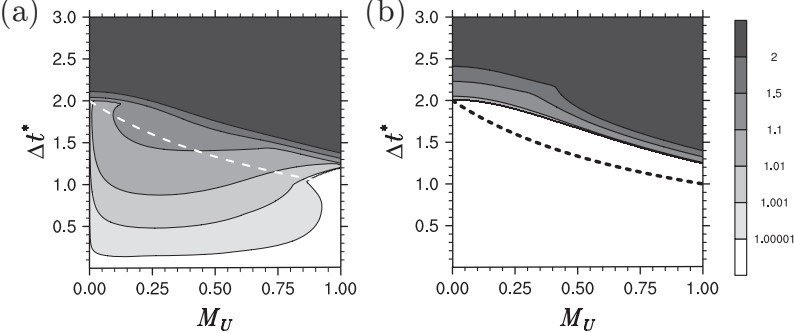

**Figure 17.** Stability region of (a) common Trap2 time integration scheme and (b) enhanced TRAP2 time integration scheme (Colavolpe et al., 2017). The vertical axis of the stability graphs show the time-step while the horizontal axis represents the advection velocity. Values larger than 1 mean that the scheme is unstable. As we can see the new scheme provides a significantly improved stability region.

$\sigma$, the storage factor (i.e. the minimal number of explicit sub-stages that need to be stored to complete the time-step), and $p$, the overall order of accuracy of the scheme. The particular RK-IMEX schemes that have been analysed in this work, are the so-called UJ3(1,3,2), ARK2(2,3,2), and Trap2(2,3,2) scheme for the finite-difference (FD) or finite-volume (FV) discretisation and additionally SSP3(3,3,2) and SSP3(4,3,3) for a Discontinuous Galerkin (DG) method, according to this nomenclature. The

double Butcher tableau for each of these schemes is given in Lock et al. (2014).

The most suitable RK-IMEX HEVI scheme for time-critical NWP applications is the one that achieves the best balance between low computational cost and good stability property. In this prospect, the RK-IMEX time-schemes UJ3, ARK2, and TRAP2, identified in the literature as promising time-discretisation in HEVI context, have been analysed in more detail taking into account the stability effects of the advection processes and orographic forcing terms introduced by the use of terrain-

following coordinate. Drawing from all these analyses, it firstly appears that RK-IMEX HEVI approaches based on only two Butcher tableaux might not be optimal for dealing with the different dynamical processes involved in a fully compressible system, and their multiple interactions, and notably the feedback between the horizontal divergence and pressure gradient terms. Secondly, in presence of steep sloped orography, the use of the covariant horizontal wind component as a prognostic variable has been found to be beneficial for the stability of all examined RK-IMEX HEVI schemes. Exploiting all these results,

a optimal HEVI scheme, termed as TRAP2 "covariant-implicit", has been designed. This newly proposed scheme offers the largest stability limit while being as cheap as the original TRAP2 scheme in terms of computational cost. As a proof-of-concept, the suitability of this scheme has been confirmed experimentally in a complete fully-compressible vertical plane model. Figure 17 shows the improvement of the stability region when compared with the traditional scheme. More details of this work have been published in Colavolpe et al. (2017). A HEVI formulation of the general DG approach has been implemented, too.

Problems occuring with the splitting of the numerical flux and the formulation of proper boundary conditions have been solved for simple test problems as the linear advection and the linear wave equation. For the more upwind biased numerical fluxes used in DG, the use of the SSP3-schemes seems to be more appropriate.





### 3.4 Semi-Lagrangian advection

#### 3.4.1 Background

We consider a three-dimensional semi-Lagrangian advection scheme. Advantages of semi-Lagrangian schemes are stable and accurate integration with long time steps and efficiency for multiple tracers. The semi-Lagrangian transport scheme solves the

following evolution equation

$$
\frac{D\psi}{Dt} = 0 \qquad \text{with} \qquad \frac{D}{Dt} = \frac{\partial}{\partial t} + \mathbf{v} \cdot \nabla \,, \tag{8}
$$

where $\psi$ is a scalar field advected by the wind $\mathbf{v} = (u, v, w)$.

To solve Eq. (8) we integrate along the trajectory of a fluid parcel in the time interval $[t, t + \Delta t]$ which yields $\psi_a^{t+\Delta t} = \psi_d^t$. The subscripts $a$ and $d$ denote the arrival and departure points of the flow trajectory, respectively. The arrival point is the location

of a parcel at time $t + \Delta t$ and coincides with a grid-point. The departure point represents the parcel's location at time $t$, and typically lies somewhere in the space between grid-points. Hence, the departure point has to be found. The semi-Lagrangian scheme solves for each arrival point with coordinates r the trajectory equation

$$
\frac{D\mathbf{r}}{Dt} = \mathbf{v} \tag{9}
$$

to determine the departure points $\mathbf{r}_d$, and thereafter remaps the field $\psi$ to the set of these points. In the ESCAPE project, we

implemented a prototype for the semi-Lagrangian advection using Atlas. To solve the trajectory equation an iterative method based on the second-order mid-point integration scheme is used (Hortal, 1999; Temperton et al., 2001). For the interpolation, the LAITRI (LAgrangian Interpolation TRIlinear) procedure of IFS is adopted. More details about these two dwarfs can be found in Mengaldo (2016).

#### 3.4.2 Optimisation for CPUs and Xeon Phi

Work on optimising semi-Lagrangian advection focused on optimising the LAITRI dwarf. LAITRI is a heavily-used subroutine in the European IFS global weather modelling system. It accounts for about 4% of the total runtime of IFS. The work on optimising this dwarf was done at the beginning of the ESCAPE project when Knights Landing (KNL) was new, and the interesting question was how KNL would compare with the older Ivy Bridge CPU and Knights Corner (KNC) accelerators. Strong-scaling studies of the LAITRI dwarf over increasing OpenMP thread count have shown that best performance is achieved on the Ivy

Bridge node by using 40 OpenMP threads and 60 threads on KNC and KNL. We compare the best time-to-solution for different hardware settings, and with thread count being maintained as constant across each platform, at the optimum point of the scaling plots. We experimented with the following settings: without automatic vectorisation (by compiling with the -no-vec switch in ifort), with vectorisation, and with data alignment at 64 byte boundaries. The runtime measurements are shown in Figure 18. The results show that improvements in time-to-solution are incremental for any given optimization on a fixed platform,

with no particular setting giving considerable speedup compared to others. However, there is a marked performance boost on the KNL platform over both KNC and the Intel Ivy Bridge Xeon processors tested. This is a turnaround from previous work





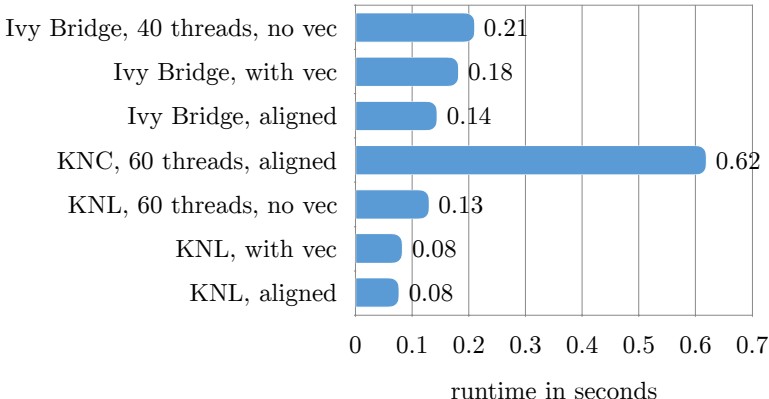

**Figure 18.** Time to solution of LAITRI dwarf for Ivy Bridge, Knights Corner and Knights Landing processors.

comparing KNC to Ivy Bridge, where it was found (with similar results elsewhere in HPC) that the KNC performance was disappointing. Not only does KNL surpass KNC (by an order of magnitude in our tests), it also outperforms Ivy Bridge by 2 to 1 in time to solution. The settings to achieve this are straightforward, involving 3 main ideas - correct use of OpenMP (e.g. ensuring correct 'first touch' of data), suitable compiler data alignment directives to facilitate good vectorisation performance, and prudent setting of runtime variables such as KMP AFFINITY and KMP HW SUBSET. More details about this work can be found in Robinson et al. (2016).

### 3.5 MPDATA advection

#### 3.5.1 Background

MPDATA refers to a class of non-oscillatory forward-in-time high-resolution numerical schemes for the advective terms in flux-form formulations of fluid equations. In contrast to the semi-Lagrangian approach based on Eq. (8), MPDATA solves the advective transport problem in the form of an Eulerian conservation law

$$\frac{\partial \rho \psi}{\partial t} + \nabla \cdot (\mathbf{v} \rho \psi) = 0 , \tag{10}$$

where $\rho$ represents the fluid density. MPDATA schemes are at the basis of the Finite-Volume Module of IFS (henceforth IFS-FVM), a novel dynamical core formulation under development at ECMWF (Smolarkiewicz et al., 2016; Kühnlein et al., 2018). The finite-volume MPDATA implementation in IFS-FVM is described in Kühnlein and Smolarkiewicz (2017), including a comprehensive list of references on MPDATA methods and their broad applicability.

The basic principle of MPDATA is most suitably described as an iterated upwind (alias donor cell) scheme: The initial iteration represents the first-order accurate upwind scheme with the advective velocity given by the physical flow. Subsequent iterations are also based on the upwind scheme, but the updated field is advected with a properly defined pseudo-velocity designed to compensate to selected order (typically second) the spatial and temporal truncation errors of the previous iteration.



The resulting scheme is at least second-order accurate in time and space, fully multidimensional and conservative. Due to the consistent application of the upwind differencing, MPDATA retains the characteristic features of a relatively small phase error and strict sign preservation of the transported field. Various extensions of the basic MPDATA scheme are available, such as for the incorporation of arbitrary right-hand-sides, the transport of fields with variable sign, and the nonoscillatory option

that ensures solution monotonicity. Moreover, structured-grid flux-form finite-difference and unstructured-mesh finite-volume formulations of MPDATA exist, see Smolarkiewicz and Margolin (1998); Smolarkiewicz and Szmelter (2005); Kühnlein and Smolarkiewicz (2017). More details about the dwarf can be found in Müller et al. (2017).

### 3.5.2   CPU optimisation

Profiling with a solid body rotation on a O1024 mesh indicates that for a large number of computing nodes most of the runtime
is spent in MPI communication. For this reason we focused our CPU optimisation on optimising MPI communication.

The MPI library, especially the Intel MPI one, relies on several interconnection protocols. It has to be wisely chosen according to the cluster hardware. In the benchmark cluster the interconnect network is based on the Infiniband protocol. Traditional Infiniband support uses the Reliable Connection (RC) protocol to exchange MPI messages, but the User Datagram (UD) protocol has emerged as a lower memory consumption, more scalable alternative. The first optimization has been to enable this
protocol in the Intel MPI library.

A second optimization is the replacement of the "manual" implementation of the AlltoAllV algorithm inside the halo_exchange subroutine by the MPI_AllToAllV function. To enhance the scalability, one may consider overlapping the MPI communications with computation loops. This has been introduced by two variants based on a single core idea: the computation loops have to be evaluated on the data to be exchanged before sending them and then they are evaluated on the private data while the
communications are received. The first one is implemented via a mask array separating the shared versus the private data. The second one is implemented by an indirection array storing the shared (resp. the private) indices of the data.

The modified versions have been launched on a cluster with nodes based on Skylake processors (16 cores, 192GB) interconnected by an Infiniband EDR network. Figure 19 shows scalability results of multi-node optimizations on a 16 physical cores SKX with 1 MPI task per socket and 1 OpenMP thread per physical core. The two combined optimizations (UD Infiniband
protocol and AllToAllV) show a speedup of nearly 15% (20s vs 23s on 64cores, 12s vs 14s on 128 cores). The two variants of the async patterns are less performant than the AlltoAllV implementation. In the first variant, the loops are executed two times with a conditional inside them. This alters the performance. The second option leads to non-contiguous data which is also less efficient.

### 3.5.3   GPU optimisation

Previous work on optimising MPDATA for GPUs considered the finite-difference formulation on structured meshes (Wyrzykowski et al., 2014a, b). In the ESCAPE project, we focused on the finite-volume formulation of MPDATA which supports unstructured meshes. As a test case we use a solid body rotation over the pole on an octahedral mesh O128 with three levels in the vertical. The most computationally expensive kernel beside communication was identified as "compute_fluxzdiv" which computes the





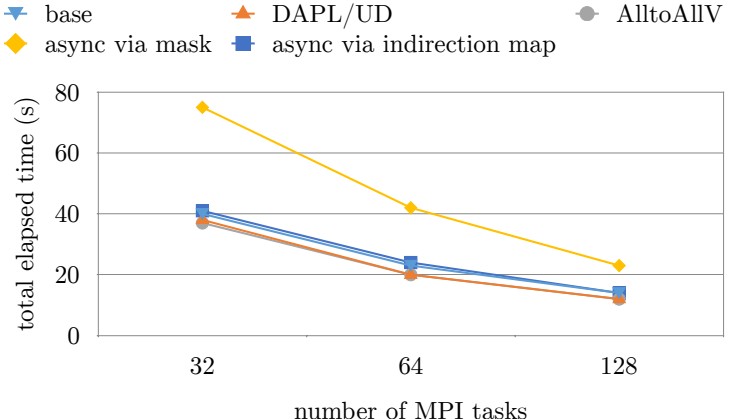

**Figure 19.** MPDATA scalability results of multi-node optimisations on a 16 physical cores SKX with 1 MPI task per socket and 1 OpenMP thread per physical core. The two combined optimisations (UD Infiniband protocol and AllToAllV) show a speedup of nearly 15% (20s vs 23s on 64cores, 12s vs 14s on 128 cores). The two variants of the async patterns are less performant than the AllToAllV implementation.

divergence. This was found through profiling on the CPU where it is responsible for 25% of overall runtime. There exist three nested loops in this kernel which must be mapped to the parallelism of the GPU. The extents of these, for the test case in use, are: L1 loop over 71424 nodes, L2 loop over 3 levels and L3 loop over 213199 edges. The resulting performance of the initial implementation on the GPU was low: it is only able to achieve 44GB/s data throughput on the NVIDIA Tesla P100 GPU, which is less than 10% of that achieved by the STREAM benchmark. The reasons for this are related to suboptimal parallel decomposition and data layout, and the accessing of data in deep structures.

There are a number of possibilities for how the parallel mapping can be implemented and the original choice was suboptimal. The reason for this is that the loop assigned to CUDA threads within each block (at L2) has an extremely small extent, where typically we need much more parallelism at the CUDA thread level to make good use of the vector nature of the CUDA execution model. Instead, we choose to collapse the two outermost loops (through use of the OpenACC parallel loop collapse(2) directive) and assign the parallelism across this collapsed loop to both CUDA blocks and threads within each block. This allows the compiler to decide a much more suitable extent of vectorisation. An OpenACC loop seq directive is applied to the innermost loop, such that each thread will perform all of this loop in a sequential manner (satisfying the requirements of the reduction). With this new parallelisation strategy, it is important to ensure that data is accessed in a coalesced manner for the different arrays, in order to achieve a high percentage of memory bandwidth. For coalescing, we need consecutive threads (corresponding to consecutive indices of the vertical level) to access consecutive memory addresses, and fortunately the original data layout of these arrays (with the level being the fastest moving innermost index in Fortran) already satisfies this requirement, so no further data layout modifications are necessary.

The kernel accesses several read-only data elements and structures. For these, best performance is achieved when the compiler maps the data to the fast on-chip constant cache on the GPU. However, we find that, for the case of the deep array access





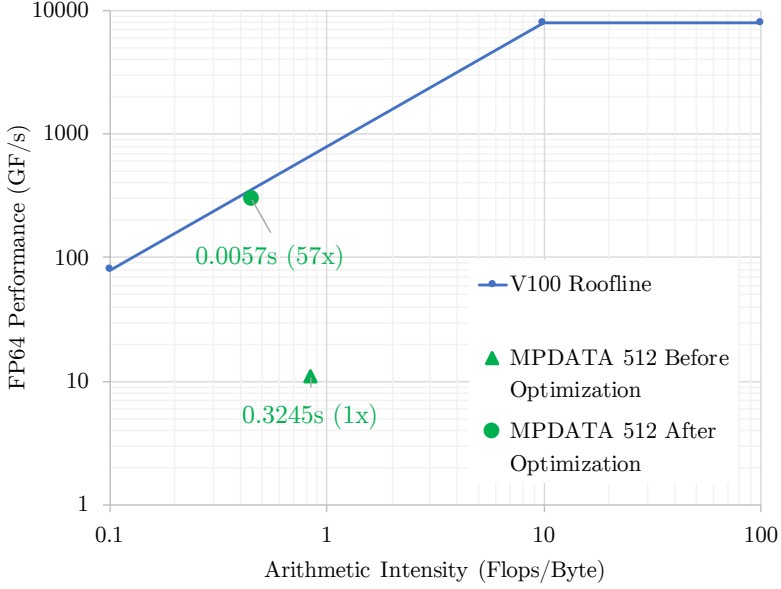

**Figure 20.** Roofline plot of the original and optimised versions of MPDATA for the solid body rotation over the pole at O512 resolution on the NVIDIA Tesla V100 GPU. The points are positioned in the plot according to their arithmetic intensity: points under the sloping region of the roofline are limited by available memory bandwidth, and points under the horizontal region are limited by peak computational performance. These results were obtained with PGI 17.9, CUDA 8.0.61 and OpenMPI 1.10.7.

like this%geom%node2edge_sign the compiler does not make full use of this capability. But, if we copy this to a regular "flat" array, ahead of kernel execution and use this in place of the original structure we see an increase in constant cache utilisation and improved performance. Furthermore, we can see that the operation involves division by a constant. We replace this by multiplication by the reciprocal of the constant (calculated in advance), which further boosts performance.

These optimisations decrease the time taken by the kernel by a factor of 9.4x. The achieved throughput of the optimised version is measured by the NVIDIA profiler to be 344GB/s, which is 66% of the value measured using STREAM benchmark, indicating that we are reasonably close to the hardware limit. More information about the optimisation of this kernel can be found in Mazauric et al. (2017b).

  Optimising all of the MPDATA kernels in a similar way gives us an overall speedup of 57x compared to the initial OpenACC

port of the CPU version which brings the dwarf close to the roofline (Figure 20). This includes optimised data management such that the fields stay resident on the GPU for the whole timestep loop: all allocations/frees have been moved outside the timestep loop with temporary work arrays being re-used, and all host/device data transfer has been minimized.

  When utilising multiple GPUs, two halo exchanges are required each timestep for each of two fields. The subroutine for the halo exchange is provided by the Atlas library, which expects the data to be resident on the host CPU. Therefore, the OpenACC

directives are required to copy the array from GPU to CPU before the operation, and back to the GPU after. As will be seen,





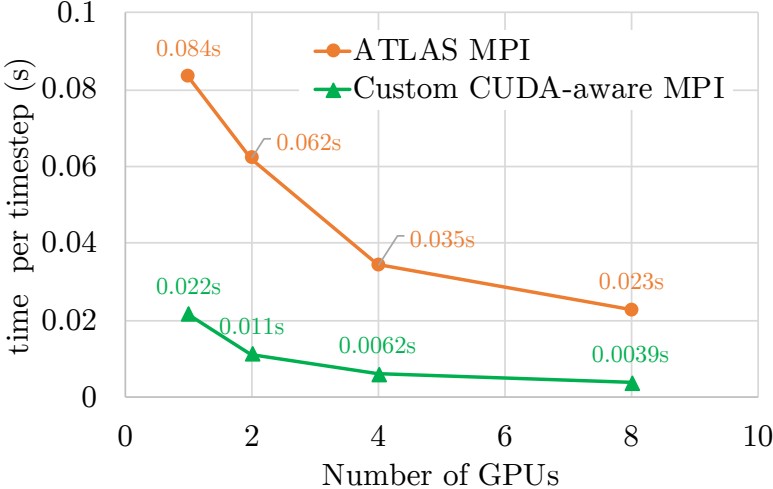

**Figure 21.** The dependence of the MPDATA time-step time, for for the solid body rotation over the pole at O1024 resolution, on the number of V100 GPUs in use within a single DGX-1V server. Orange circles denote the original halo-exchange communication mechanism which involves host-device data transfer and the host-based Atlas library. Green triangles denote use of a new custom CUDA-aware halo-exchange mechanism.

this data movement has a huge overhead. Therefore, we developed a new custom halo-exchange mechanism to be used in place of the original, which allows data to be kept resident on the GPU throughout, which we will now describe.

With structured grids, halo exchanges are relatively straightforward since the halo data on each subdomain corresponds to a small subset of "neighboring" subdomains in a clear manner. However, the data structures in this dwarf are unstructured grids, 5 and as a result each subdomain may have halo data elements corresponding to any of the other subdomains. Therefore, the halo exchange requires an "all-to-all" communication pattern, where the sizes vary. We implement this as follows:

– Pack "send" buffer containing "edge" data on each GPU (using OpenACC), for each corresponding remote GPU.

– Exchange this data using CUDA-aware MPI_AlltoAllV (like in the GPU optimised spectral transform dwarf).

– Unpack "receive" buffer on each GPU (using OpenACC), with each element being stored in its appropriate halo location.

10 Figure 21 shows performance measurements with this approach. When only a single GPU is in use, no halo exchange is required, and the runtime per time-step is measured (using the optimisations described above) to be 0.0214s. But when we enable the halo exchange (still on a single GPU), using the pre-existing communication mechanism, then host/device data transfers occur for the entire field which has a huge overhead, and the time increases by an unworkable factor of 3.9x to 0.0835s. With our new mechanism, the overhead is only around 1% with the time at 0.0216s. By keeping data resident on the 15 GPU we have removed the overhead and allowed effective multi-GPU scaling.



### 3.5.4  Domain Specific Language

The GridTools framework provides a set of tools for developing numerical methods of weather and climate applications. The emergence of new computing architectures and accelerators in the supercomputing systems where weather and climate applications are run pose a challenge to efficiently maintain and run weather models. Typically weather models are complex

systems with large codebases (from hundred thousands to millions lines of code). Differences in the computing architectures make adapting models to new architectures a daunting task. Often, different architectures offer different memory spaces that must be managed explicitly (like GPUs), efficient computations on gridded fields require storing the multidimensional fields with different memory layouts, etc. And the nested loops over dimensions and performance optimisations (such as tiling/loop blocking, loop fusion, etc.) are specific to each hardware architecture. Additionally, they might require the use of different

programming models.

The main goal of the GridTools library is to provide a solution for weather and climate models to run one code base on many different architectures (portability) and achieve good performance (performance portability). However, the main operational product of GridTools so far focused on solutions for lat-lon grid models like COSMO. The work developed in the ESCAPE project aimed at extending the DSL support for irregular grids and the efficient generation of backends for multiple

architectures.

In order to index fields in their corresponding location and to establish the connectivity with neighbors, the GridTools backend for irregular grids introduces the concept of a location type, which can be edges, cell centers and vertices for any type of elementary shape. The location type can be used in the declaration of the storages that hold fields as well as in the operators.

Another concept of the GridTools backend for irregular grids is the topology of the grids, which knows the connectivity

among the grid points in each location type for each of the supported grids.

Some irregular grids, like the octahedral/icosahedral grid will still retain a structure which allows deriving rules for extracting the connectivity of the topology without the use of unstructured meshes. Other grids, like a generic reduced Gaussian have a structure, although more complex, and do not allow to derive easy methods to extract the connectivity. The latter group will exhibit a totally irregular pattern and will require the use of an unstructured mesh.

Memory data layouts of the fields where computations operate are crucial for performance, particularly on modern accelerators. Modern CPU processors operate on large vector widths (like AVX-512) while accelerators like NVIDIA GPUs compute on warps of 32 CUDA threads. Both provide more efficient use of memory if loads and stores are performed on aligned and coalescing accesses. A coalescing access will require that the memory loads/stores of different parallel cores of a GPU warp or elements of a vector instruction in a modern CPU processor are contiguous in memory.

This is straightforward to achieve for lat-lon grids, with memory layouts organized in rows/columns for the (i,j) indexing space. On the contrary, irregular grids without any regular pattern in the grid do not allow to obtain coalescing accesses. However, many grids employed by weather and climate models are derived from platonic solids and retain their original structure. Examples are the octahedral grid (widely used in the ESCAPE dwarfs), the icosahedral grid or the cubed sphere.





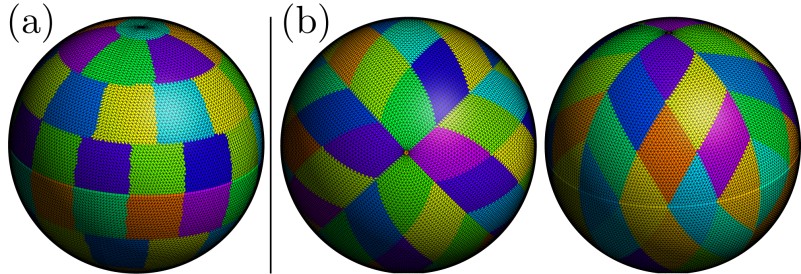

**Figure 22.** (a) Equal partitioning domain decomposition of the octahedral grid. (b) Structured domain decomposition based on parallelograms (as seen from two different angles).

In the ESCAPE project, we explored how to obtain structured domain decompositions for the octahedral grids and conducted performance comparisons of different types of indexing on accelerators, with the goal to determine basic properties and optimal memory layouts for optimal performance of the ESCAPE dwarfs. Figure 22b shows a possible domain decomposition of the octahedral grid that preserves the original structure of the octahedron.

With this domain decomposition, a new indexing method that uses colouring for cells/edges and vertices was introduced, with the important property that all the loads/stores of the computational patterns found in the dwarfs are coalescing and a large fraction of them are also aligned (Figure 23).

Figure 24 shows that the structured numbering yields best performance when combined with the indirect addressing. The DSL backend of GridTools for irregular grids developed within the ESCAPE project implements structured direct addressing

and irregular indirect addressing. Future developments will support structured indirect addressing which gives best performance on GPU accelerators for grids that contain a structure and irregular indirect addressing using a Hilbert space filling curve that provides still a good bandwidth for those grids for which a structure cannot be exploited (e.g. following coastlines).

These DSL developments have been used to implement a portable version of the MPDATA dwarf. The DSL version hides such details as the nested loops and the OpenACC directives used to specify properties of the GPU kernel and data layouts

of the FORTRAN arrays. Furthermore, the DSL allows to compose several of these operators together, which is used by the library to apply advanced performance optimisations like loop fusion or software managed caches.

Since most of the weather and climate applications are memory bandwidth bound on modern processors and accelerators, many of the performance optimisations focus on the best utilisation of the memory subsystem of the computing architectures. In order to optimise memory bound kernels, one of the most prominent optimisations is the combination of tiling and loop

fusion that increases data locality. All computing architectures offer a memory system with different levels of cache or scratch pad. Since the bandwidth of a cache level is typically orders of magnitude larger than main memory, the use of the cache of the memory system to reduce main memory accesses increases significantly the performance of the memory bound applications. Architectures like traditional CPUs or Intel XeonPhi have an automatic caching mechanism that does not require explicit instructions at the software level. However techniques like tiling or loop fusion are crucial in order to fit temporary computations

into the fastest levels of the cache system.





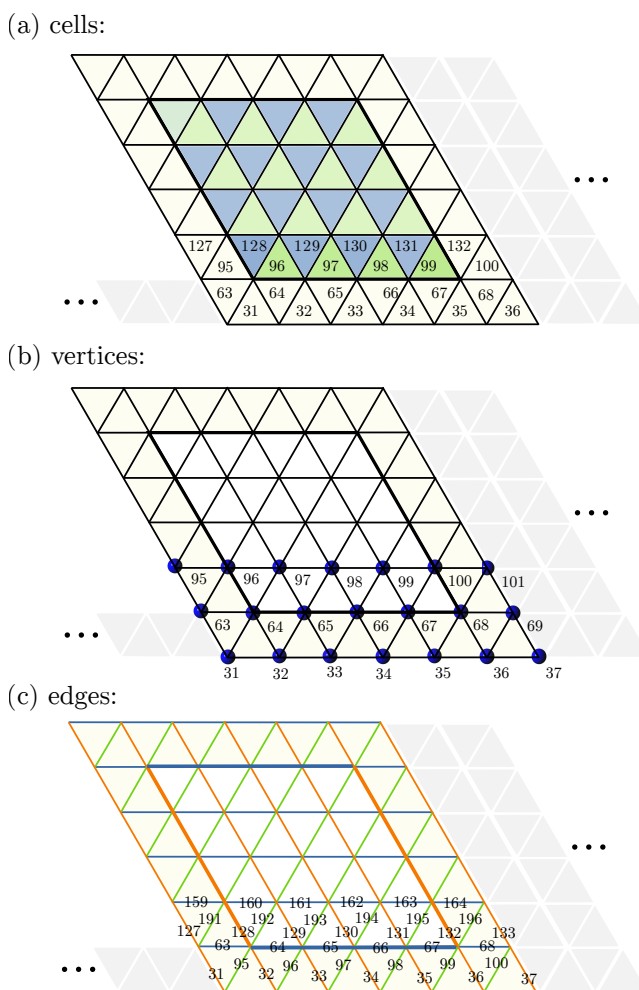

**Figure 23.** Structured indexing of (a) cells, (b) vertices and (c) edges of a parallelogram of the icosahedral/octahedral grid. Gray cells indicate padding inserted in order to align accesses to vertices within the compute domain. Number of colors is 1 for vertices, 2 for cells (downward / upward triangles) and 3 for edges. Each cell is indexed with a tuple (row, color, column).



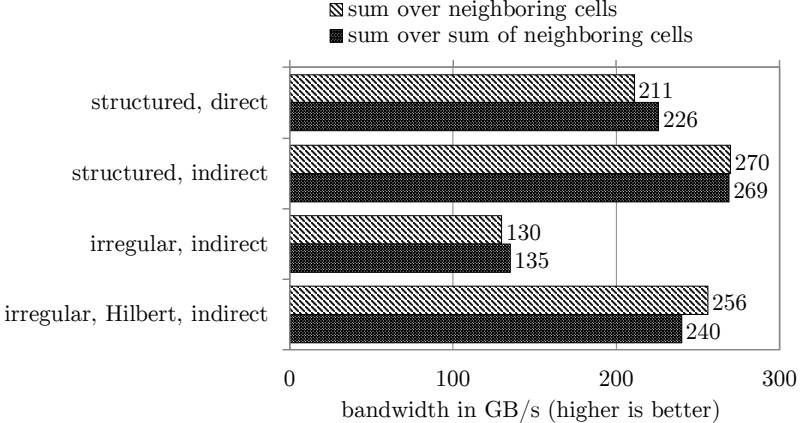

**Figure 24.** Bandwidth (required data transfer / computation runtime) in GB/s of two stencil computations for the different indexing and access methods on an octahedral patch of size 128x128x80 for a P100 GPU. These results show that the structured numbering yields best performance when combined with the indirect addressing. The DSL backend of GridTools for irregular grids developed within the ESCAPE project implements structured, direct and unstructured, indirect addressing. Future developments might support structured, indirect addressing which gives best performance on GPU accelerators for grids that contain a structure and Hilbert curve, indirect addressing that provides still a good bandwidth for those grids for which a structure cannot be exploited.

On the other hand, NVIDIA GPUs require an explicit declaration in the programming model for the use of the different levels of the cache system, like the shared memory.

The composition of stages of the DSL allows the library to apply these loop tiling and fusion. In the MPDATA example shown for the computation of the fluxes, multiple fields are reused between the different computations. The theoretical calculations give us a number of 1140638 main memory accesses without fusion and a number of 357120 memory accesses with fusion. Measurements on an Intel Haswell E5-2690 CPU demonstrate that the time per grid point update of the computation of the fluxes of the MPDATA dwarf is reduced by almost 50%. Among other optimisations, the use of a DSL allows to fuse all the stages that form a single computation of the MPDATA, using high bandwidth scratch pad for intermediate variable, which increases the data locality of the algorithm. Such optimisations can only be performed since the library assumes a parallel model that supports only specific and limited computational patterns that can be expressed by the DSL, as opposed to general purpose language compilers that cannot make such assumptions. Comparing the Fortran OpenACC kernel with the DSL version gives us a speedup of 2.1x for the DSL version. This speedup could also be achieved by hand-tuned optimisation. The DSL prevents the repeated manual effort of tuning the code for multiple architectures. At the same time the DSL allows to perform optimisations which would otherwise make the code unreadable. More details about this work including code examples on how to use the new backend to GridTools can be found in Osuna (2018).





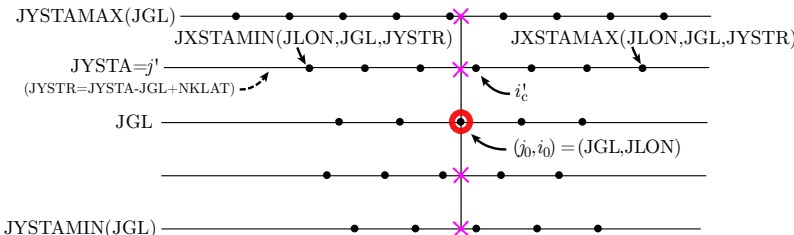

**Figure 25.** A stencil used to calculate a fourth order meridional derivative.

## 3.6 High order finite difference

As an alternative to the spectral transform and finite volume discretisation we created a global shallow-water model named GRASS (Global Reduced A-grid Spherical-coordinate System), which uses high-order finite differences while still supporting the reduced octahedral grid that is used operationally in IFS and ALARO. This means that the number of grid points decreases

with increasing latitude.

Longitudinal derivatives are computed using Lagrange Sine/Cosine representation with a one-dimensional stencil along the longitudinal row. A Fourier representation along longitudes is also possible near poles. Meridional derivatives are calculated by first interpolating values at remote latitudes along longitudes in order to get a meridionally-aligned set of data at each grid-point, as illustrated in Figure 25, and then a one-dimensional discrete meridional derivative operator is applied. The longitudinal

interpolations are made with Lagrange Sine/Cosine representation, which takes advantage of the longitudinal periodicity to give a more accurate interpolation if the stencil is spanning a large part of t he circle. The meridional derivative is calculated by applying the classical centred derivative to the interpolated values.

This approach was implemented and parallelised with MPI and OpenMP. Solid body rotation experiments were carried out to make sure there were no discretisation errors for this flow, and to show that the scheme can be stable. Drawing from these

results most favourable configurations were identified.

GRASS is specifically designed to achieve the best quality for complex and challenging flows at high resolutions on the sphere. Since the exact solutions are not known for these flows, the references are taken from simulated solutions recognised as of the best-quality in the scientific literature. For this purpose the most challenging case to date for a two-dimensional model on the sphere was chosen. The reference is the converged solution presented in Scott et al. (2015). Our results demonstrate

very good agreement with the reference solution which uses a spectral transform method (Figure 26). More details about this work will be in a pair of papers submitted for publication (Bénard and Glinton, 2019a, b).

## 4 Physics dwarfs

Physics parameterisations account for about 30% to 40% of the overall forecast runtime (compare Figure 5). Out of the large number of different parameterisations we focused on the two computationally most expensive parameterisations: a radiation



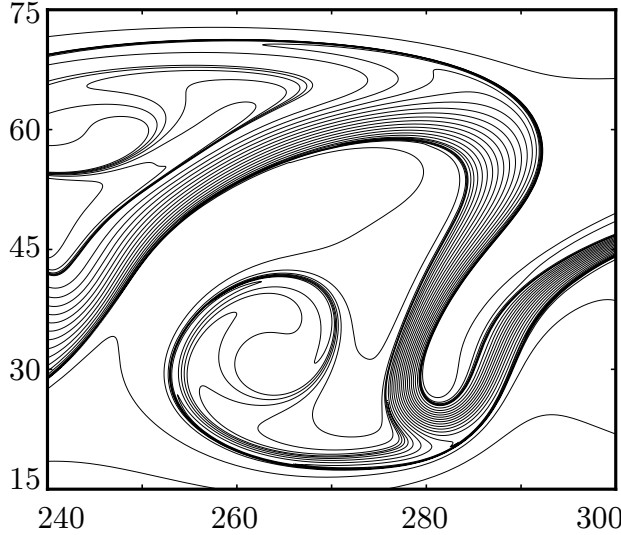

**Figure 26.** High order Finite Difference result for the shallow water test case from Scott et al. (2015) at day 6 and resolution N5761 (i.e. 5761 latitudinal rows with approximately 3.5 km horizontal grid spacing).

scheme called ACRANEB2 from ALARO and a cloud microphysics routine called CLOUDSC from IFS. Both of these two dwarfs have many features of physics modules in weather and climate models in general. Thus, they include frequent usage of transcendental functions, and complex loop and conditional structures. This makes them interesting also in a broader context.

### 4.1 Radiation scheme ACRANEB2

5 The radiation schemes in numerical weather prediction and climate models take up a considerable amount of the overall running time of these models. Here, "radiation" is implicitly taken to mean *electromagnetic radiation*. The heating due to absorption of shortwave (solar) and longwave (terrestrial heat) radiation is the initial driver of all atmospheric processes with the exception of volcanic events.

Neither the shortwave nor the longwave radiative transfer can be solved within a reasonable amount of time from basic 10 principles. In addition to the spatial and temporal approximations that are necessary to make for all physical processes in atmospheric models, it is necessary to make approximations in the spectral dimension and the directional dimensions. Thus, it is not feasible to calculate the radiative transfer for each absorbing and emitting line of the atmospheric gases; instead a limited number of spectral bands are defined for which the radiative transfer is calculated. For shortwave irradiance the radiative transfer in most current models is only considered for the direct solar beam, upward diffuse irradiance and downward diffuse 15 irradiance. Thus, the complex directional variability of shortwave irradiance is not considered. This is called the two-stream approximation. For the longwave irradiance the two-stream approximation is also used in most current models. To sum up, many approximations are currently made in order to calculate radiative transfer in weather and climate models, and even with these approximations they are still computationally very expensive.



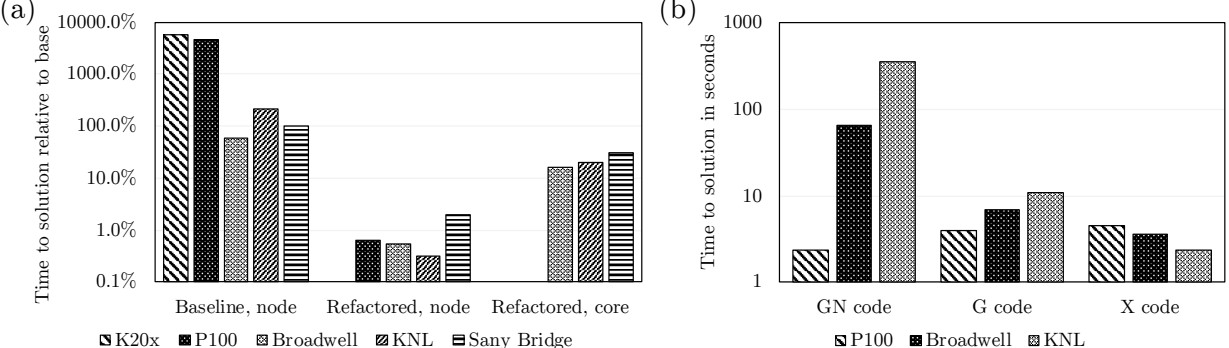

**Figure 27.** Optimisation of ACRANEB2 radiation dwarf from Poulsen and Berg (2017). (a) Time-to-solution relative to the baseline implementation on a SNB node for transt in the full acraneb2 dwarf. The baseline code performance on single nodes of different architectures is shown to the left. Bars in the middle show the single node performance of the refactored codes, and the right bars show the single core performance of the refactored codes. Using the Cray compiler on NVIDIA GPU K20x and the PGI compiler on NVIDIA GPU P100, and the Intel compiler on Intel Broadwell (processor 2S E5-2699v4), KNL (type 7210) and Intel Sandy Bridge (2S E5-2680v1). Single-core performance is not sensible for the GPU. (b) Time-to-solution for the three different code bases on three different architectures. X is the Xeon target code. G is the GPU target code using the data structures as X, but split into seven chunks. GNM is the GPU target with transposed data structures as compared to X and G, reformulated power function and even more splits (into 12 chunks).

Given these many ways, radiation schemes can be approximated in various ways. For radiation schemes in medium to long range weather models, it makes sense to utilise more spectral bands to capture the complex shortwave radiative heating in the stratosphere while saving resources in the spatial and temporal dimensions by running the radiation scheme at coarser resolution and intermittently relative to the general model time stepping. Here we have chosen to work with the ACRANEB2

radiation scheme, which has been designed for short range weather models (Mašek et al., 2015; Geleyn et al., 2017). Detailed descriptions of the physics in ACRANEB2 have been made by Mašek et al. (2015) and Geleyn et al. (2017) for the shortwave and longwave radiation, respectively. More details about the dwarf can be found in Müller et al. (2017).

This dwarf has been entirely refactored which leads to massive speedups on CPU, GPU and KNL processors (Figure 27a). To explore the full code optimisation potential three different code bases have been created which use different data layouts

(Figure 27b). These results show that different data layouts can have a huge impact on performance. For more details about the work on optimising the radiation dwarf we refer to Poulsen and Berg (2017).

Further speedup beyond optimising the code can be achieved by computing some of the radiation at coarser resolution. Radiation schemes used in operational models consider only radiation in vertical columns. At resolutions around 1 km or finer we find that the accuracy of the radiation parameterisation degrades because it does not consider 3D effects like cloud shadows.

Computing radiation at coarser resolution compared to the rest of the model has the potential to improve accuracy and gives us another significant speedup. Experiments with ACRANEB2 indicate that we can expect a speedup of 10 times in realistic scenarios. A series of full 3D experiments with an operational model at DMI using 65 vertical levels and 800*600 grid points



give a reduced execution time of about 25% for a cheap version of the radiation scheme and by about 70% if the radiation scheme is run in an expensive version every time step.

## 4.2 Cloud-microphysics scheme CLOUDSC

The CLOUDSC dwarf is the parametrizaton scheme for cloud and precipitation processes in the IFS, described by prognostic
equations for cloud liquid water, cloud ice, rain, snow and a grid-box fractional cloud cover. The cloud scheme represents the sources and sinks of cloud and precipitation due to the major generation and destruction processes, including cloud formation by detrainment from cumulus convection, condensation, ice deposition, evaporation, hydrometeor collection, melting and freezing. The scheme is based on Tiedke (1993) but with an enhanced representation of the ice-phase in clouds and precipitation. A multi-dimensional implicit solver is used for the numerical solution of the cloud and precipitation prognostic equations.
A more detailed description of the formulation of the parametrization can be found in ECMWF (2015) with further discussion in Forbes and Tompkins (2011) and Forbes et al. (2011).

Independent of ESCAPE there has been some work on optimising CLOUDSC for GPUs (Xiao et al., 2017). This work added a hybrid MPI and OpenACC approach to the CLOUDSC dwarf and explored different optimisation methods for the GPU. Xiao et al. (2017) found that the performance is highly dependent on the size of the blocking as defined by the parameter
NPROMA. The CPU version is most efficient with a relatively small size of the NPROMA-block between 12 and 128 while the GPU version is most efficient with NPROMA over 10,000. In terms of pure computation time (Xiao et al., 2017) found the K80 GPU to be about two times faster than one CPU socket with 12 cores. If data movement between CPU and GPU is included the GPU was slightly slower than the CPU socket. We expect that having a faster interconnect like NVLink between CPU and GPU would improve the total GPU runtime significantly.

The work on CLOUDSC in ESCAPE focused on using domain specific languages via the CLAW DSL (Clement et al., 2018). In particular the use of the Single Column Abstraction (SCA), where physical parameterisations are defined solely in terms of a single horizontal column, enables domain scientists to define physics equations purely in terms of vertical dependencies without needing to account for parallelisation issues. The CLAW DSL then inserts loops over the data-parallel horizontal dimension specific to the hardware architecture and programming model (OpenMP, OpenACC) via source-to-source translation, allowing
multiple architectures to be targeted from a single source code. A GPU implementation of the CLOUDSC dwarf generated by automated source translation tools has been used to generate similar performance results to the ones presented by Xiao et al. (2017) on K80 GPUs via the OpenACC backend of the CLAW DSL.

## 5 Comparison between different discretisation methods

Looking at a higher level comparison between different discretisation methods represented by our dwarfs, Kühnlein et al.
(2018) compared the IFS-FVM (using MPDATA and GCR elliptic solver) with the spectral-transform formulation of IFS. Figure 28 shows close agreement between both model formulations for the baroclinic instability benchmark. Kühnlein et al.



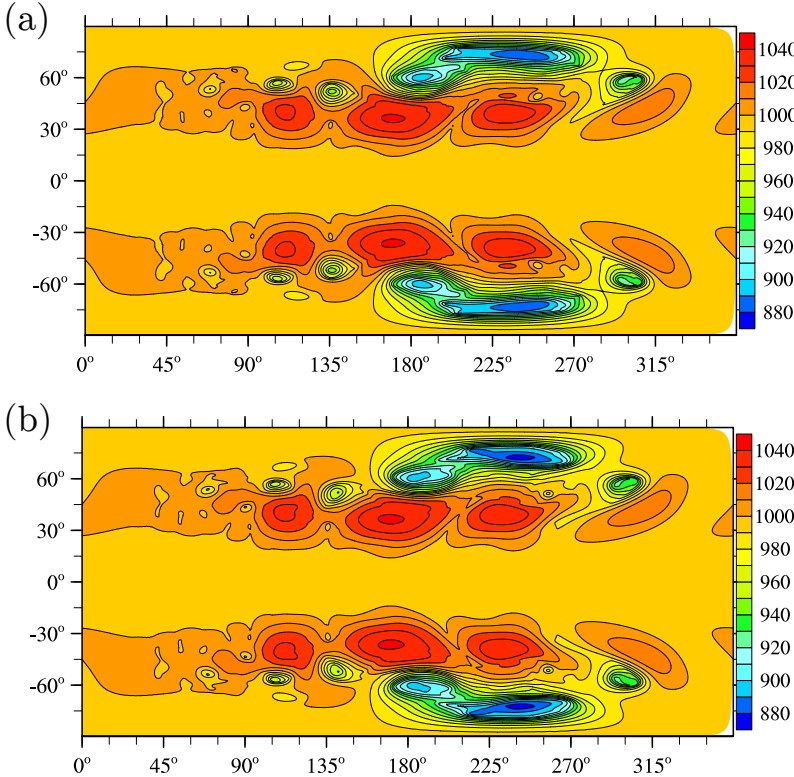

**Figure 28.** Comparison of the surface pressure with (a) IFS-FVM and (b) the spectral-transform IFS for the baroclinic instability benchmark after 15 simulation days. The depiction illustrates comparable solutions with the different discretisation methods.

(2018) also demonstrate that IFS-FVM operates at a competitive computational performance compared to the operational spectral-transform IFS formulation at ECMWF.

In terms of performance and energy efficiency of the entire forecast we compared ALARO (based on spectral transform using biFFT) with COSMO-EULAG (based on finite difference with MPDATA and elliptic solver). The results are shown in

5    Figure 29. More details about this comparison can be found in Van Bever et al. (2018a). This comparison does not include many of the optimisations presented in this paper. As shown for the individual dwarfs both models are expected to still have significant potential for optimisations. We do not know if there will be a clear winner.

Having energy vs. runtime plots like in Figure 29 is very important for NWP applications due to the severe restrictions in terms of runtime and energy cost. As described before the forecast cannot start earlier because not all of the required

10    observations would be available and it cannot finish later because it would not reach our customers on time. Finding the best compromise between energy consumption and runtime is therefore crucial.





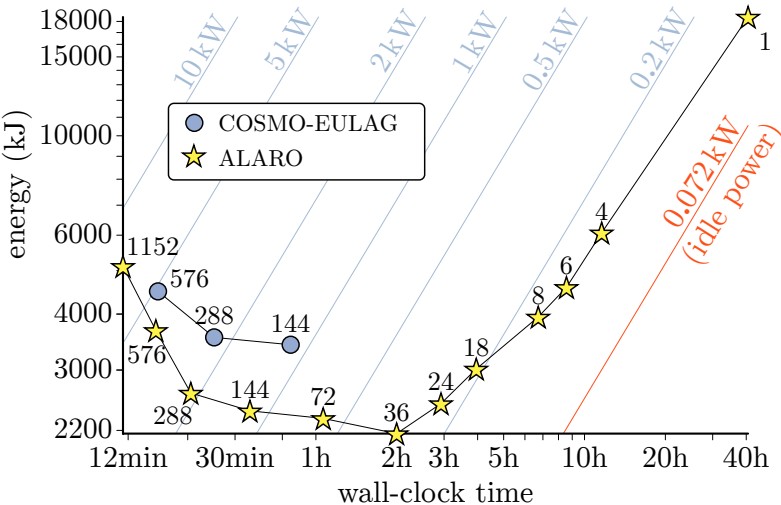

**Figure 29.** Full model forecast run. Comparison of energy efficiency between ALARO (based on spectral transform using biFFT) and COSMO-EULAG (based on finite difference with MPDATA and elliptic solver). Log-log plot of the energy consumption vs. wall-clock time for the ALARO 2.5km and COSMO-EULAG 2.2km reference configuration. Only pure MPI jobs were simulated. The colours of the data points and added lines have the same meaning as in Figure 15.

## 6 Conclusions

This paper gives an overview of the work done in the ESCAPE project. The project aims at preparing weather prediction and climate models for new computing architectures towards exascale machines. The project introduced the concept of fundamental building blocks called dwarfs. Feedback from the European and international community at our dissemination workshops and

at international conferences have shown that this work was well received and the entire NWP community has a strong interest in pursuing the creation of dwarfs. Having dwarfs reduces the complexity of the code and enables HPC centres and hardware vendors to significantly improve the performance of these smaller and self-contained portions of code. Another important development is a common data structure and mesh handling framework as given by Atlas. Atlas has been extended in ESCAPE to support limited area grids and DSL with GridTools.

We have started to incorporate the obtained code optimisations into operations. The first results of this effort look promising. Besides optimisations of the existing code improved algorithms have been developed which are specifically targeted at improving performance on large scale systems. We developed a multigrid preconditioner for the elliptic solver, found a HEVI time-integration scheme with significantly improved stability, explored alternative finite difference methods on the sphere and explored FFTs and spherical harmonics on optical processors.

Code optimisations performed in the ESCAPE project targeted Intel CPUs, Intel Xeon Phi processors, NVIDIA GPUs and Optalysys optical processors. In NWP applications, the bottleneck in terms of performance is usually the memory bandwidth between processor and main memory. Having fast interconnect like NVLink and NVSwitch can provide a massive speedup.





Having all of the processing units used by the simulation connected with such a fast interconnect is still a challenge. Using accelerators only for a small part of the code destroys a lot of the benefit in terms of overall cost if the CPUs are idle while the accelerators perform their computations. We either need to move a large part of the code to the accelerator or we need to overlap computations on the CPU with computations on the accelerator. Both of these options still require more work.

The best way to compare different processors that we could find for NWP applications is to compare the number of devices and the energy consumption required to reach a certain runtime. The spectral transform showed a huge benefit from using batched matrix multiplication which involves adding a large number of zero operations. This makes it impossible to compare any metrics that involve the number of operations performed by the processor.

Our work on the radiation dwarf indicates that the changes between different architectures required to reach good perfor-
mance can be very substantial. DSLs are a very promising tool to enable good performance on multiple architectures while still having one code base. However, designing a domain specific language that is user friendly and at the same time close to hand tuned performance on each architecture is still challenging and further work needs to be done.

Comparing different methods requires to include all costs. Spectral transform has always been considered as being poorly suited for large supercomputers due to the large amount of communication per time-step and fairly large communicators.
Our work indicates that thanks to much larger time-steps and better strong scaling than for halo communication the overall communication cost is not necessarily worse than for other methods. Again overlapping different parts of the model such that useful computation can be done while data is communicated is a way forward and needs to be high priority in future research.

When moving towards very high resolution global simulations of O(1km) or less and considering exascale computations on a variety of emerging HPC architectures, there is a continued interest and need in pursuing fundamentally different algo-
rithmic approaches that simply do not communicate beyond a certain halo size while retaining all other favourable properties of the semi-Lagrangian advection and semi-implicit time-stepping (SISL) approach, and such approaches are to be further investigated in ESCAPE2.

*Code availability.* The code of the ESCAPE dwarfs is available under license, which permits educational and non-commercial research (see http://www.hpc-escape.eu/media-hub/escape-pub/license for more information).

**Appendix A: Simple theoretical model for heterogenous computing**

We derive in this appendix a simple theoretical model for cost of one time-step on a heterogenous supercomputer. For simplicity we assume that we have one type of accelerator with $N_{acc}$ devices and our code consists of two parts: part A can only run on the CPUs whereas part B can run entirely on the accelerator. We start by deriving the number of CPU sockets $N_{CPU}$ and accelerator devices $N_{acc}$ if we have no overlap between computations on the CPU and the accelerator. In the second subsection we derive
these numbers for the case of having perfect overlap between CPUs and accelerators. The section ends by using these numbers in a simple cost model and with a discussion of limitations of this simple cost model.



## A1 No overlap between CPU and accelerator

In this subsection we assume that we have no overlap between computations on the CPU and computations on the accelerator, i.e. for the total runtime of the model is given by

$$T = T_{\text{A,CPU}} + T_{\text{B,acc}}, \tag{A1}$$

where $T_{\text{A,CPU}}$ is the runtime of part A running on the CPU and $T_{\text{B,acc}}$ is the runtime of part B running on the accelerator. We can obtain these partial runtimes out of the serial runtimes by using Amdahl's law. For simplicity we assume that part A on the CPUs and part B on the accelerator devices have the same proportion $p$ that benefits from parallelisation:

$$T_{\text{A,CPU}} = \left( 1 - p + \frac{p}{N_{\text{CPU}}} \right) \tau_{\text{A,CPU}}, \tag{A2}$$

$$T_{\text{B,acc}} = \left( 1 - p + \frac{p}{N_{\text{acc}}} \right) \tau_{\text{B,acc}}, \tag{A3}$$

where $\tau_{\text{A,CPU}}$ is the runtime of part A on one CPU socket and $\tau_{\text{B,acc}}$ is the runtime of part B on one accelerator device. The total runtime $T$ is a fixed given number in NWP applications. We can therefore solve $T$ for $N_{\text{CPU}}$ and obtain

$$N_{\text{CPU}} = \frac{N_{\text{acc}} \, p \, \tau_{\text{A,CPU}}}{N_{\text{acc}} \left( T + (p-1) \left( \tau_{\text{A,CPU}} + \tau_{\text{B,acc}} \right) \right) - \tau_{\text{B,acc}} \, p}. \tag{A4}$$

For simplicity we assume that one CPU socket has the same hardware cost like one accelerator device. Under this assumption we can minimise the total cost by minimising the total number of devices

$$N = N_{\text{CPU}} + N_{\text{acc}}. \tag{A5}$$

Inserting (A4) into (A5) and minimising this expression as a function of $N_{\text{acc}}$ gives us

$$N_{\text{acc}} = \frac{p \left( \tau_{\text{B,acc}} + \sqrt{\tau_{\text{A,CPU}} \, \tau_{\text{B,acc}}} \right)}{T + (p-1)(\tau_{\text{A,CPU}} + \tau_{\text{B,acc}})}. \tag{A6}$$

We now introduce the speed-up of the accelerator device by defining

$$S_{\text{acc}} = \frac{\tau_{\text{B,CPU}}}{\tau_{\text{B,acc}}}, \tag{A7}$$

where $\tau_{\text{B,CPU}}$ is the runtime of the accelerator part of the code if a CPU version of this code is run on one CPU socket. We denote the fraction of the code that is running on the accelerator with

$$\alpha = \frac{\tau_{\text{B,CPU}}}{\tau}, \tag{A8}$$

where $\tau = \tau_{\text{A,CPU}} + \tau_{\text{B,CPU}}$ is the total runtime on one CPU socket. For $\alpha = 0$ the entire code would need to run on the CPU and for $\alpha = 1$ the entire code would run on the accelerator. This gives us

$$\tau_{\text{B,acc}} = \alpha \, \tau / S_{\text{acc}}, \tag{A9}$$

$$\tau_{\text{A,CPU}} = (1 - \alpha)\tau. \tag{A10}$$





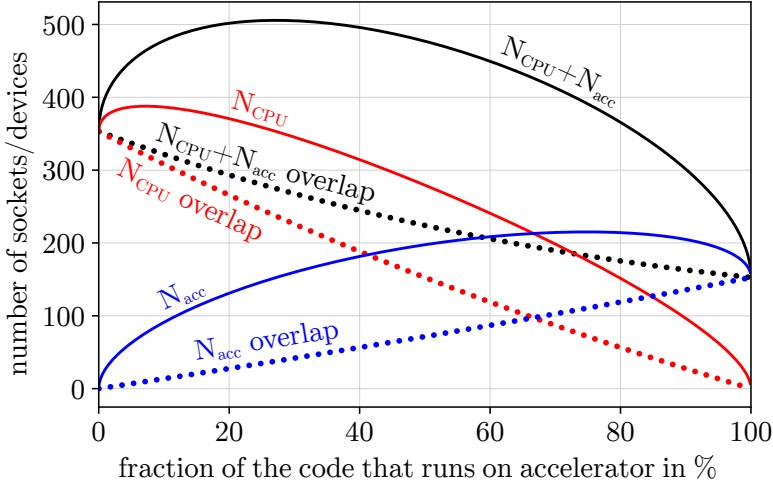

**Figure A1.** Number of CPU sockets or accelerator devices as a function of how much of the code runs on the accelerator. Solid lines show the results with no overlap between CPUs and accelerators as derived in Section A1 while dotted lines show the results with overlap as derived in Section A2. These results use the specific case of the AROME model at Météo-France including the experimentally through linear regression found strong scaling efficiency of $\beta = 53\%$ on $N_{\text{eff}} = 1000$ CPU sockets compared to one CPU socket. One accelerator device is assumed to be two times faster than one CPU socket for this figure, i.e. $S_{\text{acc}} = 2$.

The variables $\tau$, $\alpha$ and $S_{\text{acc}}$ are given parameters for NWP applications.

To make our results easier to interpret we substitute the parallelisation factor $p$ with the strong scaling efficiency $\beta$ when comparing $N_{\text{eff}}$ devices with one single device:

$$\beta = \frac{T_{p=1}}{T} = \frac{1}{N_{\text{eff}}(1-p)+p}. \tag{A11}$$

5    As a specific example we consider the case of Météo-France with the model AROME. Météo-France has two clusters with each containing 1800 bi-socket Broadwell nodes. AROME runs operationally on 180 nodes (360 Broadwell sockets). The runtime of the entire operational forecast is 30 minutes. The forecast range is 30 hours with 50s steps (2160 time-steps). This means that $T = 0.8$s. Through linear regression of real simulations with the AROME model we find that $\tau = 215.19$s and the strong scaling efficiency would be $\beta = 53\%$ for $N_{\text{eff}} = 1000$. Using these values gives us the solid lines shown in Figure A1 if we plot

10    $N_{\text{CPU}}$, $N_{\text{acc}}$ and the sum of the two as a function of $\alpha$.





## A2 Perfect overlap between CPU and accelerator

If we assume perfect overlap between computations on the CPUs and accelerators the number of devices required for our simulation are simply given by solving $T = T_{A,CPU}$ for $N_{CPU}$ and solving $T = T_{B,acc}$ for $N_{acc}$. This gives us

$$N_{CPU} = \frac{p\,\tau_{A,CPU}}{T - \tau_{A,CPU} + p\,\tau_{A,CPU}}, \tag{A12}$$

$$N_{acc} = \frac{p\,\tau_{B,acc}}{T - \tau_{B,acc} + p\,\tau_{B,acc}}. \tag{A13}$$

If we replace (A4) with (A12) and (A6) with (A13) we obtain the dotted curves in Figure A1. The difference between solid and dotted lines in Figure A1 show the overhead caused by having idle devices in the case of no overlap.

## A3 Cost model

The energy consumption of the idle processors during one single time-step $\Delta t$ is given by:

$$E_{idle} = T(N_{CPU}P_{0,CPU} + N_{acc}P_{0,acc}), \tag{A14}$$

where $P_{0,*}$ is the power consumption of one device when it is idle. The additional energy consumption due to running the dwarfs would be during one single time-step $\Delta t$:

$$E_{load} = T_{A,CPU}N_{CPU}\Delta P_{CPU} + T_{B,acc}N_{acc}\Delta P_{acc}, \tag{A15}$$

where $\Delta P_* = P_* - P_{0,*}$ is the increase in energy consumption by not being idle. This gives us the total energy consumption per time-step

$$E_{total} = E_{load} + E_{idle}. \tag{A16}$$

The total energy cost of the machine is therefore

$$C_{total} = \gamma_E E_{total}, \tag{A17}$$

where $\gamma_E$ is the cost per energy.

Supercomputers at NWP centres are typically rented. If we denote the cost of renting one CPU socket with $C_{CPU}$ and one accelerator device with $C_{acc}$ we get for the cost of the hardware during one time-step

$$C_{hw} = T(N_{CPU}C_{CPU} + N_{acc}C_{acc}). \tag{A18}$$

Therefore, we get for the total cost of one time-step

$$C_{total} = C_E + C_{hw}. \tag{A19}$$

As a specific example we consider again the case of running the NWP model AROME at Météo-France. The monthly rent of the two clusters at Météo-France is 500k€. We measured that an idle node uses about 0.03kW, while a node running



AROME uses 0.3kW. The cost of energy is approximately 0.15€/kWh. For simplicity we assume that one accelerator device has the same energy consumption and renting cost like one CPU socket, i.e. $P_{0,\text{CPU}} = P_{0,\text{acc}} = 15\text{W}$, $\Delta P_{\text{CPU}} = \Delta P_{\text{acc}} = 135\text{W}$, $C_{\text{acc}} = C_{\text{CPU}} = 2.7 \times 10^{-5}/\text{s}$. Using these values in equation (A19) gives us the results in Figure 4.

### A4    Limitations of the cost model

The reader needs to be aware of the simplifications used in deriving the simple cost model in this section. As described before the purpose of this cost model is to show that having no overlap between computations on the CPU and on the accelerator can cause a significant overhead on hybrid machines. The cost model does not explicitly consider data transfer between CPU and accelerator. One could assume that the data transfer is included in the speedup of the accelerator but then the speedup would depend on $\alpha$ (i.e. how much of the model is running on the accelerator) and if all of the code ported to the accelerator is in one

block resulting in one data transfer in every time step.

The cost model assumes that one CPU socket has the same energy consumption like one accelerator device and the code achieves the same strong scaling efficiency on both. As stated before these assumptions might be wrong in reality. Same scaling efficiency will be difficult to achieve because accelerators usually require a lot of inherent parallelism inside each device.

Despite these simplifications, our cost model illustrates the overhead caused by not overlapping computations on CPU and

accelerator. The reader needs to be aware that in order to make good use of accelerators we need to port a large part of the model to the accelerators and we need to invest in research on overlapping different computations.

*Author contributions.*  Andreas Müller created the cost model based on an initial version from Philippe Marguinaud, performed energy measurements for the spectral transform dwarf, implemented the optimised spectral transform for post-processing, worked on the implementation of the MPDATA dwarf, supported maintenance for all dwarfs and assembled the paper. Willem Deconinck is the main developer of the data

structure framework Atlas and contributed the work on improving and extending Atlas. He also was involved in creating, supporting and maintaining all of the dwarfs. Christian Kühnlein and Piotr K. Smolarkiewicz are the main developers of IFS-FVM. They contributed text for the description of the elliptic solver and MPDATA dwarfs and the comparison between IFS-FVM and the spectral transform IFS. Gianmarco Mengaldo was involved in the creation of the dwarfs and provided improvements to early drafts of the manuscript. Michael Lange and Valentin Clement provided the DSL work for the cloud microphysics dwarf. Nils Wedi and Peter Bauer supervised the entire ESCAPE

project as coordinators. They also provided improvements to early versions of the manuscript. Nils Wedi performed the measurements for the cost profiles in Figure 5. Michail Diamantakis implemented the semi-Lagrangian dwarf. Sarah-Jane Lock supported the work on HEVI time integration methods. Mats Hamrud, Sami Saarinen and George Mozdzynski were involved in creating initial versions of the dwarfs. Sami Saarinen also contributed the code for energy measurements on the Cray XC40 supercomputer. Daniel Thiemert provided project management for the ESCAPE project. Michael Glinton and Pierre Bénard implemented the GRASS model and provided the comparison between

high-order finite difference and spectral transform methods. Fabrice Voitus and Charles Colavolpe worked on HEVI time-stepping methods. Philippe Marguinaud and Yongjun Zheng provided an initial version of the cost model and performed simulations of communication costs. Joris Van Bever, Daan Degrauwe, Geert Smet and Piet Termonia provided the energy and runtime measurements for ALARO, analysed the components of the RMI ensemble prediction system and implemented limited area support in Atlas. Kristian P. Nielsen, Bent H. Sass, Jacob



W. Poulsen and Per Berg provided the work on the radiation dwarf. Carlos Osuna and Oliver Fuhrer provided the work on GridTools and were involved in adding GPU support to Atlas. Michael Baldauf contributed the work on HEVI DG. Mike Gillard and Joanna Szmelter worked on improving the preconditioning for the elliptic solver. Enda O'Brien, Alastair McKinstry, Oisín Robinson, Parijat Shukla and Michael Lysaght provided the LAITRI dwarf and created initial CPU optimisations and OpenACC ports for all of the dwarfs. Michał Kulczewski,

Milosz Ciznicki, Wojciech Piątek, Sebastian Ciesielski, Marek Błażewicz, Krzysztof Kurowski, Marcin Procyk, Pawel Spychala and Bartosz Bosak worked on performance modeling using the roofline model which allowed a better understanding of the experimental measurements. Zbigniew Piotrowski and Andrzej Wyszogrodzki compared different time-integration methods, supported optimisation of IFS-FVM and provided energy measurements of COSMO-EULAG. Erwan Raffin, Cyril Mazauric, David Guibert, Louis Douriez and Xavier Vigouroux provided the CPU optimisations for the spectral transform and MPDATA dwarfs. Alan Gray and Peter Messmer contributed the GPU op-

timisations for the spectral transform and MPDATA dwarfs. Alexander J. Macfaden and Nick New contributed the work on using optical processors for the spectral transform.

*Acknowledgements.* The ESCAPE project has received funding from the European Union's Horizon 2020 research and innovation programme under grant agreement No 671627. This work has been also supported by HPC resources provided by Poznan Supercomputing and Networking Center and corresponding research activities under the MAESTRO grant number DEC-2013/08/A/ST6/00296 in National

Science Centre (NCN).



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
