# Peer review of "The ESCAPE project: Energy-efficient Scalable Algorithms for Weather Prediction at Exascale"

_Geoscientific Model Development, 2018_

## Referee Comment (RC1) · Anonymous Referee #1 · 25 Feb 2019

Overall impression

The manuscript under consideration was submitted to GMD as a "development and technical paper". The paper topically spans mathematical formulation, numerical integration techniques, parallelization strategies, language-specific aspects of implementation, hardware-specific optimizations, hardware construction, and operational considerations in the context of both global-circulation and limited-area models.

As the main conclusion from the hereby review, I propose to significantly shorten the article (currently 50-pages and 30 figures) and change its type to a "Review and perspective paper" to match the stated intention of the authors to create "*the flagship publication for the EU project ESCAPE ... introduce the concept of weather & climate*

*dwarfs and discuss first results in terms of optimization and performance portability*". This path seems to me as much more reasonable than working towards matching the requirements for a "development and technical paper" as defined by GMD guidelines.

The authors aim at: (i) providing a technical report touching upon current-hardware-specific performance measures, (ii) structuring the work as a research paper, and (iii) presenting a project-promoting overview article. These aims are incompatible in my opinion, and trying to achieve all of them at once results in unclear target audience and an apparent lack of a storyline, despite high potential for strong conclusions to be based on the presented results.

Notwithstanding, I do see a point in publishing such a "perspective" paper with the aim of promoting the project results and giving due credit to participating parties. I expect such a shorter "perspective" paper to achieve a higher impact and I encourage the editorial team to offer this option to the authors.

Code availability

The "Code availability" section on page 42 is derisory. The standard that GMD is fostering among the community is to enable readers and reviewers to reproduce results presented in GMD papers. Here, the reader is only given a link to project website where one may not even find a properly defined software license – just a statement that it "permits free of charge use for educational and/or non-commercial research". The final sentence of the referenced website reads: "*If you wish to access any of the implementations, please contact us via the contact form and we will provide further information on the process of obtaining a license*". This stands in clear opposition to the anonymous public access recommendations of GMD. Basing on an educated guess (the most one can anonymously base on given the above), I consider the results presented in the paper as not independently reproducible for reasons including software and hardware availability, as well as lack of availability of the details of the test cases.

As outlined above, a solution would be to move much of the technical details to another publication (a technical report issued by one of the participating institutions – several of those are already cited) and present a "perspective" paper for which GMD does offer an exception in terms of reproducibility level (see https://www.geoscientific-model-development.net/about/manuscript_types.html).

Nevertheless, the code availability section should contain comprehensive information, and clearly inform about the **code** availability, not license availability. Please point to repositories, state precisely the licenses or clearly indicate if the code is not publicly available or its reuse is constrained. In case of lack of public availability, GMD requires to state the reasons for it. Please include information for all the software that was essential in obtaining the presented results, including the participating weather prediction models: IFS, ALARO, COSMO-EULAG as well as the described tools such as GRASS, CLAW and GridTools (the https://github.com/eth-cscs/gridtools repository linked from the GridTools website does not exist as of time of writing this review).

Code availability for hardware-specific tools such as those essential in GPU code development should also be included in the section (see doi:10.1002/2016WR020190 for a recent discussion in the context of hydrological modelling), and in my opinion should also be included in the discussion if a proper "perspective" is to be given. The paper gives an overview of several paths forward in NWP systems development and aims at discussing longer-term strategies. Such discussion calls for mentioning which optimisation strategies are prone to the vendor lock-in threat.

The dwarf nomenclature and technicalities

The authors highlight throughout the paper the concept of separation of concerns in software engineering using the notion of a "dwarf" which the paper introduces in the context of weather and climate models. The reason to introduce a new term is not given. What does the new concept replace (monoliths)? The adopted term is seemingly wrongly attributed (Colella as opposed to Asanović et al.?) and, in my understanding, used in a misleading way. The reason why the 7 dwarfs of Colella, and later the 13 Berkeley dwarfs, the 7 dwarfs of Symbolic Computation, the 13 Parallel Dwarfs (and likely others) were introduced is that the concept they generalize does not easily fit into existing encapsulation nomenclature of: components, frameworks, layers, substystems, libraries, kernels, modules, services, drivers, plug-ins, controllers, etc. Why dynamical core layers, physics modules and numerical libraries are to be renamed? In principle, why not – let us embrace the introduced notion of Weather & Climate Dwarfs, but please do clarify in the paper the reasons to introduce the new nomenclature and clearly differentiate it from existing solutions.

Moreover, among the community, dwarfs are being defined as computation/communication idioms (e.g., OpenDwarfs project) or (design) patterns of high-performance computing (e.g., the mentioned work of Asanović) – is it not misleading to refer to the "components of an Earth system model" borrowing the term that was introduced to define idioms/patterns?

In the conclusions section, the authors note that "*... the entire NWP community has a strong interest in pursuing the creation of dwarfs*". Such community-wide initiatives are, in my opinion, only feasible if backed by interface standardization and "liberation" of the software in question. Please do elaborate on how (technically and legally) the reusability of the envisaged community "dwarfs" is to be assured? Furthermore, the dwarfs' interfaces will determine if the sought after capability of overlapping computations will be feasible.

Other remarks

- Optimization is the leitmotif of the paper, and is exemplified with detailed results obtained with current hardware and described in zealous detail (a somehow anecdotal example: replace division by constant with multiplication by the

reciprocal calculated in advance). Yet at the same time, the paper aims at providing "*strategy to evolve weather and climate prediction models to next-generation computing technologies*", exascale is mentioned in the title. Please extend the conclusions section by indicating which of the employed optimizations have the chance to offer "software sustainability" over the years (also in the context of software maintainability tradeoffs), and where, e.g., indirection would be the key to performance (DSLs)?

- There is an abundance of arcane nomenclature in the paper used without introduction and spanning a broad range of domains, e.g.: "*loop fusion optimization*", "*generic reduced Gaussian grid*", "*parallelogram*", "*a perturbative method which determines the (constrained) optical phase*", "*Spherical Harmonics TCo639 test case*", "*baroclinic instability benchmark*", "weighted line Jacobi method", "*V-cycle configuration*", "*major generation and destruction processes, including cloud formation by detrainment from cumulus convection*";

- There is an imbalance in the level of detail of different sections of the paper, e.g. the well-established MPDATA is introduced with an outline touching upon numerical analysis, while the "created" "global shallow-water model named GRASS" is presented with just a reference to a pair of submitted papers; there is a two-page-long introduction to the application of spectral transform methods in NWP, while less than that is devoted to the entire discussion of physics dwarfs; authors do admit that this was intentional (p. 5 last paragraph) for some of the work has not been published elsewhere – this however rather supports publishing it elsewhere than sneaking into an overview paper.

- Some of the tools mentioned target Fortran development (e.g., CLAW DSL) while other cater to a wider set of technologies (e.g., Atlas), this is not mentioned explicitly and the reader is left without a clear statement if the proposed directions of development deviate or not from the Fortran ecosystem;

- The conclusions section contains statements of overly contrasting time horizons: on the one hand, the authors mention "*adding a large number of zero operations*" what is explained in the text to be caused simply by lack of support for a particular feature in the current version of a third-party library; on the other hand, prerequisites and challenges for subkilometer global simulations are mentioned. Please reconsider what are the main project conclusions worth to be listed in the concluding section and abstract.

- That the great majority of referenced works is [co]authored by the manuscript authors amplifies the feeling of some of the methodology, design or vendor choices being given without a proper context on the alternatives:

  - How representative is the chosen set of models (IFS, ALARO, ALADIN and COSMO-EULAG) among the "competition" and how the considered speed-up techniques compare with what has been explored recently (see, e.g., doi:10.1175/BAMS-D-15-00278.1 and references therein)?

  - How the proprietary software and hardware solutions like cuBLAS/cuFFT and NVLink/NVSwitch compare to those provided by other vendors?

  - Overlapping CPU-GPU computation strategy for dynamics/physics has been recently discussed in GMD in context of cloud-resolving simulations (doi:10.5194/gmd-2018-281, e.g. fig. Fig. 1), could the discussion here be supported with references to existing solutions from other domains?

  - mentions of GPU-resident weather forecasting call for citing other recent works (e.g., doi:10.1175/BAMS-D-14-00114.1)

  - What are the alternatives for the used radiation and cloud-physics schemes, are the chosen ones representative of what the community envisages for the mentioned global subkilometer-scale future simulations?

  - Are the CLAW and GridTools DSLs the sole solution available in this context?

- The word "code" is used in a somehow casual way, e.g. "*redesign of the algorithms and codes*", "*our work on optimising codes*", "*code used for data assimilation*", "*models from which the dwarf code originated*"; let me suggest to consider employing more cross-domain notions of implementation, software, etc; similar nomenclature issue: restructure vs. refactor;

- Please remove any mentions of internal labels within the code – this information is unneeded for a research paper audience: "halo_exchange subroutine", "compute_fluxzdiv", "this%geom%node2edge_sign".

- Please limit the use of acronyms/short-forms, and remove those clearly unneeded: PSNC in Fig. 3, EBTI on page 24, GP_dynamics/SP_transforms/SI_solver/RAD in caption of Fig. 5, semi-Lagrange in Fig. 6; Some references are listed with DOI number, some without - please be consistent; FORTRAN/Fortran, TRAP2/Trap2 spelling - please be consistent; Among the affiliations listed, some are given with detailed street addresses, some without - please be consistent.

- The title of the paper reads "*The ESCAPE project: Energy-efficient Scalable Algorithms for Weather Prediction at Exascale*". Exascale is not discussed or defined and barely mentioned only in the conclusions, while the phrase "weather & climate" is used throughout the paper.

- Statements such as "*ECMWF is world leading in terms of track forecast*", "*extreme computational capabilities typically required in operational forecast production*", "*[IFS code] has been continuously optimized over multiple decades*", "*Feedback from the European and international community at our dissemination workshops and at international conferences has shown that this work was well received*" are, in my opinion, good candidates for removal when shortening the paper – please avoid promotional language and statements which are not falsifiable; another candidates for removal are numerous vague statements: "*most*

*speedup seems to be due to avoiding some of the temporary arrays*", "*some more fundamental changes which are more difficult to apply*", "*whole cycle might employ some form of smoother/solver*", "*has to be wisely chosen according to the cluster hardware*", "*we do not know if there will be a clear winner*", "*The first results of this effort look promising*".

---

## Referee Comment (RC2) · Anonymous Referee #2 · 22 Mar 2019

The paper presents a review of the work done in the Escape project. This reviewer is familiar with some of this work and reporting the work done as part of the project is certainly of interest to the community. Unfortunately, the paper is not well written, if is full of mistakes, informal language and confusing or unclear explanations. I have read and documented changes as far as page 17, but this has taken a long time as the paper has not been properly proof read before submission. Referee #1 calls for a substantial revision, and a possible change of paper type, therefore there doesn't seem much point in fully detailing necessary changes beyond this point. A shorter, more focused article as a review and perspective paper would improve the readability and is probably more appropriate for the content.

I include the detailed points below, which need to be addressed.

[Figure]

Page 2 line 7 Weather prediction (models?)

Line 20, sentence reads as if heavy precipitation patterns could lead to tropical cyclones, modify

Line 21 being satisfying "being"

Intro 1st paragraph is rather clumsy, there are plenty of reasons improved forecasts in general would have economic and societal benefit besides heavy precipitation. The need to improve resolution is given as the main motivation for improved forecasts but then Climate is thrown into the following sentence. Improved resolution versus complexity for improved climate is a matter of debate. There is no mention of current resolutions for the reader to compare 1km. What does global resolution range mean? The last sentence is also confusingly written. This paragraph needs re-rewriting with proper thought on what is the motivation for improving resolution of weather and climate simulations. There are plenty of justifications.

Line 27 "guarantee the continued efficiency" is probably a bit strong. "Enable efficient implementations of " or similar is probably more realistic.

Line 33 "The authors there" is rather informal language.

Page 3. Line 7 Citation needed to back authors claim that ECMWF is world leading in terms of track forecast. It would be quite odd for a higher resolution forecast to "compromise the accuracy" of a lower resolution, especially as this is the motivation for escape.

The paragraph on supercomputers is also rather clumsily written and there should be a reference. An obvious one would be "Crossing the Chasm : how to develop weather and climate models for next generation computers?". Some of the authors of this paper are also authors of the chasm paper. Other references would also be beneficial.

Section 2, line 18, The use of "at once" suggests either "immediately" or "at the same time" but "too big" implies neither. This sentence needs re-writing.

Page 4 The text in figure 1 is far too small to read.

Figure 2a "halo's" has no apostrophe.

Page 5 figure 3 What does the vertical line denote? Line 5 what does "released dwarf" mean?

Page 7 line 5 "use 3D var like the " is too informal language, use for example or e.g. Page 7 line 23 "need" -> "needs"

Page 8 Figure 8. The figures are not very well explained. A careful reading of a technical appendix is required and even then it is not clear how they illustrate the point that overlapping the comms of data transfer is necessary. The main constraint for NWP is time to completion, it is not obvious that the assumptions (necessarily) made to express the scaling in terms of energy are sufficiently universal to infer the point the authors wish to make. Whilst using less energy reduces the cost, the run-time constraint is the primary motivation for NWP. This section needs to be re-worked to describe the performance model and explain how any figures used enable the authors draw the conclusions that they do.

Page 9 section 3 line 23 Last sentence. What does this mean?

Page 10, refers to figure 5 which is on page 11. How are the profiles produced? What machine are they produced from, especially the node count etc would change the profile? The "Anticipated" future are these profiled from data or some performance model? The 1.25km run may well have a different profile on a different machine?

Page 11 refers to figure 7 which appears on page 13, the text "like on the left" is rather informal and inappropriate for a scientific paper.

Page 12 refers to figure 8 which appears on page 14. In the figure, there is no open diamond referred to in the legend and the caption, the open rectangle referred to in the caption doesn't appear but a dash or line does which isn't referred to in the caption.

Pages 12 and 13 there is a complicated discussion of code changes. This would be illuminated by some code fragments as examples.

Figure 10. Page 16 The data points are connected with a line (something spreadsheet application does readily), however, the horizontal access is Number of GPUs, which is a discrete variable, so a line graph is wrong. Whilst it not unusual to see such a plot, the authors are not predicting the speed up on 16.5 GPUs so why the line? The plot should be re-drawn appropriately.

Page17 lines 6-15 the paragraph discusses the pack-unpack operation. The sentence sender and receiver share their memory layout as the may differ is confusing. How can they share a memory layout if they are different? What was the change that made the performance improvement? Were the pack and unpack scanning memory unnecessarily?

The final paragraph on page 17 is a discussion of implementing some of the GPUs on CPUs. Again some code fragments here would be helpful. If the GPU optimisations are two intensive, how can they be used elsewhere for other architectures? The stated goal of Escape is to re-assemble the models from optimised dwarfs. How is this managed? If there are conflicting optimisations how are they resolved? Is single source code possible?

Figure 11, the "barrier" optimisation is not clearly explained. Again, line plots of non-continuous variable should be changed.

---

## Referee Comment (RC3) · Anonymous Referee #3 · 8 Apr 2019

Review of
**The ESCAPE project: Energy-efficient Scalable Algorithms for Weather Prediction at Exascale**
by A. Mueller et al.

**General comment:**
In this manuscript an overview about the achievements in the ESCAPE project is given. The main concept is explained, some of the developments are explained in details and finally some tests are mentioned.
Although I think that this manuscript is a valuable contribution for GMD, I cannot recommend to accept the manuscript in the actual state. The manuscript must be revised in a substantial way before it can be considered again. Therefore I recommend major revision of the manuscript. In the following I will explain my concerns.

**Major issues**

1. Balance of the manuscript:
   The manuscript is very long and not really balanced. Some parts are explained in details, as e.g. the development of the MPDATA dwarf, but some parts are just mentioned. Especially for the very shortly explained parts, there are very often references to technical reports, i.e. documentation which is generally not peer reviewed. Although there are some performance tests, there is only one figure showing a test for atmospheric flows, and also this test is only marginally described.

   I would recommend to significantly reorganize the manuscript, maybe also considering to split the manuscript into three parts: First, an overview part, where mostly the concept and the new architecture can be explained in a concise way. Second, a model description part, i.e. a detailed description of the different parts of the model, especially of the parts, which are contained in the technical memoranda but not described in peer-reviewed literature. Third, a part dedicated to test cases for atmospheric flows - and maybe also clouds and radiation, since these parts are also included into the model.

   Especially test cases of atmospheric flows would be very interesting, since it is not clear if all the new models represent the atmospheric flow and other atmospheric phenomena in a physically consistent way. Therefore I highly recommend to use well-documented test cases for atmospheric flows, as e.g. Jablonowski & Williamson (2006). It would be interesting to see also tests for clouds and radiation, although I am not really aware of large scale tests, beyond the standard tests as e.g. Weismann & Klemp (1982).

2. Selection of the dwarfs:
   It is not really clear how and why the different dwarfs were chosen. Although I think that this is a well chosen sample of possible models, it should be justified much better. Especially, the choice of the shallow water model is not really clear, because no real results of this model are shown in the manuscript. Therefore, I recommend to describe the choice of the models is a clearer way.

**Minor issues:**
Cost model:
The benefit of the cost model is not really clear to me. It is introduced in a comparable length as the dwarfs, but it is not really clear why this is so important for the whole manuscript, justifying a large part in the appendix.

**References**

Jablonowski, C., and D. L. Williamson, 2006: A Baroclinic Instability Test Case for Atmospheric Model Dynamical Cores, Quart. J. Roy. Met. Soc., 132, 2943-2975

Weisman, M. and J. Klemp, 1982: The Dependence of Numerically Simulated Convective Storms on Vertical Wind Shear and Buoyancy. Mon. Wea. Rev., 110, 504-520.

---

## Author Comment (AC1) · 4 Jun 2019

**Response to the referee 1 for "The ESCAPE project: Energy-efficient Scalable Algorithms for Weather Prediction at Exascale"**

(grey background: text of the reviewer comment, white background: our response)

> Overall impression
>
> The manuscript under consideration was submitted to GMD as a "development and technical paper". The paper topically spans mathematical formulation, numerical integration techniques, parallelization strategies, language-specific aspects of implementation, hardware-specific optimizations, hardware construction, and operational considerations in the context of both global-circulation and limited-area models.
>
> As the main conclusion from the hereby review, I propose to significantly shorten the article (currently 50-pages and 30 figures) and change its type to a "Review and perspective paper" to match the stated intention of the authors to create "the flagship publication for the EU project ESCAPE ... introduce the concept of weather & climate dwarfs and discuss first results in terms of optimization and performance portability". This path seems to me as much more reasonable than working towards matching the requirements for a "development and technical paper" as defined by GMD guidelines.
>
> The authors aim at: (i) providing a technical report touching upon current-hardwarespecific performance measures, (ii) structuring the work as a research paper, and (iii) presenting a project-promoting overview article. These aims are incompatible in my opinion, and trying to achieve all of them at once results in unclear target audience and an apparent lack of a storyline, despite high potential for strong conclusions to be based on the presented results.
>
> Notwithstanding, I do see a point in publishing such a "perspective" paper with the aim of promoting the project results and giving due credit to participating parties. I expect such a shorter "perspective" paper to achieve a higher impact and I encourage the editorial team to offer this option to the authors.

We would like to thank the reviewer for this advice. Our first submission attempted to describe all the work that was done in the ESCAPE project. We agree that this was too much material and distracted from the main message of the paper. We have significantly shortened the paper and focus now on properly motivating the dwarf concept and to demonstrate this concept by describing the work for one of the dwarfs in detail.

> Code availability
>
> The "Code availability" section on page 42 is derisory. The standard that GMD is fostering among the community is to enable readers and reviewers to reproduce results presented in GMD papers. Here, the reader is only given a link to project website where one may not even find a properly defined software license – just a statement that it "permits free of charge use for educational and/or non-commercial research". The final sentence of the referenced website reads: "If you wish to access any of the implementations, please contact us via the contact form and we will provide further information on the process of obtaining a license". This stands in clear opposition to the anonymous public access recommendations of GMD. Basing on an educated guess (the most one can anonymously base on given the above), I consider the results presented in the paper as not independently reproducible for reasons including software and hardware availability, as well as lack of availability of the details of the test cases.
>
> As outlined above, a solution would be to move much of the technical details to another publication (a technical report issued by one of the participating institutions – several of those are already cited) and present a "perspective" paper for which GMD does offer an exception in

terms of reproducibility level (see https://www.geoscientific-model-development.net/about/manuscript_types.html).

The rules in terms of code availability and reproducibility for a development and technical paper are according to the link given by the reviewer the same as for a model description paper. The website of GMD states that "When copyright or licensing restrictions prevent the public release of model code, or in the cases where there is some other good reason for not allowing public access to the code, topical editors must still be given access to the model code. Access must also be granted to the reviewers whilst preserving their anonymity, if this is legally possible." We are happy to offer the editor and the reviewers a license to access the code and test cases used in our work. There is no need to make an exception for this paper in terms of reproducibility.

Nevertheless, the code availability section should contain comprehensive information, and clearly inform about the code availability, not license availability. Please point to repositories, state precisely the licenses or clearly indicate if the code is not publicly available or its reuse is constrained. In case of lack of public availability, GMD requires to state the reasons for it. Please include information for all the software that was essential in obtaining the presented results, including the participating weather prediction models: IFS, ALARO, COSMO-EULAG as well as the described tools such as GRASS, CLAW and GridTools (the https://github.com/eth-cscs/gridtools repository linked from the GridTools website does not exist as of time of writing this review).

We have changed the license statement and added information about all software used in the ESCAPE project.

Code availability for hardware-specific tools such as those essential in GPU code development should also be included in the section (see doi:10.1002/2016WR020190 for a recent discussion in the context of hydrological modelling), and in my opinion should also be included in the discussion if a proper "perspective" is to be given. The paper gives an overview of several paths forward in NWP systems development and aims at discussing longer-term strategies. Such discussion calls for mentioning which optimisation strategies are prone to the vendor lock-in threat.

We have added information about the compilers and tools that we used and their versions. As described in the paper we explore vendor specific strategies but we also aim at avoiding vendor lock-in through the use of domain specific languages.

The dwarf nomenclature and technicalities

The authors highlight throughout the paper the concept of separation of concerns in software engineering using the notion of a "dwarf" which the paper introduces in the context of weather and climate models. The reason to introduce a new term is not given. What does the new concept replace (monoliths)? The adopted term is seemingly wrongly attributed (Colella as opposed to Asanović et al.?) and, in my understanding, used in a misleading way. The reason why the 7 dwarfs of Colella, and later the 13 Berkeley dwarfs, the 7 dwarfs of Symbolic Computation, the 13 Parallel Dwarfs (and likely others) were introduced is that the concept they generalize does not easily fit into existing encapsulation nomenclature of: components, frameworks, layers, substystems, libraries, kernels, modules, services, drivers, plug-ins, controllers, etc. Why dynamical core layers, physics modules and numerical libraries are to be renamed? In principle, why not – let us embrace the introduced notion of Weather & Climate Dwarfs, but please do clarify in the paper the reasons to introduce the new nomenclature and clearly differentiate it from existing solutions.

We referred to Asanović et al. because we could not find a good reference for Colella's presentation. We have revised the motivation section for the dwarf concept and we have added new references.

Moreover, among the community, dwarfs are being defined as computation/communication idioms (e.g., OpenDwarfs project) or (design) patterns of high performance computing (e.g., the mentioned work of Asanović) – is it not misleading to refer to the "components of an Earth system model" borrowing the term that was introduced to define idioms/patterns?

Our goal is to identify patterns in terms of computation and communication that are characteristic for weather and climate models. Each of our dwarfs possesses very characteristic patterns of high performance computing. We have added a column to Table 1 describing the characteristics of each dwarf in terms of communication and computation.

In the conclusions section, the authors note that "... the entire NWP community has a strong interest in pursuing the creation of dwarfs". Such community-wide initiatives are, in my opinion, only feasible if backed by interface standardization and "liberation" of the software in question. Please do elaborate on how (technically and legally) the reusability of the envisaged community "dwarfs" is to be assured? Furthermore, the dwarfs' interfaces will determine if the sought after capability of overlapping computations will be feasible.

We have revised the quoted part of the conclusions. We agree from a technical point on clean interfaces, but scientifically disagree that standard interfaces are useful (i.e. plug and play concepts of specific parametrisations or physics and dynamics). In our view standardisation of interfaces in operational NWP software, despite being a very attractive technical concept, it is often detrimental to forecast quality and computational efficiency. It is not the aim of ESCAPE to develop standardised interfaces, even if we explore overlapping concepts. In fact bespoke interfacing may become more important with increasing resolution.

Other remarks

- Optimization is the leitmotif of the paper, and is exemplified with detailed results obtained with current hardware and described in zealous detail (a somehow anecdotal example: replace division by constant with multiplication by the reciprocal calculated in advance). Yet at the same time, the paper aims at providing "strategy to evolve weather and climate prediction models to next-generation computing technologies", exascale is mentioned in the title. Please extend the conclusions section by indicating which of the employed optimizations have the chance to offer "software sustainability" over the years (also in the context of software maintainability tradeoffs), and where, e.g., indirection would be the key to performance (DSLs)?

We have added new results about running the spectral transform dwarf on the supercomputer Summit (currently the fastest supercomputer in the world) and we have added a discussion about the sustainability of the chosen techniques.

- There is an abundance of arcane nomenclature in the paper used without introduction and spanning a broad range of domains, e.g.: "loop fusion optimization", "generic reduced Gaussian grid", "parallelogram", "a perturbative method which determines the (constrained) optical phase", "Spherical Harmonics TCo639 test case", "baroclinic instability benchmark", "weighted line Jacobi method", "V-cycle configuration", "major generation and destruction processes, including cloud formation by detrainment from cumulus convection";

We would like to thank the reviewer for identifying these issues. Many of these statements have been removed by shortening the paper. We have revised the remaining statements.

- There is an imbalance in the level of detail of different sections of the paper, e.g. the well-established MPDATA is introduced with an outline touching upon numerical analysis, while the "created" "global shallow-water model named GRASS" is presented with just a

reference to a pair of submitted papers; there is a two-pagelong introduction to the application of spectral transform methods in NWP, while less than that is devoted to the entire discussion of physics dwarfs; authors do admit that this was intentional (p. 5 last paragraph) for some of the work has not been published elsewhere – this however rather supports publishing it elsewhere than sneaking into an overview paper.

It was never our goal to cover every dwarf in detail. Our goal was to demonstrate our optimisation workflow for a small selection and briefly describe the others. To make this clearer we have now removed the dedicated subsections for the dwarfs which we did not cover in detail. Instead we refer to the corresponding publications in the dwarf table.

- Some of the tools mentioned target Fortran development (e.g., CLAW DSL) while other cater to a wider set of technologies (e.g., Atlas), this is not mentioned explicitly and the reader is left without a clear statement if the proposed directions of development deviate or not from the Fortran ecosystem;

All of the work presented is compatible with Fortran. Even the GPU optimisation is mostly done with OpenACC in Fortran. Gridtools requires currently C++ but we plan to provide Fortran support in future DSLs. We have added a statement about this in the paper.

- The conclusions section contains statements of overly contrasting time horizons: on the one hand, the authors mention "adding a large number of zero operations" what is explained in the text to be caused simply by lack of support for a particular feature in the current version of a third-party library; on the other hand, prerequisites and challenges for subkilometer global simulations are mentioned. Please reconsider what are the main project conclusions worth to be listed in the concluding section and abstract.

We have revised the conclusions section.

- That the great majority of referenced works is [co]authored by the manuscript authors amplifies the feeling of some of the methodology, design or vendor choices being given without a proper context on the alternatives:

  – How representative is the chosen set of models (IFS, ALARO, ALADIN and COSMO-EULAG) among the "competition" and how the considered speedup techniques compare with what has been explored recently (see, e.g., doi:10.1175/BAMS-D-15-00278.1 and references therein)?

  – How the proprietary software and hardware solutions like cuBLAS/cuFFT and NVLink/NVSwitch compare to those provided by other vendors?

  – Overlapping CPU-GPU computation strategy for dynamics/physics has been recently discussed in GMD in context of cloud-resolving simulations (doi:10.5194/gmd-2018-281, e.g. fig. Fig. 1), could the discussion here be supported with references to existing solutions from other domains?

  – mentions of GPU-resident weather forecasting call for citing other recent works (e.g., doi:10.1175/BAMS-D-14-00114.1)

  – What are the alternatives for the used radiation and cloud-physics schemes, are the chosen ones representative of what the community envisages for the mentioned global subkilometer-scale future simulations?

  – Are the CLAW and GridTools DSLs the sole solution available in this context?

  We have added some references to the papers mentioned by the reviewer. A comparison with solutions by vendors not involved in the project is outside the scope of this paper. We

are not able and it is not our goal to provide an exhaustive discussion of all available strategies. The goal of the paper is to present the dwarf concept to the weather and climate community and to demonstrate through a few examples how it can be used to enable close collaboration between NWP centres and hardware vendors.

- The word "code" is used in a somehow casual way, e.g. "redesign of the algorithms and codes", "our work on optimising codes", "code used for data assimilation", "models from which the dwarf code originated"; let me suggest to consider employing more cross-domain notions of implementation, software, etc; similar nomenclature issue: restructure vs. refactor;

We followed the reviewers advice, checked every instance of the word "code" and replaced the nomenclature where suitable.

- Please remove any mentions of internal labels within the code – this information is unneeded for a research paper audience: "halo_exchange subroutine", "compute_fluxzdiv", "this%geom%node2edge_sign".

We have removed the internal labels.

- Please limit the use of acronyms/short-forms, and remove those clearly unneeded: PSNC in Fig. 3, EBTI on page 24, GP_dynamics/SP_transforms/SI_solver/RAD in caption of Fig. 5, semi-Lagrange in Fig. 6; Some references are listed with DOI number, some without - please be consistent; FORTRAN/Fortran, TRAP2/Trap2 spelling - please be consistent; Among the affiliations listed, some are given with detailed street addresses, some without - please be consistent.

We have removed the mentioned acronyms and made the text and affiliations consistent. The DOI numbers are given whenever they are available.

- The title of the paper reads "The ESCAPE project: Energy-efficient Scalable Algorithms for Weather Prediction at Exascale". Exascale is not discussed or defined and barely mentioned only in the conclusions, while the phrase "weather & climate" is used throughout the paper.

We have added recent results on Summit and a discussion which is more targeted at exascale. The title of the paper is the name of the project.

- Statements such as "ECMWF is world leading in terms of track forecast", "extreme computational capabilities typically required in operational forecast production", "[IFS code] has been continuously optimized over multiple decades", "Feedback from the European and international community at our dissemination workshops and at international conferences has shown that this work was well received" are, in my opinion, good candidates for removal when shortening the paper – please avoid promotional language and statements which are not falsifiable; another candidates for removal are numerous vague statements: "most speedup seems to be due to avoiding some of the temporary arrays", "some more fundamental changes which are more difficult to apply", "whole cycle might employ some form of smoother/solver", "has to be wisely chosen according to the cluster hardware", "we do not know if there will be a clear winner", "The first results of this effort look promising".

Many of these statements were removed in the process of shortening the paper as suggested by all reviewers. We have revised the remaining statements.

**Response to the referee 2 for "The ESCAPE project: Energy-efficient Scalable Algorithms for Weather Prediction at Exascale"**

(grey background: text of the reviewer comment, white background: our response)

> The paper presents a review of the work done in the Escape project. This reviewer is familiar with some of this work and reporting the work done as part of the project is certainly of interest to the community. Unfortunately, the paper is not well written, if is full of mistakes, informal language and confusing or unclear explanations. I have read and documented changes as far as page 17, but this has taken a long time as the paper has not been properly proof read before submission. Referee #1 calls for a substantial revision, and a possible change of paper type, therefore there doesn't seem much point in fully detailing necessary changes beyond this point. A shorter, more focused article as a review and perspective paper would improve the readability and is probably more appropriate for the content.

We would like to thank reviewer 2 for his advice. We have significantly shortened the paper. The main purpose of the paper is to present the dwarf concept to the weather & climate community and to demonstrate it with a detailed example.

> I include the detailed points below, which need to be addressed.
>
> Page 2 line 7 Weather prediction (models?)

We have changed the sentence.

> Line 20, sentence reads as if heavy precipitation patterns could lead to tropical cyclones, modify

We have revised the introduction.

> Line 21 being satisfying "being"

We have changed this sentence.

> Intro 1st paragraph is rather clumsy, there are plenty of reasons improved forecasts in general would have economic and societal benefit besides heavy precipitation. The need to improve resolution is given as the main motivation for improved forecasts but then Climate is thrown into the following sentence. Improved resolution versus complexity for improved climate is a matter of debate. There is no mention of current resolutions for the reader to compare 1km. What does global resolution range mean? The last sentence is also confusingly written. This paragraph needs re-rewriting with proper thought on what is the motivation for improving resolution of weather and climate simulations. There are plenty of justifications.

We have revised the introduction.

> Line 27 "guarantee the continued efficiency" is probably a bit strong. "Enable efficient implementations of " or similar is probably more realistic.

We have made the suggested replacement.

Line 33 "The authors there" is rather informal language.

We have changed this sentence.

Page 3. Line 7 Citation needed to back authors claim that ECMWF is world leading in terms of track forecast. It would be quite odd for a higher resolution forecast to "compromise the accuracy" of a lower resolution, especially as this is the motivation for escape.

We have revised this part of the introduction.

The paragraph on supercomputers is also rather clumsily written and there should be a reference. An obvious one would be "Crossing the Chasm : how to develop weather and climate models for next generation computers?". Some of the authors of this paper are also authors of the chasm paper. Other references would also be beneficial.

We have revised the introduction and we have added the suggested reference.

Section 2, line 18, The use of "at once" suggests either "immediately" or "at the same time" but "too big" implies neither. This sentence needs re-writing.

We have changed this sentence.

Page 4 The text in figure 1 is far too small to read.

The small text in figure 1 contained only license terms of material that was used to create the figure. We tried to minimise the distraction from the message of the figure by making the text very small. We have now made the text larger to enable everyone to read the text.

Figure 2a "halo's" has no apostrophe.

We removed this figure from the paper in order to shorten the paper.

Page 5 figure 3 What does the vertical line denote? Line 5 what does "released dwarf" mean?

We have added an explanation of the vertical line to the caption and we have revised the text.

Page 7 line 5 "use 3D var like the " is too informal language, use for example or e.g.

We have removed this text in order to shorten the paper.

Page 7 line 23 "need" -> "needs"

We have removed this text in order to shorten the paper.

Page 8 Figure 8. The figures are not very well explained. A careful reading of a technical appendix is required and even then it is not clear how they illustrate the point that overlapping

the comms of data transfer is necessary. The main constraint for NWP is time to completion, it is not obvious that the assumptions (necessarily) made to express the scaling in terms of energy are sufficiently universal to infer the point the authors wish to make. Whilst using less energy reduces the cost, the run-time constraint is the primary motivation for NWP. This section needs to be re-worked to describe the performance model and explain how any figures used enable the authors draw the conclusions that they do.

We have removed the cost model in the process of shortening the paper. Just for clarification: all the curves in the plots of the cost model were created under the constraint that the forecast needs to finish within the required runtime. The cost model attempted to answer the question: under the given runtime constraint how much code needs to run on an accelerator to make the use of accelerators financially beneficial. As mentioned in the first version of the paper this cost model requires many assumptions and should only be used as a motivation to port as much of the model as possible to the accelerator. It may not be used as a basis for management decisions.

Page 9 section 3 line 23 Last sentence. What does this mean?

The inverse transform uses first an inverse Legendre transform to compute Fourier coefficients and applies then an FFT to obtain grid point values. The direct transform uses the opposite order: first an FFT to compute Fourier coefficients and then a direct Legendre transform to compute the spectral coefficients. We agree that the term "opposite direction" to describe this difference was misleading. We have revised this part of the text.

Page 10, refers to figure 5 which is on page 11. How are the profiles produced? What machine are they produced from, especially the node count etc would change the profile? The "Anticipated" future are these profiled from data or some performance model? The 1.25km run may well have a different profile on a different machine?

All of these profiles have been measured with the actual model on the Cray XC40 supercomputer of ECMWF. The number of nodes has been chosen such that the model run would be suitable for operational requirements. We agree that different machines will produce different profiles. These profiles are highly relevant for ECMWF because they represent the situation on the machine that is currently used for the operational forecast.

Page 11 refers to figure 7 which appears on page 13, the text "like on the left" is rather informal and inappropriate for a scientific paper.

We have changed the figure accordingly.

Page 12 refers to figure 8 which appears on page 14. In the figure, there is no open diamond referred to in the legend and the caption, the open rectangle referred to in the caption doesn't appear but a dash or line does which isn't referred to in the caption.

The open diamond of the legend is half covered by the filled circle in the plot. The open rectangle of the legend was an oversight and should have been the dash. We have made the filled circle open and changed the colour to make the overlapping points easier to recognise.

Pages 12 and 13 there is a complicated discussion of code changes. This would be illuminated by some code fragments as examples.

We have added code examples in section 3.3.

Figure 10. Page 16 The data points are connected with a line (something spreadsheet application does readily), however, the horizontal access is Number of GPUs, which is a discrete variable, so a line graph is wrong. Whilst it not unusual to see such a plot, the authors are not predicting the speed up on 16.5 GPUs so why the line? The plot should be re-drawn appropriately.

We agree with the reviewer that the connecting line between the data points is not supposed to provide a scientific message. We believe that the connecting line makes the plot more readable. We have reduced the thickness of the line by 50% to make it less prominent and we added a statement to the caption describing that the data points were connected with lines purely for the purpose of improving readability.

Page17 lines 6-15 the paragraph discusses the pack-unpack operation. The sentence sender and receiver share their memory layout as the may differ is confusing. How can they share a memory layout if they are different? What was the change that made the performance improvement? Were the pack and unpack scanning memory unnecessarily?

We have replaced "share" with "exchange" in the sense that sender and receiver exchange their own layout among each other. The performance improvement came from reordering the loops for both pack and unpack following the memory layout of the scattered buffer. This optimisation decreased the number of tests (i.e. copy or not copy) and avoids scanning memory multiple times. Scanning the memory multiple times was unnecessary.

The final paragraph on page 17 is a discussion of implementing some of the GPUs on CPUs. Again some code fragments here would be helpful. If the GPU optimisations are two intensive, how can they be used elsewhere for other architectures? The stated goal of Escape is to re-assemble the models from optimised dwarfs. How is this managed? If there are conflicting optimisations how are they resolved? Is single source code possible?

The GPU optimisation required a major redesign of the code which allowed to avoid transpositions of small temporary arrays. This optimisation can be applied as well to the CPU version. As a first step we applied it to a serial version of the spectral transform on CPU which is now used operationally at ECMWF and provides a major speedup as shown in the paper. Applying the same idea to the parallel CPU code requires more work that is planned in the future.

There is of course some risk that different optimisations are conflicting. We have not encountered this issue in ESCAPE so far. In the end we take whichever optimisation gives us the best performance in the full model. We currently have different source codes for CPUs and GPUs. We try to achieve a single source code through the use of the DSL except for highly specialised low level libraries like the spectral transforms.

Figure 11, the "barrier" optimisation is not clearly explained. Again, line plots of noncontinuous variable should be changed.

In the original code, some MPI barriers were present to profile the MPI communications at low level. By disabling these barriers (as allowed and documented in the code), a performance gain was identified. The main lesson learnt by this removal is the following: these barriers imply useless synchronisations which have the following consequences:

1. The application suffers from imbalance (as reported in D3.3 section 4.3.1.3 on a single node run). Thus, adding non mandatory synchronisation via barriers decreased the global performance (as each barrier implies to wait for the slowest process);

2. Moreover, these synchronisations created contention;

3. Last but not least, due to the two first points which change the behaviour of the application, there is a bias in the communications profiling. In other words, this profiling change the application behaviour.

As in the response to the comment on figure 10 we made the line thinner and added a statement to the caption.

**Response to the referee 3 for "The ESCAPE project: Energy-efficient Scalable Algorithms for Weather Prediction at Exascale"**

(grey background: text of the reviewer comment, white background: our response)

> General comment:
>
> In this manuscript an overview about the achievements in the ESCAPE project is given. The main concept is explained, some of the developments are explained in details and finally some tests are mentioned.
>
> Although I think that this manuscript is a valuable contribution for GMD, I cannot recommend to accept the manuscript in the actual state. The manuscript must be revised in a substantial way before it can be considered again. Therefore I recommend major revision of the manuscript. In the following I will explain my concerns.
>
> Major issues
>
> 1. Balance of the manuscript: The manuscript is very long and not really balanced. Some parts are explained in details, as e.g. the development of the MPDATA dwarf, but some parts are just mentioned. Especially for the very shortly explained parts, there are very often references to technical reports, i.e. documentation which is generally not peer reviewed. Although there are some performance tests, there is only one figure showing a test for atmospheric flows, and also this test is only marginally described.

The purpose of the paper is to present the concept of the dwarfs and to describe our work with a few detailed examples. Following the suggestions by reviewers 1 and 2 we have significantly shortened the paper. Having all of the details from the technical reports inside this paper would make it too long and would distract from the main message of the paper. To our knowledge GMD allows references to non peer reviewed technical reports if no peer reviewed reference is available.

The optimisations performed in the ESCAPE project do not affect the accuracy of the result. The figure which showed a comparison between finite volume and spectral transform method was an additional information. We removed this figure in our effort to shorten the paper.

> I would recommend to significantly reorganize the manuscript, maybe also considering to split the manuscript into three parts: First, an overview part, where mostly the concept and the new architecture can be explained in a concise way. Second, a model description part, i.e. a detailed description of the different parts of the model, especially of the parts, which are contained in the technical memoranda but not described in peer-reviewed literature. Third, a part dedicated to test cases for atmospheric flows - and maybe also clouds and radiation, since these parts are also included into the model.

The suggested structure of the paper does not fit to the intended purpose of the paper to introduce the dwarf concept and illustrate the workflow with a detailed example. According to the

advice from reviewer 1 and 2 we have significantly shortened the paper with the goal to make the intended message clearer. As stated before the results of test cases for atmospheric flows can be found in the literature given in the references. In this paper we allowed optimisations only if they did not affect the results in any significant way. Under this constraint a description of the optimisation and the performance analysis are sufficient to present our results.

> Especially test cases of atmospheric flows would be very interesting, since it is not clear if all the new models represent the atmospheric flow and other atmospheric phenomena in a physically consistent way. Therefore I highly recommend to use well-documented test cases for atmospheric flows, as e.g. Jablonowski & Williamson (2006). It would be interesting to see also tests for clouds and radiation, although I am not really aware of large scale tests, beyond the standard tests as e.g. Weismann & Klemp (1982).

As stated before our work on optimising code does not affect the accuracy of the results. Test cases for the underlying methods can be found in the literature referenced throughout the paper.

> 2. Selection of the dwarfs: It is not really clear how and why the different dwarfs were chosen. Although I think that this is a well chosen sample of possible models, it should be justified much better. Especially, the choice of the shallow water model is not really clear, because no real results of this model are shown in the manuscript. Therefore, I recommend to describe the choice of the models is a clearer way.

We have revised our conclusions section to make it clearer that we intend this paper to be a starting point and with a hope that the community will join our efforts and identify characteristic patterns in terms of computation and communication (dwarfs) and implement prototypes which can be used to work on optimising the key building blocks of weather & climate models.

> Minor issues:
>
> Cost model: The benefit of the cost model is not really clear to me. It is introduced in a comparable length as the dwarfs, but it is not really clear why this is so important for the whole manuscript, justifying a large part in the appendix.

The cost model was meant to illustrate the importance of porting as much of the model to the accelerator as possible. We have removed the cost model in order to shorten the paper as requested by reviewers 1 and 2.

---

## Referee Report (RR1)

This is a second round of review of the proposed GMD paper summarising the ESCAPE project. As the main conclusion from my earlier review of the initial version, I suggested to significantly reduce in scope and shorten the manuscript (50 pages, 30 figures). The length was clearly not the only deficiency, and the other two reviewers concurred. Yet, given the apt match between the ESCAPE project aims and the scope of GMD, the journal does seem to be a valid venue for promoting the project and allowing to give credit to its participants.

The current version is noticeably more readable, and indeed significantly shorter (28 pages, 13 figures), yet I still find that the work lacks some balance and style consistency. For instance, I doubt if the paper is a proper place to explain that one should care about index order in multi-dimensional loops or that temporary memory allocation should be done once and not within loops, or to verbosely enumerate parameters of new NVidia products (all on page 13). Similarly, I do not think there is any value in explaining to the readers that program performance benefited from removal of debug leftovers (p17/l5-6) or to promote machine-learning-technology suitability of optical processors (p20/l21-22). I recommend removing Figure 1 (if not, please reduce its resolution, currently page 5 weights over 2M, while the whole article pdf is less than 6M). Overall, please try not to limit further corrections to what is literally brought up in the reviews, but read through the whole text, and try to maintain a consistent style (e.g., passive voice vs. we-did narrative). Do not hesitate to shorten the paper even more.

Please also make sure that the abstract and conclusions convey correctly the main lessons learnt in the project and a take-home message from the paper. For example, DSLs and optical processors are not mentioned in the abstract. In the "Conclusions and outlook" section, I suggest to limit references to ESCAPE-2 to one single paragraph (perhaps the last one).

**The key remaining major concern is the necessity to deposit many of the referenced non-journal reports in persistent repositories. The current practice of solely providing links to pdfs on Google Drive with no metadata or guarantee of persistency is unacceptable (9 such references).**

Below, I'm listing more specific comments that might be useful.

- p1/l6: what (in this context) "scientifically required" means? consider rephrasing

- p1/l7: "algorithms should ... be ... resilient in case of ... software failure" – is it discussed, please elaborate what is meant here by software failure?

- p2/l1: "is leading" or "was leading"? (puzzling given references to ongoing ESCAPE-2)

- p2/l3: "under initiative FET-HPC" → "under the FET-HPC initiative"

- p2/l4: "goal is" or "goal was" (as above)

- p2/l4: "to next generation" → "to the next generation"

- p2/l8: "nearly all" → "multiple"?

- p2/l23: "$US" → "US dollars"

- p2/l28: "by so-called" → "by the so-called"

- p3/l16: does "above" refer to resolution or location in the text, please rephrase

- p3/l22: "thanks to" → "owing to"

- p3/l24: "As this ... down," → "Consequently,"

- p3/l29: explain what a "highly varying kernel" is, generally it would be worth to introduce the notion of kernel in the paper

- p3/l29: what is "scientific dependency"? (data dependency?)

- p3/l29: "rather complex algorithms" - please be more specific

- p3/l33: "algorithms and codes" → "algorithms and their implementations"

- p3/l34: "scientifically and computationally well defined" - "well defined" reads like "well posed" and suggests that other components are not "well defined", consider rephrasing

- p4/l1: add "herein introduced" before first mention of Weather & Climate dwarfs?

- p4/l7: comma after 2004

- p4/l3: motives → motifs

- p4/l24: runnable → standalone?

- p4/l1: "dwarfs created" → "dwarfs developed" or even better "List of ESCAPE dwarfs"?

- p4/l4-5: "For many of the dwarfs we created so called prototypes. Each prototype implementation ..." → "Prototype implementations addressing specific hardware were developed for selected dwarfs"?

- p5/l2: "and DSL..." → "and two domain-specific language (DSL) solutions: CLAW and GridTools."

- p5/l2: add a final sentence mentioning usage of Fortran

- p6/l12: "dwarfing" ??? (perhaps isolating)

- p8/l19: comma after methods

- p8/l25: "truncation" → "truncation error"

- p8/l25: "very cheap direct solve" → "efficient direct scheme"?

- p8/l16: comma after paper

- p8/l28: comma after Otherwise

- p10/Fig3 caption: "is without ocean and waves" → "was run without ocean coupling"?

- p10/l11: "overlap different parts of the model" → "enable concurrent execution of different model components"?

- p11/l2 and p11/l6: repetition of "parallelism exposed to the GPU", avoid duplication and explain what it means

- p11/l11: ensure listing is placed after "as follows" (or make it a figure)

- p11/11: "a factor 10", rephrase using "tenfold increase of"

- p12/l3: "a FFT" → "an FFT"?

- p15/l11: please remove the "Also there is still room ..." sentence

- p17/l5-6: please remove the "A few percent of ..." sentence

- p17/l13: comma after CPU

- p18/l2-4: suggest starting the section with the second sentence ("The fundamental...")

- p20/l1: "introducing known terms into the functions" – what terms, what functions?

- p20/l15-19: explain that binary means black-white (right?), otherwise binary precision might sound confusing

- p20/l21-22: remove the "The inherent ability ..." sentence

- p20/l23: please mention that results from the optical processing tests will be discussed later, otherwise it seems that the subject is abruptly dropped

- p21/l2: "is about 15 percent" of what?

- p21/l9-10: please remove "There might still be a potential" sentence

- p21/l13: rephrase not to repeat "necessary"

- p22/l2: "this section" → "preceding section"

- p22/l18-19: please remove "More work on"' sentence

- p22/22: "physics equations"?, please rephrase

- p22/l33: "scientific correctness" of dwarfs, implementations, hardware? (please limit the use of them/their/these)

- p23/l4-5: "The paper gives" sentence sounds like a copy-paste from an abstract

- p23/l32: Please make the "Comparing different methods..." sentence more specific

- general: please do not CAPITALISE Fortran – starting with Fortran 90 the all-caps name has been dropped

References (in general, be consistent: capitalise only the first word of a title; use proper journal name abbreviations):

- Asanović et al. 2006: www2 → www (both work, but people seem to cite the latter)

- Asanović et al. 2009: Comm. ACM

- Bénard and Glinton 2019: Q. J. Royal Meteorol. Soc.; add volume, pages

- Clement et al. 2018: remove "on"

- Colavolpe et al. 2017: Q. J. Royal Meteorol. Soc.

- Colella 2004: if inpossible to locate, mention in the text after whom cited

- Deconinck 2017a: use persistent repository (e.g., OpenAIRE, arXiv), **NOT A GOOGLE DRIVE LINK!**

- Deconinck 2017b: use persistent repository (e.g., OpenAIRE, arXiv), **NOT A GOOGLE DRIVE LINK!**

- Deconinck et al. 2017: Comput. Phys. Commun.

- Douriez et al. 2018: use persistent repository (e.g., OpenAIRE, arXiv), **NOT A GOOGLE DRIVE LINK!**

- Dziekan et al. 2019: GMDD → Geosci. Model Dev. 12, 2587-2606; doi:10.5194/gmd-12-2587-2019

- Feng et al. 2012: use full title (incl. "work in progress"); capital letters in conf. name

- Flamm 2018: add publisher and report number (NBER Working Paper No. 24553)

- Glinton and Bénard 2019: Q. J. Royal Meteorol. Soc.; add volume, pages (or remove elsewhere)

- Johnston and Milthorpe 2018: use full title (incl. "Enhancing OpenCL...");

- Kaltofen 2011: add quotation marks within title as in the original; use booktitle "Numerical and Symbolic Scientific Computing" instead of book series "Texts & Monographs in Symbolic Computation"; add editor names; remove Vienna

- Katzav and Parker 2015: Clim. Change

- Krommydas et al. 2015: remove "chun" in the surname of second author; J. Signal Process. Syst.

- Kühnlein et al. 2019: Geosci. Model Dev.

- Macfaden et al 2017: Sci. Rep.

- Mazauric et al. 2017a: use persistent repository (e.g., OpenAIRE, arXiv), **NOT A GOOGLE DRIVE LINK!**

- Mazauric et al. 2017b: use persistent repository (e.g., OpenAIRE, arXiv), **NOT A GOOGLE DRIVE LINK!**

- Mengaldo 2016: use persistent repository (e.g., OpenAIRE, arXiv), **NOT A GOOGLE DRIVE LINK!**

- Mengaldo et al. 2018: Arch. Comput. Methods Eng.

- Messer et al. 2016: Int. J. High Perform. Comput. Appl

- Michalakes et al. 2015: is the "Tech. Rep. TN-484, NCAR, Boulder" correct? NCAR does not seem to list it, please use a (hopefully) more persistent url: https://repository.library.noaa.gov/view/noaa/18654

- Müller et al. 2017: use persistent repository (e.g., OpenAIRE, arXiv), **NOT A GOOGLE DRIVE LINK!**

- Müller et al. 2018: add volume, pages

- Neumann et al. 2019: Philos. Trans. Royal Soc. A; remove 20180 148

- Osuna 2018: use persistent repository (e.g., OpenAIRE, arXiv), **NOT A GOOGLE DRIVE LINK!**

- Robinson et al. 2016: Prace → PRACE; add doi: 10.5281/zenodo.832025; remove "- Evaluations on Intel MIC"?;

- Schalkwijk et al. 2015: Bull. Am. Meteorol. Soc.

- Schulthess et al. 2019: Comput. Sci. Eng.

- Shukla et al. 2010: Bull. Am. Meteorol. Soc.

- Van Bever et al. 2018: use persistent repository (e.g., OpenAIRE, arXiv), **NOT A GOOGLE DRIVE LINK!**

- Wallemacq et al. 2018: correct author list (Wallemacq, P. and House, R.); use publisher's url instead of ResearchGate doi: https://www.unisdr.org/we/inform/publications/61119

- Wedi et al. 2015: add doi: 10.21957/thtpwp67e

- Wehner et al. 2011: J. Adv. Model. Earth Sys.

- Xiao et al. 2017: add doi:10.21957/g9mjjlgeq

- Zheng 2018: correct volume, pages, doi: Geosci. Model Dev., 11, 3409-3426, doi:10.5194/gmd-11-3409-2018

Hope that helps.

---

## Author Response (AR2)

**Second revision of the paper "The ESCAPE project: Energy-efficient Scalable Algorithms for Weather Prediction at Exascale"**

**Response to the referee 1**

(grey background: text of the reviewer comment, white background: our response)

> This is a second round of review of the proposed GMD paper summarising the ESCAPE project.
>
> As the main conclusion from my earlier review of the initial version, I suggested to significantly reduce in scope and shorten the manuscript (50 pages, 30 figures). The length was clearly not the only deficiency, and the other two reviewers concurred. Yet, given the apt match between the ESCAPE project aims and the scope of GMD, the journal does seem to be a valid venue for promoting the project and allowing to give credit to its participants.
>
> The current version is noticeably more readable, and indeed significantly shorter (28 pages, 13 figures), yet I still find that the work lacks some balance and style consistency. For instance, I doubt if the paper is a proper place to explain that one should care about index order in multi-dimensional loops or that temporary memory allocation should be done once and not within loops, or to verbosely enumerate parameters of new NVidia products (all on page 13). Similarly, I do not think there is any value in explaining to the readers that program performance benefited from removal of debug leftovers (p17/l5-6) or to promote machinelearning-technology suitability of optical processors (p20/l21-22). I recommend removing Figure 1 (if not, please reduce its resolution, currently page 5 weights over 2M, while the whole article pdf is less than 6M). Overall, please try not to limit further corrections to what is literally brought up in the reviews, but read through the whole text, and try to maintain a consistent style (e.g., passive voice vs. we-did narrative). Do not hesitate to shorten the paper even more.

We thank the reviewer for the thorough reading of the revised manuscript and the helpful review. We streamlined and shortened the presentation, and carefully re-read the whole paper while improving the balance and consistency of the presentation. Although we trimmed the original text quite substantially, we also had to add some text to address the reviewers' comments, so the revised paper is only one page shorter than its previous version. We also reduced the size of Figure 1 substantially. The entire paper has now a size of less than 2MB.

> Please also make sure that the abstract and conclusions convey correctly the main lessons learnt in the project and a take-home message from the paper. For example, DSLs and optical processors are not mentioned in the abstract. In the "Conclusions and outlook" section, I suggest to limit references to ESCAPE-2 to one single paragraph (perhaps the last one).

We revised the conclusions and added a paragraph including DSLs and optical processors to the abstract.

> The key remaining major concern is the necessity to deposit many of the referenced non-journal reports in persistent repositories. The current practice of solely providing links to pdfs on Google Drive with no metadata or guarantee of persistency is unacceptable (9 such references).

We published all of these documents on arxiv.org and adjusted the references in the paper accordingly.

> Below, I'm listing more specific comments that might be useful.
>
> p1/l6: what (in this context) "scientifically required" means? consider rephrasing

Scientifically required accuracy is for us the level of accuracy required to adequately represent the physical processes. We revised this sentence in the paper.

> p1/l7: "algorithms should ... be ... resilient in case of ... software failure" – is it discussed, please elaborate what is meant here by software failure?

We removed the term "software".

> p2/l1: "is leading" or "was leading"? (puzzling given references to ongoing ESCAPE-2)

We changed this to "was leading".

> p2/l3: "under initiative FET-HPC"  →  "under the FET-HPC initiative"

We revised the text.

> p2/l4: "goal is" or "goal was" (as above)

We decided to use "goal was" since the paper focusses on the ESCAPE 1 project which has ended.

> p2/l4: "to next generation"  →  "to the next generation"
>
> p2/18: "nearly all"  →  "multiple"?
>
> p2/l23: "$US"  →  "US dollars"
>
> p2/l28: "by so-called"  →  "by the so-called"
>
> p3/l16: does "above" refer to resolution or location in the text, please rephrase
>
> p3/l22: "thanks to"  →  "owing to"
>
> p3/l24: "As this ... down,"  →  "Consequently,"

We revised the text as suggested in these seven comments.

> p3/l29: explain what a "highly varying kernel" is, generally it would be worth to introduce the notion of kernel in the paper

The term kernel refers to compute kernels. We explain this now in the paper.

> p3/l29: what is "scientific dependency"? (data dependency?)

Scientific dependency meant the dependency due to data as well as due to physical processes. We revised the text accordingly.

> p3/l29: "rather complex algorithms" - please be more specific
>
> p3/l33: "algorithms and codes" → "algorithms and their implementations"
>
> p3/l34: "scientifically and computationally well defined" - "well defined" reads like "well posed" and suggests that other components are not "well defined", consider rephrasing
>
> p4/l1: add "herein introduced" before first mention of Weather & Climate dwarfs?
>
> p4/l7: comma after 2004
>
> p4/l3: motives → motifs
>
> p4/l24: runnable → standalone?
>
> p4/l1: "dwarfs created" → "dwarfs developed" or even better "List of ESCAPE dwarfs"?
>
> p4/l4-5: "For many of the dwarfs we created so called prototypes. Each prototype implementation ..." →"Prototype implementations addressing specific hardware were developed for selected dwarfs"?
>
> p5/l2: "and DSL..." → "and two domain-specific language (DSL) solutions: CLAW and GridTools."
>
> p5/l2: add a final sentence mentioning usage of Fortran
>
> p6/l12: "dwarfing" ??? (perhaps isolating)
>
> p8/l19: comma after methods

We followed these 13 suggestions and revised the text accordingly.

> p8/l25: "truncation" → "truncation error"

We revised the text. The truncation in this context is not related to the error but rather to the number of wavenumbers included in the transform and the resulting problem size.

> p8/l25: "very cheap direct solve" → "efficient direct scheme"?
>
> p8/l26: comma after paper
>
> p8/l28: comma after Otherwise
>
> p10/Fig3 caption: "is without ocean and waves" → "was run without ocean coupling"?
>
> p10/l11: "overlap different parts of the model" → "enable concurrent execution of different model components"?

We revised the text as suggested in these five comments.

> p11/l2 and p11/l6: repetition of "parallelism exposed to the GPU", avoid duplication and explain what it means

The code needs to be restructured in such a way that the GPU is enabled to take full advantage of the inherent parallelism of the algorithm. We revised the text to make this clearer.

> p11/l11: ensure listing is placed after "as follows" (or make it a figure)

We removed this listing in order shorten the paper. We believe the message of the listing is well described by the text.

> p11/11: "a factor 10", rephrase using "tenfold increase of"
>
> p12/l3: "a FFT" → "an FFT"?

We revised the text as suggested in these two comments.

> p15/l11: please remove the "Also there is still room ..." sentence
>
> p17/l5-6: please remove the "A few percent of ..." sentence

We removed these two sentences.

> p17/l13: comma after CPU
>
> p18/l2-4: suggest starting the section with the second sentence ("The fundamental...")
>
> p20/l1: "introducing known terms into the functions" – what terms, what functions?

We revised the text as suggested in these three comments.

> p20/l15-19: explain that binary means black-white (right?), otherwise binary precision might sound confusing

Binary precision means that only bright and dark values are distinguished. We revised the text.

> p20/l21-22: remove the "The inherent ability ..." sentence

We removed this sentence.

> p20/l23: please mention that results from the optical processing tests will be discussed later, otherwise it seems that the subject is abruptly dropped

We revised the text.

> p21/l2: "is about 15 percent" of what?

The 15 percent refer to the portion of the overall runtime. We revised the text.

> p21/l9-10: please remove "There might still be a potential" sentence

We removed this sentence.

> p21/l13: rephrase not to repeat "necessary"

We replaced it with "required".

> p22/l2: "this section" → "preceding section"

In Latex terminology it is the preceding subsection but still the same section. We added the section number to be precise in our text.

> p22/l18-19: please remove "More work on"' sentence

We removed this sentence.

> p22/22: "physics equations"?, please rephrase

We revised the text.

> p22/l33: "scientific correctness" of dwarfs, implementations, hardware? (please limit the use of them/their/these)

The scientific correctness referred to the implementations. We revised the text.

> p23/l4-5: "The paper gives" sentence sounds like a copy-paste from an abstract

We revised this sentence.

> p23/l32: Please make the "Comparing different methods..." sentence more specific

We revised the text.

> general: please do not CAPITALISE Fortran – starting with Fortran 90 the all-caps name has been dropped

We changed all occurrences of Fortran.

> References (in general, be consistent: capitalise only the first word of a title; use proper journal name abbreviations):
>
> Asanovic et al. 2006: www2 → www (both work, but people seem to cite the latter)
>
> Asanovic et al. 2009: Comm. ACM
>
> Benard and Glinton 2019: Q. J. Royal Meteorol. Soc.; add volume, pages
>
> Clement et al. 2018: remove "on"
>
> Colavolpe et al. 2017: Q. J. Royal Meteorol. Soc.
>
> Colella 2004: if impossible to locate, mention in the text after whom cited

Deconinck 2017a: use persistent repository (e.g., OpenAIRE, arXiv), NOT A GOOGLE DRIVE LINK!

Deconinck 2017b: use persistent repository (e.g., OpenAIRE, arXiv), NOT A GOOGLE DRIVE LINK!

Deconinck et al. 2017: Comput. Phys. Commun.

Douriez et al. 2018: use persistent repository (e.g., OpenAIRE, arXiv), NOT A GOOGLE DRIVE LINK!

Dziekan et al. 2019: GMDD → Geosci. Model Dev. 12, 2587-2606; doi:10.5194/gmd-12-2587-2019

Feng et al. 2012: use full title (incl. "work in progress"); capital letters in conf. name

Flamm 2018: add publisher and report number (NBER Working Paper No. 24553)

Glinton and Benard 2019: Q. J. Royal Meteorol. Soc.; add volume, pages (or remove elsewhere)

Johnston and Milthorpe 2018: use full title (incl. "Enhancing OpenCL...");

Kaltofen 2011: add quotation marks within title as in the original; use booktitle "Numerical and Symbolic Scientific Computing" instead of book series "Texts & Monographs in Symbolic Computation"; add editor names; remove Vienna

Katzav and Parker 2015: Clim. Change

Krommydas et al. 2015: remove "chun" in the surname of second author; J. Signal Process. Syst.

Kuhnlein et al. 2019: Geosci. Model Dev.

Macfaden et al 2017: Sci. Rep.

Mazauric et al. 2017a: use persistent repository (e.g., OpenAIRE, arXiv), NOT A GOOGLE DRIVE LINK!

Mazauric et al. 2017b: use persistent repository (e.g., OpenAIRE, arXiv), NOT A GOOGLE DRIVE LINK!

Mengaldo 2016: use persistent repository (e.g., OpenAIRE, arXiv), NOT A GOOGLE DRIVE LINK!

Mengaldo et al. 2018: Arch. Comput. Methods Eng.

Messer et al. 2016: Int. J. High Perform. Comput. Appl

Michalakes et al. 2015: is the "Tech. Rep. TN-484, NCAR, Boulder" correct? NCAR does not seem to list it, please use a (hopefully) more persistent url: https://repository.library.noaa.gov/view/noaa/18654

Muller et al. 2017: use persistent repository (e.g., OpenAIRE, arXiv), NOT A GOOGLE DRIVE LINK!

Muller et al. 2018: add volume, pages

Neumann et al. 2019: Philos. Trans. Royal Soc. A; remove 20180 148

Osuna 2018: use persistent repository (e.g., OpenAIRE, arXiv), NOT A GOOGLE DRIVE LINK!

Robinson et al. 2016: Prace →PRACE; add doi: 10.5281/zenodo.832025; remove "-Evaluations on Intel MIC"?;

Schalkwijk et al. 2015: Bull. Am. Meteorol. Soc.

Schulthess et al. 2019: Comput. Sci. Eng.

Shukla et al. 2010: Bull. Am. Meteorol. Soc.

Van Bever et al. 2018: use persistent repository (e.g., OpenAIRE, arXiv), NOT A GOOGLE DRIVE LINK!

Wallemacq et al. 2018: correct author list (Wallemacq, P. and House, R.); use publisher's url instead of ResearchGate doi: https://www.unisdr.org/we/inform/publications/61119

Wedi et al. 2015: add doi: 10.21957/thtpwp67e

Wehner et al. 2011: J. Adv. Model. Earth Sys.

Xiao et al. 2017: add doi:10.21957/g9mjjlgeq

Zheng 2018: correct volume, pages, doi: Geosci. Model Dev., 11, 3409-3426, doi:10.5194/gmd-11-3409-2018

We made all of the suggested changes and we uploaded the technical ESCAPE reports to arxiv.org.

Hope that helps.

**Response to the referee 2**

(grey background: text of the reviewer comment, white background: our response)

We thank the reviewer for the careful reading of the revised manuscript and the helpful comments. We accommodated all the comments in the letter or spirit. We also streamlined and shortened the presentation, and carefully re-read the whole paper with the aim to improve the presentation following guidance of both reviewers' comments. Please find the detailed point-by-point reply below.

> In the abstract.
>
> "Satisfy strict service requirements in terms of time-to-solution and energy-to-solution". This reviewer is unaware of any service agreements for energy to solution. For any given machine, there is a time-to-solution requirement. Exploiting architectures which are more energy efficient for a given time-to-solution are explored here, but that is not the same thing. Can the authors clarify what they mean or change this sentence.

How much money can be spent on energy is limited by the budget of the institution which is paying for the supercomputer. The energy consumption has a direct impact on the decision which supercomputer the institution can afford to buy. We revised the text to make this aspect clearer.

> Page 3 Line 29 "highly varying kernels", What does this mean. Also the sentence is rather long and unwieldy and could benefit from restructuring.

Different compute kernels used in weather and climate prediction models have very different computational characteristics. We revised the text to make this clearer.

> Page 4 Line 13 Motifs is a plural of Motif. English is an elastic language whose usage does change and evolve and it maybe that motives is becoming an accepted plural of motif (I have seen this usage elsewhere). It could be confused motives meaning reasons for doing something. This reviewer suggests using Motifs for clarity.
>
> Page 10, last sentence. "The only true solution ..." this is an odd phrase, perhaps "One approach to avoiding …." Is better.

We revised the text as suggested in these two comments.

> Page 10/11 and figure 4. The results of Michalakes et al 2015 are used to show the amount of data that has to be communicated. Is this data from model run, or is it based on scaling from a smaller run in the paper. If it is the latter, it needs to be made explicitly clear and again, if it is the latter, is it the scaling model from the Michalakes paper?

This figure is extrapolating the results from Michalakes. We revised the text to make this clearer.

> Page 11, line 11. This sentence doesn't appear to connect with anything else and should be removed.

The sentence belonged to the listing which by accident got moved to the next page. We removed the code example in order to shorten the paper. We believe that the text explains the message of the example well enough.

> Page 12 figure 5.

> The red open circle and green open diamond sit atop one another. This makes them hard to see and distinguish. Can this be described in the caption to make it clear. How is the operational intensity determined for a whole time-step? What was the non-optimised performance of the Kernels only and Matmult.

We revised the caption. The operational intensity was derived from measurements with nvprof as well as from counting the number of floating point operations by hand. Combining the counted floating point operations with the measured memory traffic gives us the operational intensity. Numbers for the non-optimised kernels only are not readily available.

> Page 17 figure 9a
>
> Why is the x-axis not "node valued"? i.e. 1 and 2 are separated by 1 unit but so is 20-24 and 24-30. Figure 9b has a correct scale.

We corrected Figure 9a.

> Section 3.6 What is the conclusion to the work on the comparison between processors for run-time and energy consumption?

We added a statement about the conclusion of this comparison.

> Page 22 Line 17
>
> The DSL allows to perform optimisations —> the DSL allows optimisations to be performed
>
> Page 23 lines 22 and 23
>
> Referring to papers with "like in" is rather informal language for a scientific paper.
>
> See, or see for example is better.
>
> Page 23 line 32
>
> Requires to include all costs —> requires all costs to be included.

We revised the text as suggested in these three comments.

[revised manuscript text omitted]